# Vanadium and Cobalt Occurrence in the Fe-Ti-V Oxide Deposits Related to Mesoproterozoic AMCG Complex in NE Poland

Stanisław Z. Mikulski *, Katarzyna Sadłowska, Janina Wiszniewska and Rafał Małek

Polish Geological Institute—National Research Institute, Rakowiecka 4, 00-975 Warszawa, Poland;
katarzyna.sadlowska@pgi.gov.pl (K.S.); janina.wiszniewska@pgi.gov.pl (J.W.); rafal.malek@pgi.gov.pl (R.M.)
* Correspondence: stanislaw.mikulski@pgi.gov.pl

**Abstract:** On the basis of geochemical whole-rock and mineralogical point analyses, the concentrations of V and Co were determined in magnetite-ilmenite oxide ores, associated with sulphides, at the Krzemianka and Udryn deposits in the Mesoproterozoic Suwałki Anorthosite Massif (SAM) in NE Poland. EPMA analyses showed that the main carrier of vanadium was magnetite (mean = 0.42 wt%) and, to a lesser extent, ilmenite (mean = 0.14 wt%) and minor Al-spinels (mean = 0.04 wt%). In turn, cobalt was found mainly in the form of isomorphic substitutions in magmatic sulphides such as pentlandite (mean = 4.41 wt% Co), pyrrhotite (mean = 0.16 wt%), and chalcopyrite (mean = 0.11 wt%). Moreover, Co-enrichments were also recognized in the secondary sulphides, such as pyrite and bravoite, replacing pyrrhotite (means = 1.6 and 2.7 wt% Co, respectively), and in the form of different thiospinels $((Fe, Ni) (Co, Ni)_2S_4)$, mainly siegenite (mean = 22.0 wt% Co), replacing pyrrhotite and pentlandite. Vanadium cations were substituted in Fe, Ti oxide minerals in place of $Fe^{+3}$ cations, and in the case of cobalt, $Fe^{+2}$ cations were substituted in sulphides and thiospinels. Vanadium and cobalt showed high Person's correlation coefficients ($r$ = 0.70), indicating their close spatial coexistence and a common source, which was parental anorthosite-norite magma of the SAM suites. The common magma genesis of magnetite-ilmenite and sulphide mineralization was also confirmed by the very similar shapes of the curves of REE content in the oxide-sulphide ores in relation to chondrite, in which negative Eu anomalies and positive Sm anomalies are clearly visible. Although the average contents of vanadium and cobalt were low (arithmetic means = 960 ppm, and 122 ppm, respectively), the resources of these metals were estimated to be large due to the enormous reserves of magnetite-ilmenite ores hosted by the SAM. However, the Fe-Ti-V ores associated with Fe, Ni, Co, and Cu sulphides were considered to be sub-economic because of their depth of occurrence (mainly 1.0 km below the surface level); their metal contents, which were usually too low; and additionally the fact that the location is in a highly environmentally protected landscape and lake area.

**Keywords:** trace elements; strategic elements; vanadium; cobalt; Fe-Ti-V deposits; anorthosites; norites; Mesoproterozoic AMCG; Poland

## 1. Introduction

The development of modern green technologies and the pursuit of a zero-emissions economy have necessitated the use of critical elements on a much larger scale than before [1,2]. These elements which are defined as critical generally do not form independent deposits but accompany another major metallic element in the ores, in the form of isomorphic substitution in other major minerals. The most recent list of elements considered critical for the European Union includes vanadium and cobalt [3]. Cobalt is used in numerous metallurgical and chemical applications, mainly in cathode materials for rechargeable batteries and as super alloys that exhibit outstanding strength and surface stability at high temperatures [4,5].

The land-based cobalt resources identified worldwide constitute about 25 million tons [6]. Three main types of metal deposits supply almost 90% of cobalt to world mar-

kets; these are sediment-hosted stratiform Cu deposits, Ni-bearing laterite deposits, and magmatic Ni-Cu-Co sulphide deposits hosted in mafic and ultramafic rocks [6]. During magma crystallization, cobalt behaves similarly to nickel, crystallizing in sulphides. The main carrier of cobalt is pentlandite *(Ni, Fe, Co)$_9$S$_8$*, in which the Co content ranges from below one to several percent by weight. The average content of cobalt in igneous rocks is 23 ppm [7]. The rocks that are richest in Co are ultrabasic rocks, dunites, and peridotites (127–148 ppm Co), whereas norites and gabbros contain on average 50 to 60 ppm Co, and granites contain approximately 5 ppm [7]. In the hydrothermal stage, a large group of cobalt sulphides, arsenides, antimonides and arsenic, and antimony sulphides are formed. The most common among these are cobaltite (up to 36 wt% Co), glaucodot (up to 31 wt% Co), danaite (3–10 wt% Co), safflorite (~28 wt% Co), skutterudite (up to 30 wt% Co), siegenite (20–30 wt% Co), bravoite (up to 6 wt% Co), and costibite (~27 wt% Co).

Vanadium is one of the most important alloying additives in the production of full-alloy and high-strength low-alloy steels. The average vanadium content in the continental crust is high, at approximately 150 ppm [8]. Vanadium occurs through isomorphic substitution in numerous minerals, but these minerals are disseminated as accessory minerals in rocks. Among the orthomagmatic deposits, which are genetically related to ultrabasic (dunite) and basic rocks (gabbro, norite, and anorthosite), the content of $V_2O_5$ ranges from 0.2 to 2 wt%, respectively. The main V carriers in these deposits are generally titanium-rich magnetite (*$Fe^{2+}$ ($Fe^{3+}$, $Ti$)$_2O_4$*) and ulvöspinel (*$TiFe_2O_4$*) and ilmenite (*$FeTiO_3$*). Vanadium ($V^{3+}$) forms a spinel-coulsonite (*$V_2FeO_4$*) an isomorphic series with other spinels, such as magnetite and chromite, and therefore $Fe^{3+}$ may be substituted by $V^{3+}$ in these minerals. Considerable admixtures of $V^{3+}$ or $V^{4+}$ are present in dark rock-forming minerals [8]. In orthomagmatic Fe-Ti(-V) deposits, a typical grade of titanomagnetite ore is 0.2–1 wt% $V_2O_5$ (max. 1–2 wt%) [8]. The world's resources of vanadium exceed 63 million tons [6]. Additional sources of vanadium are titaniferous magnetite, phosphate rock, uraniferous sandstone and siltstone, bituminous sands, and heavy crude oil [6]. One of the main types of vanadium-supplying deposits is orthomagmatic Fe-Ti-V oxide, which is accompanied by sulphide mineralization. The presence of polymetallic sulphides (Fe, Cu, Co, and Ni) in this type of deposit can also be a source of cobalt, which can be present in the form of its own sulphides and isomorphic substitutions in other sulphides.

This type of Fe-Ti-V deposit was documented in NE Poland in the 1960s up to the 1980s. However, detailed data on the economic potential of these deposits for critical elements were poorly provided during exploration. Hence, the aim of this study was to determine the whole rock contents of V and Co in mineralized samples from the Krzemianka and Udryn Fe-Ti-V deposits in the Suwałki Anorthosite Massif (SAM) in Poland. The results of electron microprobe analyses of selected minerals are also presented in order to show the residence sites of these elements.

## 2. Geological Setting

### 2.1. Regional Geology

The Krzemianka and Udryn Fe-Ti-V ore deposits are located in the Mesoproterozoic Suwałki Anorthosite Massif (SAM) in the NE part of Poland [9–13], within the Mesoproterozoic beltiform magmatic AMCG (Anorthosite–Mangerite–Charnockite-Granite rapakivi) suite known as the Mazury Complex (Figure 1) [9]. This is a belt of granitoids and associated mafic and intermediate igneous rocks following an E–W-trending lineament extending from the Baltic Sea through northern Poland and southern Lithuania to western Belarus [14]. Anorthosite occurs at three autonomous massifs: Sejny, Suwałki, and Kętrzyn. The SAM has an oval shape and occupies an area of 250 km$^2$. Most of the crystalline basement of NE Poland is represented by late Svecofennian (1.84–1.80 Ga) orogenic granitoids and supracrustal succession [15]. All the abovementioned units were intruded by plutons of the Mesoproterozoic AMCG suite around 1.5 Ga [15]. This suite is dominated by A-type granitoids with a rapakivi-like texture. The subsequent intrusions are gabbro-norite, anorthosite, and locally mangerite and charnockite rocks. The central part of the SAM consists of

anorthosites surrounded by rings of norites, gabbronorites, diorites, and granites (Figure 1). The formation of the AMCG suite was a complex process with multiple magma batches sequentially differentiating and probably undergoing mixing and crustal assimilation [16]. A network of fractures and fault zones cutting the SAM rock sequences was filled by low-temperature S-type granite and pegmatite veins, emplaced between 1495 ± 15 Ma (Udryn, Jeleniewo) and 1488 ± 5 Ma (Udryn) [15]. These younger tectonic structures, which are widespread within the SAM, postdate the last episode within the SAM. New insights also allowed for the description of the occurrence of even later jotunite and nelsonite dikes within the massif [17,18]. Nelsonites are ore-bearing apatite rocks formed through a liquid immiscibility mechanism [19,20]. They were encountered as small dikes or veins in the SAM, as the final stage of ore mineralization. All these crystalline basement rocks are covered by 750–1200 m of sedimentary Phanerozoic rock complexes that dip towards the SW border of the East European Craton [21]. The sedimentary cover is dominated by sandstones, shales, and mudstones deposited in the Ediacaran to early Cambrian and in the early Triassic. The Cretaceous unit is built up mostly of chalk-type facies carbonates. The Cenozoic rock s are composed of unconsolidated, clastic sediments [12].

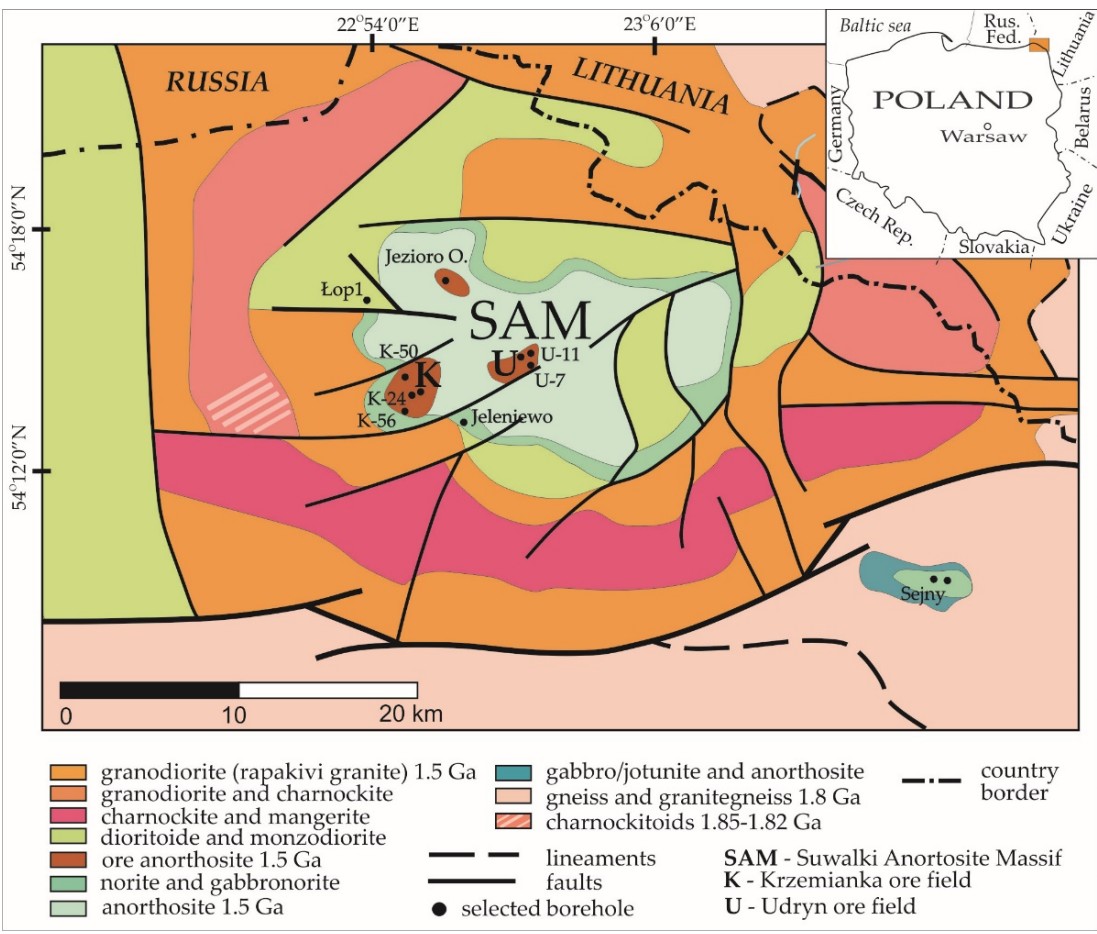

**Figure 1.** Schematic geological map (without Palaeozoic–Cenozoic sedimentary rock cover) of the Suwałki Anorthosite Massif (modified after [9]) with the location of the Krzemianka and Udryn Fe-Ti-V deposits in NE Poland.

## 2.2. Geology of Fe-Ti-V Oxide Deposits

The Fe-Ti-V deposits were discovered in the NE part of Poland within the crystalline basement mafic rocks through an intense drilling exploration program launched in the 1960s–1980s. Two large Fe-Ti-V deposits, the Krzemianka and Udryn deposits, were documented within the eastern part of the Mesoproterozoic SAM of the AMCG affinity

(Figure 1). The AMCG suite is unconformable, covered by Mesozoic-Cenozoic sedimentary rocks of thicknesses varying from 500 m on the east to 1000 m on the north-west [9,12]. Anorthosite and norite intrusions compose the core part of the massif and are surrounded by diorites and gabbronorites [9,12]. The SAM is divided by major faults of variable direction into three tectonic blocks [22]. The western one, with the Krzemianka and Udryn Fe-Ti-V deposits, is uplifted.

The ilmenite-magnetite ores are located in tectonic nodes and fill tectonic fractures at the gabbronorite-diorite-anorthosite contacts [21,22]. The deposits under consideration here are made of titanium-and vanadium-bearing magnetite, containing ilmenite and hematite-ilmenite, occurring in various proportions from 1:1 to 1:10 [23]. Hematite-ilmenite series can be found, disseminated in the host rocks. Ilmenite grains or aggregates contain two systems of hematite exsolutions [23–27]. Five types of Fe-Ti ores were distinguished: norite ore and poor pyroxene ores (15–25% Fe), pyroxene ores (22–25% Fe), plagioclase ores (25–35% Fe), spinel ores (35–48% Fe), and brecciated ores [28]. The richest parts of orebodies, called "ferrolites", are characterized by Fe-Ti oxide concentrations exceeding 40% of the rock volume. The ore-oxides represent two solid solution series: magnetite-ulvöspinel and ilmenite-hematite, depending on the P-T and oxidation conditions in Fe-Ti-V ores and their host rocks [23,24,29].

In total, the Krzemianka and Udryn deposits have resources estimated at about 1.34 billion tons of Fe-Ti-V ore, containing approximately 388.2 million tons of iron ($Fe_2O_3$), approx. 98 million tons of titanium ($TiO_2$), and approximately 4.1 million tons of vanadium ($V_2O_5$), [30]. They are classified as sub-economic on account of their low metal contents, especially of vanadium (0.26–0.31% $V_2O_5$ on average) and because of their occurrence at a depth exceeding 300 m [31]. However, the Fe-Ti-V mineralization contains lower contents of Fe-Cu-Ni-Co sulphides (1–3% by rock volume) and some amounts of REE carbonates and other trace elements [32–36]. The obtained Re-Os model age for pyrite, pyrrhotite and magnetite is 1536 ± 67 million years [37,38]. The Krzemianka and Udryn deposits belong genetically to the iron deposits of the Allard Lake type [23]. In addition, sub-economic Fe-Ti-V mineralization has also been identified at the Jezioro Okrągłe and Jeleniewo prospects [28].

### 2.2.1. The Krzemianka Fe-Ti-V Oxide Deposit

The Krzemianka deposit has an arcuate shape (2.5 km wide and 5 km long), with an area of approximately 13.2 km$^2$, and is located in the western part of SAM, close to its contact with granite and dioritoide rock cover (Figure 1) [28]. The deposit lies under the Cambrian, Permian, Triassic, Jurassic, Cretaceous, and Cenozoic sediments; the sedimentary cover may reach a total thickness of about 900 m. The Fe-Ti-V oxide ores in the deposit lie at depths of 850 to 2300 m; however, the majority of ore lies at a depth of 1100–1700 m [28]. The largest orebody is hosted by anorthosite. The orebodies are in the form of lenses, schlieren, pseudo-seams, and veins dipping generally at an angle of about 45° towards the SE (Figure 2), [28,39]. Their orebodies are variable in size. Their thicknesses vary from a few centimetres to 150 m. The contacts of orebodies with anorthosites and diorites are sharp [28]. The construction of the south-western ore zone in the upper part is made of norites and anorthosites which gradually turn downwards from rich ores to poor mineralization. The lower part consists of plagioclase-type ore, the ore bodies having sharp boundaries with the anorthosites. Plagioclase-type ore has been distinguished for macroscopic classification, e.g., a massive titanomagnetite ore mixture with anorthosite of labradore-andesine composition [23,28].

The deposit is made up of titanium- and vanadium-bearing magnetite, containing ilmenite and hematite-ilmenite, occurring in various proportions from 1:1 to 5:1 and even 10:1 [28,39]. Magnetite contains many decomposition products, including ulvöspinels, ilmenite, and Al-spinel solid solutions [23,39]. The Fe-Ti-V mineralization is accompanied by Fe and Cu sulphides: pyrrhotite, chalcopyrite, pyrite, marcasite, cubanite, chalcocite, and Ni and Co sulphides such as pentlandite, bravoite, millerite, linnaeite, and violar-

ite [23,26–28]. Sulphides may form up to 1–3% of the volume of the Fe-Ti-V ores. In the Krzemianka deposit, about 1.07 billion tons of ilmenite-magnetite ore containing vanadium have been documented. This Fe-Ti-V mineralization has an average grade of 27% $Fe_2O_3$, 7% $TiO_2$, and 0.3% $V_2O_5$ [30].

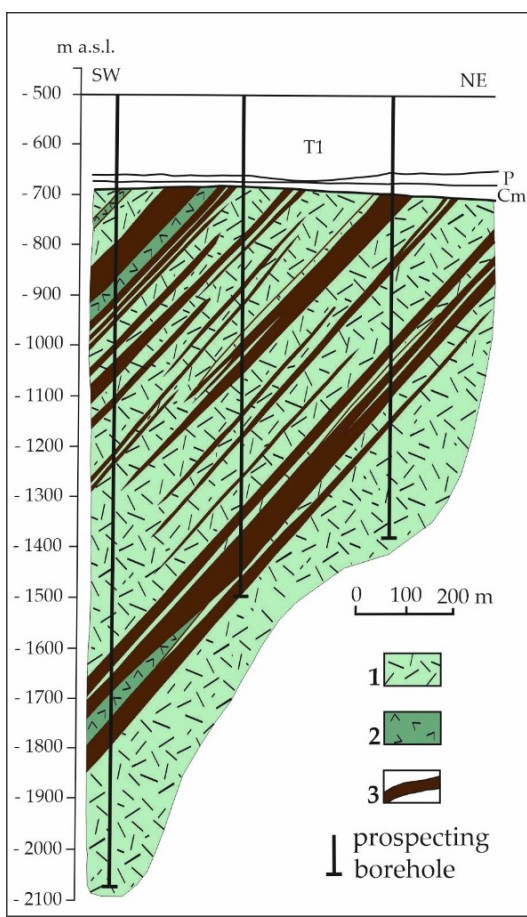

**Figure 2.** Schematic geological cross section (without Cenozoic sedimentary rock cover) of the Krzemianka Fe-Ti-V ore deposit in NE Poland (modified after [39]). Abbreviations: 1—anorthosite; 2—norite; 3—ilmenite-magnetite ore; T1—Lower Triassic; P—Permian; Cm—Cambrian.

### 2.2.2. The Udryn Fe-Ti-V Oxide Deposit and Other Prospects

The Udryn deposit is located in the central part of the Suwałki Anorthosite Massif, approximately 4 km east of the Krzemianka deposit (Figure 1). Twelve boreholes were drilled in this region down to 2300 m [28]. The total area of the Udryn deposit is approximately 0.8 km². Magnetite-ilmenite mineralizations with a content of >15% Fe are considered ore materials. The shapes of the bodies of magnetite-ilmenite mineralization are similar to those found in the Krzemianka deposit, e.g., major lenses and schlieren, dipping generally towards the SE (Figure 3) [28]. The ore bodies in the Udryn deposit are variable in size, with lengths up to 0.8 km and widths up to 0.4 km. Their thicknesses vary from a few centimetres to 100 m. The ore bodies have sharp boundaries, and the boundary reaction zone is defined by the occurrence of volatile-rich minerals such as brown mica, apatite, and cummingtonite [10]. The deposit is also cut by faults and small fissures and the ore bodies have a complicated internal structure, with variations in their texture and composition, but they can be easily traced in the drills because of their large size [28]. This deposit has a similar composition to that of Krzemianka but contains more spinel minerals. In the Udryn deposit, massive ores account for approximately 7–8% of the Fe-Ti-V type mineralization and sub-economic resources are estimated at approximately 263 million tons of Fe-Ti-V ore [39].

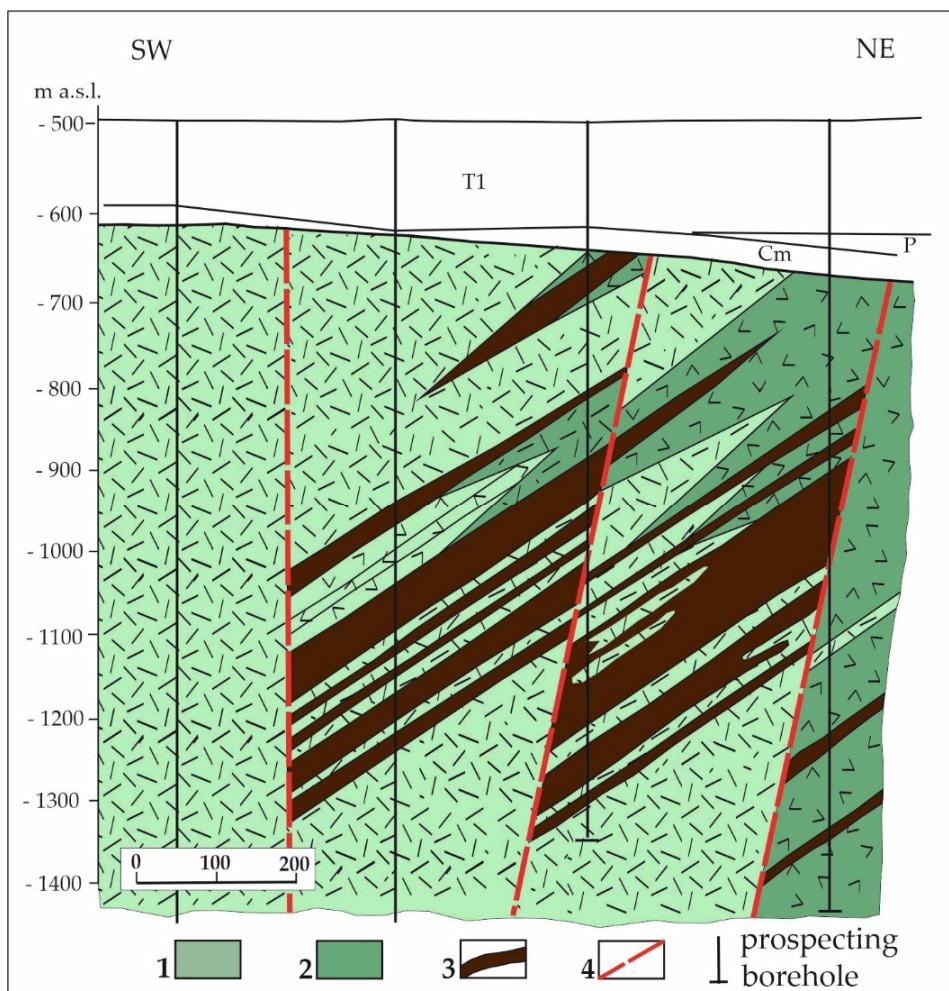

**Figure 3.** Geological cross section (without Cenozoic sedimentary rock cover) of the Udryn Fe-Ti-V ore deposit (modified after [39]). Abbreviations: 1—anorthosite; 2—norite; 3—ilmenite-magnetite ore; 4—fault. T1—Lower Triassic; P—Permian; Cm—Cambrian.

The Jeleniewo prospect was recognized at the SW side of the SAM (Figure 1). This Fe-Ti mineralization was emplaced under leucogabbronorites and anorthosites in the 1115–2300 m depth interval. Mineralizations occur in parallel lenses, running in a NW-SE direction and dipping to the SW. Massive Fe-Ti ores represent about 7–8% in volume of the boreholes' materials and their sub-economic resources are estimated at approximately 116 million tons [28].

On the western margin of the SAM, in the Łopuchowo IG-1 borehole (Figure 1), interesting sulphide-oxide mineralizations containing REE-bearing fluorapatite in nelsonite dikes, were described for the first time in Poland [17].

## 3. Analytical Methods

### 3.1. The Whole-Rock Geochemistry

In total, 39 samples, representing a variety of Fe-Ti-V oxide and sulphide mineralizations, were the subject of the whole-rock geochemistry and point analyses. Samples were collected from the archive core materials of boreholes drilled during the 1970s and 1980s. Twenty-three samples were collected from the Krzemianka deposit (boreholes: Krzemianka −24, −50, −56, −63) and 16 samples came from the Udryn deposit boreholes (Udryn −7, −10, −11) (Figure 4). The ore-grade economic intervals of the drill cores were analyzed for their metallic element contents using a portable pXRF spectrometer (Olympus Delta

Premium Model DP-4000-CC) before sampling to select suitable samples for geochemical analysis (Figure 5) [33].

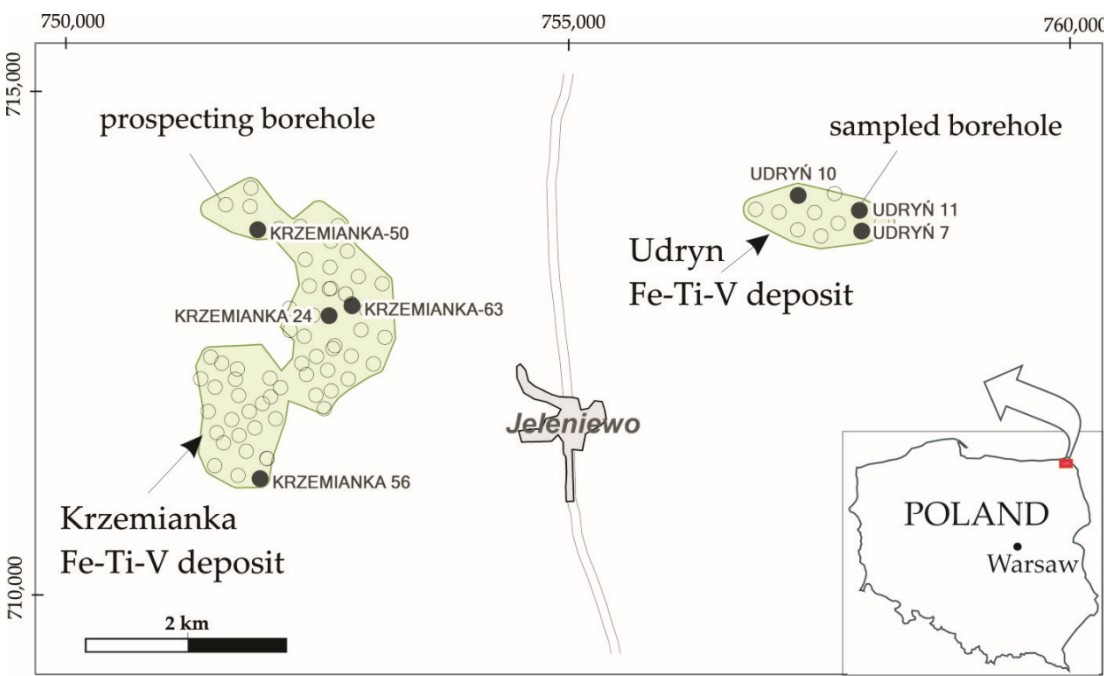

**Figure 4.** Location of boreholes sampled and studied in the Krzemianka and Udryn Fe-Ti-V deposits in NE Poland. Note: the boundaries of the deposits are projected onto the ground surface.

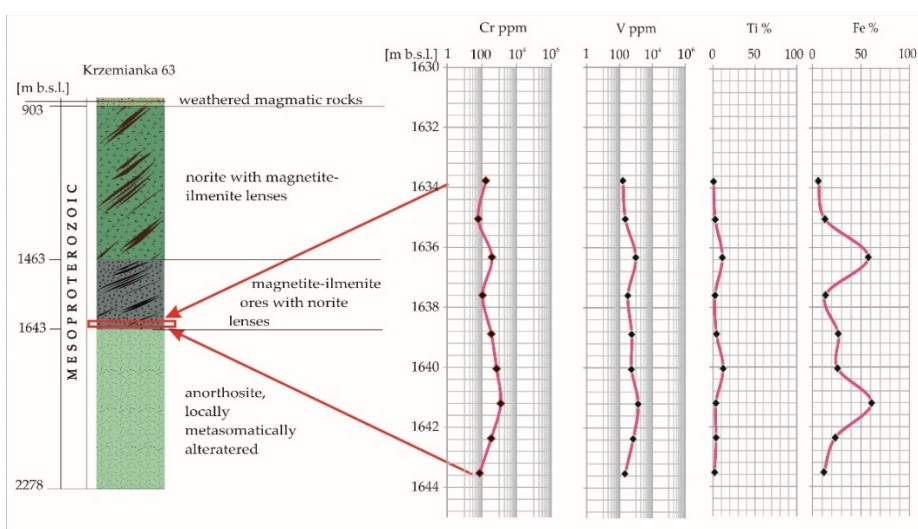

**Figure 5.** Schematic geological profile of the Krzemianka 63 drill hole, together with the results of analyses carried out with a portable XRF spectrometer to assess the content of Cr, V, Ti, and Fe (as a percentage) in the selected ore interval.

The geochemical analyses were performed at the Polish Geological Institute—National Research Institute, using international and internal standards, and included duplicate analyses. The contents of trace elements were determined in pressed powder samples (As, Ba, Bi, Br, Cd, Ce, Co, Cr, Cu, Ga, Hf, La, Mo, Nb, Ni, Pb, Rb, Sn, Sr, Th, U, V, Y, Zn, and Zr) and the major element oxides were determined in fused samples ($SiO_2$, $TiO_2$, $Al_2O_3$, $Fe_2O_3$, MnO, MgO, CaO, $Na_2O$, $K_2O$, $P_2O_5$, $SO_3$, Cl, and F), (Table 1). All analyses were performed using the wavelength dispersive spectrometry (WDS-XRF) X-ray fluorescence method, using a Philips PW-2400 spectrometer. Additionally, the contents of the rare earth

elements (REEs: Sc, Y, La, Ce, Pr, Nd, Eu, Sm, Gd, Tb, Dy, Ho, Er, Tm, Yb, and Lu) and trace elements (Ag, As, Bi, Cd, Co, Cu, Hf, In, Mn, Mo, Ni, Nb, Re, Sb, Se, Sn, Ta, Te, Th, Tl, W, and V) were determined after decomposition with a full mixture of HCl, $HNO_3$, HF, and $HClO_4$ acids using a Perkin Elmer ICP-MS Elan DRC II mass spectrometer by means of inductively-coupled plasma mass spectrometry (ICP-MS). Au, Pd, and Pt were measured using a Perkin Elmer model 4100 ZL spectrometer and the graphite furnace atomic absorption spectrometry (GF AAS) method. In the case of gold analyses, samples were preroasted and reconstituted with a mixture of HCl and $HNO_3$ acids. Liquid–liquid extraction to methyl isobutyl ketone (MIBK) was performed to separate the matrix and concentrate the test solution. The platinum and palladium samples were digested with aqua regia, and then they were isolated on activated carbon using formic acid. The basic statistical parameters (arithmetic mean, geometric mean, median, and standard deviation) of the content of elements in bulk-rock samples from Fe-Ti-V deposits were determined, as well as the correlation matrices between elements in the considering deposits. The degree of correlation of parameters was interpreted as follows. $r \leq 0.5$: no correlation, $r > 0.5$ to 0.7: weak correlation, $r > 0.7$ to 0.9: strong correlation, and $r > 0.9$: very strong correlation.

**Table 1.** The minimum detection limits of element measurements via the applied methods.

| WDS-XRF [ppm] | | | | | | | | | | | | |
|---|---|---|---|---|---|---|---|---|---|---|---|---|
| As | Ba | Bi | Br | Cd | Ce | Co | Cr | Cu | Ga | Hf | La | Mo |
| 3 | 10 | 3 | 1 | 3 | 5 | 3 | 5 | 5 | 3 | 3 | 5 | 2 |
| Nb | Ni | Pb | Rb | Sn | Sr | Th | U | V | Y | Zn | Zr | |
| 2 | 3 | 3 | 3 | 2 | 2 | 3 | 2 | 5 | 3 | 2 | 3 | |
| ICP-MS [ppm] | | | | | | | | | GF AAS [ppm] | | | |
| Sc | Y | La | Ce | Pr | Nd | Eu | Sm | | Au | Pd | Pt | |
| 0.5 | 0.5 | 0.5 | 0.5 | 0.5 | 0.5 | 0.05 | 0.05 | | 0.001 | 0.005 | 0.01 | |
| Gd | Tb | Dy | Ho | Er | Tm | Yb | Lu | | | | | |
| 0.05 | 0.05 | 0.05 | 0.05 | 0.05 | 0.05 | 0.05 | 0.05 | | | | | |
| ICP-MS [ppm] | | | | | | | | | | | | |
| Ag | Cd | In | Mn | Re | Sb | Se | Sn | Ta | V | Te | Tl | W |
| 0.1 | 0.5 | 0.05 | 1 | 0.05 | 0.5 | 2 | 1 | 0.05 | 5 | 0.5 | 0.05 | 0.1 |
| As | Bi | Co | Cu | Hf | Ni | Nb | Mo | Th | | | | |
| 2 | 0.05 | 0.5 | 0.5 | 0.05 | 0.5 | 0.5 | 0.5 | 0.05 | | | | |
| WDS-XRF [%] | | | | | | | | | | | | |
| $SiO_2$ | $TiO_2$ | $Al_2O_3$ | $Fe_2O_3$ | MnO | MgO | CaO | $Na_2O$ | $K_2O$ | $P_2O_5$ | $SO_3$ | Cl | F |
| 0.1 | 0.01 | 0.05 | 0.01 | 0.001 | 0.01 | 0.01 | 0.01 | 0.01 | 0.001 | 0.01 | 0.001 | 0.01 |

Datasets containing values below the limit of detection (LD) can lead to underestimates or overestimates of both the mean value and standard deviation, and therefore to correct this influence statistical substitution methods are commonly used to replace a value below the minimum limit of detection with a value equal to half the limit of detection (LD/2). In our bulk-rock geochemistry statistics, we considered the sample population of a specific element only in cases when the numbers of samples with values above the low detection limit constituted at least 75%, and other values were treated as LD/2.

### 3.2. The Ore Microscope and Microprobe Studies

Mineralogical and petrographic examinations, together with photo-micrographic documentation, were carried out on a NIKON ECLIPSE LV100 POL microscope (Tokyo, Japan) with NIS-Elements software (Version 3.0). The quantitative examination of ore minerals on the electron microprobe was preceded by a preliminary investigation using a ZEISS LEO-1430 scanning electron microscope with an EDS (energy-dispersive spectrometry) detector. Electron microprobe analysis (EPMA) was performed using a Cameca SX-100

microprobe (Paris, France) equipped with five WDS detectors. The following parameters were used during the EPMA analyses: HV accelerating voltage: 15 kV, beam current: 20 nA, a focused beam (<1 μm in diameter), acquisition time at the peak position: 20 s, at the background position: 10 s, and carbon sputtering. International (commercial) standards from the SPI-53 set from SPI and from the sulf-16 set from P&H were used for instrument calibration.

The determination of low concentrations in small phases with an electron microprobe is not straightforward and uncertainties differ according to the acceleration voltage, counting statistics, background subtraction, sample heterogeneity, beam drift, and other analytical factors (e.g., stray X-rays and surface contamination). Although the precision of each single measurement for a given element was different due to the specificity of the method, it was possible to determine the variability of measurement precision. In the case of trace elements, such as V in magnetite and ilmenite, the variations in measurement precisions ranged from 450 to 650 ppm (with an average of approximately 550 ppm) and from 200 to 300 ppm (with an average of approximately 250 ppm), respectively. In the case of cobalt and nickel, the variations in measurement precision ranged from 800 to 1000 ppm (with an average of approximately 900 ppm) in pentlandite, pyrrhotite, chalcopyrite, and pyrite. The statistics of the elements presented in this article include values above or close to the level of precision of a single measurement.

## 4. Results

### 4.1. Metallic Mineralization in Light of the Microscopic and EPMA Studies

All samples were collected from Fe-Ti-rich mineralizations from archival drill cores in the area of the Krzemianka and Udryn Fe-Ti-V deposits, hosted by the SAM. They were dark gray or black, massive or semi-massive rocks, with a characteristic metallic luster and a different ratio of ore mineralization to host rock minerals (Figure 6). They were mainly anorthosites and norites, and less frequently pyroxenites or gabbros, and mineralized nelsonite dikes were also found. Opaque minerals constituted 45–90% of the rock volume and they were massive, xenomorphic, and in most cases they formed a sideronite texture. Transparent minerals were plagioclases and various amounts of ortho- and clinopyroxenes, and small amounts of phlogopite were observed in most of the samples. Plagioclase was characterized by a striped structure, whereas irregular, wormy myrmekites occurred in places on the edges of plagioclase grains. As a result of the equilibrium between orthopyroxene and plagioclase, the grains of plagioclase tended to be rounded. The pyroxenes were partially biotitized and chloritized. Pleochroic fields could be observed in brown micas. These tiny mica laminas often surrounded the grains of pyroxenes; they were the product of the reaction between magnetite and plagioclase and pyroxenes. The analysis of brown micas flakes from the SAM leuco-and gabbro-norite cumulates showed a titanium phlogopite composition. The degree of Ti enrichment in the brown micas ranged from 2.59 to 9.41 wt% $TiO_2$ [40].

The main ore minerals were magnetite and ilmenite, which constituted more than 90% of the massive ores, and were commonly accompanied by sulphides, mainly pyrrhotite, pentlandite, and chalcopyrite (Figure 7A–H). These sulphides may constitute up to 3–5% of the ore's volume. In addition, Al-spinels (with diverse molecular contents of Al, Fe, Mg, and Cr from the hercynite and chromite terms, Table 2) and minor sulphides such as pyrite, siegenite, millerite, bravoite, cubanite, galena, sphalerite, and rare inclusions of greenockite, talnakhite, hessite, or native bismuth were identified.

Magnetite and ilmenite form hipautomorphic crystals up to several mm in diameter. Magnetite contains several generations of exsolutions of ulvöspinel, ilmenite, and Al-spinels (Figure 7A–F). The chemical composition of magnetite includes, on average, 70.3 wt% Fe (range 66.2–72.2 wt%), 21.3 wt% O (range 20.5–22.5 wt%) and substitutions of Al (mean 0.38 wt%), Cr (0.26 wt%), Mg (0.08 wt%), Ti (0.59 wt%), and V (0.42 wt%) (Tables 2 and A1, Figures 8 and 9A).

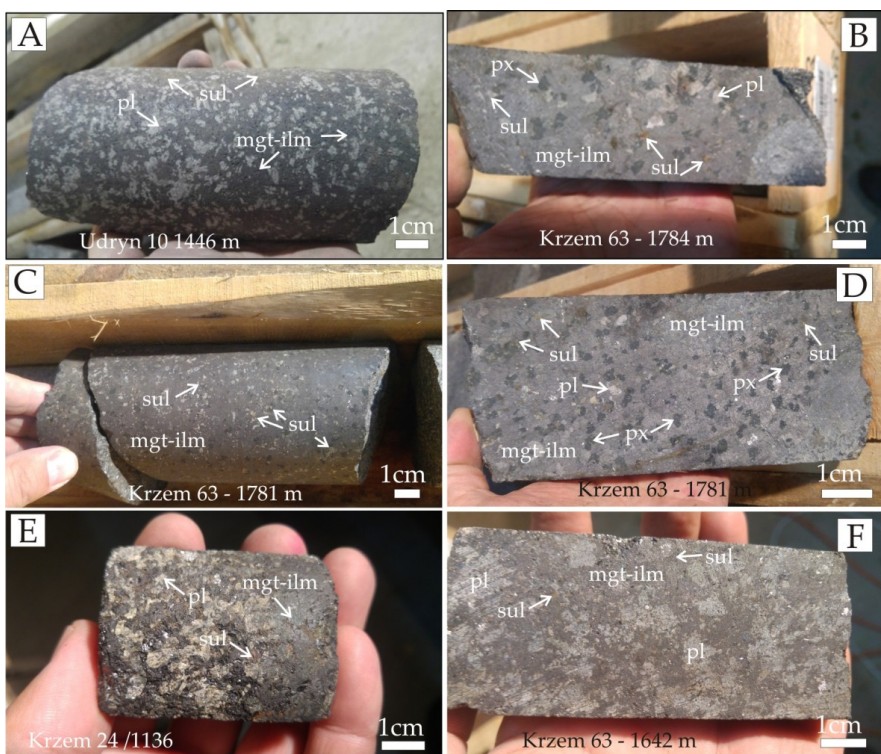

**Figure 6.** Typical massive and semi-massive magnetite-ilmenite (mgt-ilm) ores hosted by anorthosite or norite from the Udryn (**A**) and Krzemianka (**B**–**F**) deposits. Note: Krzem 63–Krzemianka 63 borehole, 1784 m—an approximate depth of sampling. Abbreviations: mgt-ilm—magnetite-ilmenite; pl—plagioclase; px—pyroxene; sul—sulphides.

**Table 2.** The basic statistical parameters of the chemical composition of magnetite, ilmenite, and Al-spinels from the Krzemianka and Udryn Fe-Ti-V deposits in Poland.

| Mineral | Value | Fe | Al | Mn | V | Cr | Mg | Ti | Zn | O |
|---|---|---|---|---|---|---|---|---|---|---|
| | | wt% | wt% | wt% | wt% | wt% | wt% | wt% | wt% | wt% |
| magnetite $Fe^{2+}Fe^{3+}_2O_4$ | arithmetic mean | 70.34 | 0.38 | 0.03 | 0.42 | 0.26 | 0.08 | 0.59 | 0.03 | 21.29 |
| | standard deviation | 1.25 | 0.33 | 0.03 | 0.14 | 0.25 | 0.09 | 0.68 | 0.02 | 0.43 |
| | geometric mean | 70.33 | 0.30 | 0.02 | 0.37 | 0.14 | 0.06 | 0.29 | 0.02 | 21.29 |
| | median | 70.59 | 0.29 | 0.02 | 0.48 | 0.14 | 0.06 | 0.26 | 0.02 | 21.26 |
| | minimum content | 66.20 | 0.01 | b.d.l. | 0.01 | b.d.l. | 0.01 | 0.02 | b.d.l. | 20.46 |
| | maximum content | 72.20 | 2.06 | 0.13 | 0.59 | 1.26 | 0.53 | 2.65 | 0.08 | 22.54 |
| | number of analyses a.d.l. | 81 | 81 | 69 | 81 | 80 | 80 | 81 | 45 | 81 |
| ilmenite $FeTiO_3$ | arithmetic mean | 35.44 | 0.25 | 0.81 | 0.14 | 0.04 | 0.87 | 30.59 | 0.04 | 31.72 |
| | standard deviation | 1.47 | 0.77 | 0.14 | 0.04 | 0.08 | 0.37 | 1.22 | 0.03 | 0.45 |
| | geometric mean | 35.41 | 0.04 | 0.80 | 0.13 | 0.03 | 0.79 | 30.56 | 0.02 | 31.72 |
| | median | 35.19 | 0.03 | 0.80 | 0.14 | 0.03 | 0.80 | 31.10 | 0.03 | 31.78 |
| | minimum content | 33.15 | b.d.l. | 0.59 | 0.05 | b.d.l. | 0.16 | 26.68 | b.d.l. | 30.17 |
| | maximum content | 39.91 | 4.39 | 1.18 | 0.24 | 0.59 | 1.66 | 31.89 | 0.12 | 32.89 |
| | number of analyses a.d.l. | 59 | 57 | 59 | 59 | 47 | 59 | 59 | 31 | 59 |
| Al-spinels $*M^{2+}M^{3+}_2O_4$ | arithmetic mean | 19.24 | 32.08 | 0.13 | 0.04 | 0.36 | 7.05 | 0.17 | 1.26 | 39.35 |
| | standard deviation | 2.99 | 1.06 | 0.03 | 0.02 | 0.37 | 1.43 | 0.17 | 0.64 | 0.90 |
| | geometric mean | 19.01 | 32.07 | 0.13 | 0.04 | 0.25 | 6.91 | 0.09 | 1.09 | 39.34 |
| | median | 18.35 | 32.13 | 0.13 | 0.05 | 0.17 | 7.37 | 0.09 | 1.03 | 39.35 |
| | minimum content | 14.82 | 29.76 | 0.08 | b.d.l. | 0.09 | 4.85 | 0.01 | 0.36 | 37.70 |
| | maximum content | 24.02 | 34.10 | 0.20 | 0.07 | 1.45 | 9.44 | 0.61 | 2.25 | 41.08 |
| | number of analyses a.d.l. | 30 | 30 | 30 | 30 | 30 | 30 | 30 | 30 | 30 |

a.d.l.—above low detection limit; b.d.l.—below low detection limit; $*M^{2+}$ is Mg, Fe, Zn, and Mn and $M^{3+}$ is Fe, Al, V, and Cr.

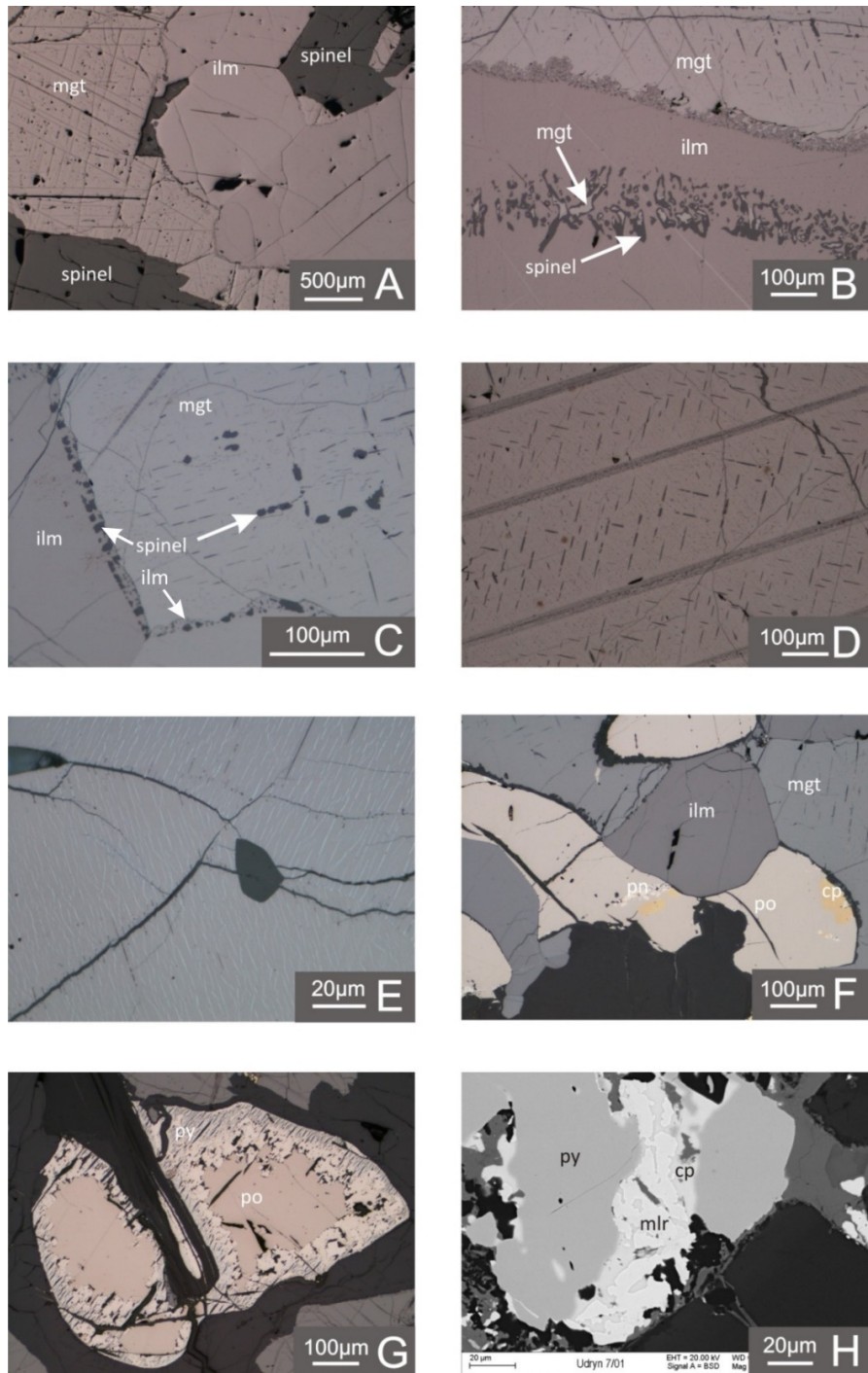

**Figure 7.** Microphotographs of characteristic oxide and sulphide ore mineralization from the Krzemianka and Udryn Fe-Ti-V deposits. (**A**–**G**) are reflected-light photomicrographs with crossed nicoles and (**H**) is a backscattered electron (BSE) image. (**A**) typical ilmenite-magnetite ore in the Krzemianka and Udryn deposits. Magnetite with exsolution of ulvöspinel, in association with ilmenite (ilm) and Al-spinels (spinel); (**B**) front of ilmenitization process on the border of magnetite (mgt) and ilmenite (ilm) is marked with Al-spinel grains. (**C**) exsolutions of ulvöspinel, ilmenite (ilm), and Al-spinels (spinel) in magnetite (mgt); (**D**) exsolutions of ulvöspinel in magnetite, and ilmenite laths; (**E**) ilmenite with hematite exsolutions; (**F**) typical sulphide (pyrrhotite—po, chalcopyrite—cp) aggregates in magnetite-ilmenite ore; (**G**) the pyrrhotite (po) is gradually replaced from the outer edges towards the center by pyrite (py); (**H**) millerite (mlr) replaces chalcopyrite (cp) in the pseudomorphs of pyrite (py) after pyrrhotite.

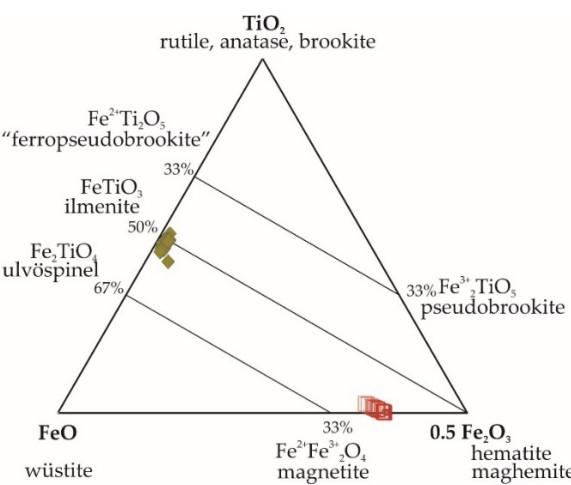

**Figure 8.** The composition of Fe-Ti oxides plotted in a TiO$_2$-FeO-0.5Fe$_2$O$_3$ ternary diagram in ore samples from the Krzemianka and Udryn Fe-Ti-V deposits in NE Poland. Explanation of calculation (in atomic proportions): TiO$_2$–Ti (IV); FeO = Fe (II) + Mg (II) + Mn (II) + 2O (II); Fe$_2$O$_3$ = Fe (III) + Cr (III) + Al (III) + V (III) + 3O (II).

Ilmenite formed wide lamellae in which Al-spinel nodules forming a palisade structure were observed (Figure 7C). Exsolutions of ulvöspinel in magnetite formed two generations of nets, usually occurring at some distance from the ilmenite lamellae (Figure 7D). A characteristic feature of Ti-rich mineralization is the reaction zone between ilmenite and magnetite. It is composed of Al-spinel inserts with magnetite relics that indicate the outline of the primary original magnetite shape (Figure 7B). Fe-Ti oxides have an interstitial habit as a result of subsolidus grain boundary readjustment [23]. The textures of Fe-Ti oxides result from subsolidus re-equilibration [25]. We observed cloth and trellis textures in magnetite. The primary cloth texture of ulvöspinel is oxidized to trellis ilmenite lamellae and granules [23,24]. We also observed Al-spinel fine lenses exsolutions from magnetite and fine granules in ilmenite. The ilmenite grains or aggregates contain exsolution intergrowths with hematite, mostly in dispersed mineralizations (Figure 7E).

Ilmenite's chemical composition showed an average of 35.4 wt% Fe, 30.6 wt% Ti, and 31.7 wt% O (Table 2). Ilmenite contained substitutions of Mg (mean = 0.87 wt%, range 0.16–1.66 wt%), Mn (mean = 0.81 wt%), and V (mean = 0.14 wt%), (Figure 9B). Several measurements showed Al contents of a few percentage points in terms of weight, up to a maximum of 4.4 wt% (Table A2).

Al-spinels commonly occur with magnetite-ilmenite mineralization. The variability of Al-spinels in terms of the content of Al, Fe, Mg, Cr, Mn, and Zn cations is presented in Figure 10A,B. The chemical composition of Al-spinels ranged from 29.8 to 34.1 wt% Al, 14.8–24.0 wt% Fe, 4.9–9.4 wt% Mg, 0.1–1.5 wt% Cr, 0.4–2.3 wt% Zn and of trace substitutions of Mn and V (Tables 2 and A3, Figures 9C and 10A,B).

The most common sulphide was pyrrhotite, coexisting with pentlandite and chalcopyrite (Figures 7F and 11A–H). However, pyrrhotite was predominant among them (ca. 80% of total sulphides), but pentlandite and chalcopyrite contents were similar. These represented the common magmatic sulphides. They formed aggregates that often showed overgrowth of magnetite and ilmenite. Two types of pyrrhotite were recognized [23,25,41]. Our metal-rich ore samples mainly contained the hexagonal form of pyrrhotite, accompanied by chalcopyrite and pentlandite. The pyrrhotite grains were most often xenomorphic in size, ranging from 0.005 to 0.05 mm in diameter. They usually made up polycrystalline aggregates of up to 2–3 mm in diameter. Some pyrrhotite exhibited lamellar structures, resulting from decomposition products of a high-temperature pyrrhotite. EPMA showed that pyrrhotite had a low Fe content (mean = 59.5 wt%) and a high S (mean = 39.1 wt%) content, as well as constant substitution of Ni and Co (Tables 3 and A4). In pyrrhotite, there were oval or lenticular exsolutions—products of pentlandite or chalcopyrite—with diameters

from a few to several hundred micrometres (Figure 11A,B,D,F). They arranged themselves parallel to the planes of separation in pyrrhotite (001). Moreover, star-shaped exsolution products of pentlandite in pyrrhotite were observed. Pentlandite and chalcopyrite were also found in the form of intergranular fillings between the pyrrhotite grains. They exhibited xenomorphic crystals with distinctive cleavage, which were up to 0.5 mm in length. The decomposition of pentlandite in pyrrhotite showed a variable chemical composition, on average 34.9 wt% Ni, 27.2 wt% Fe, and 33.3 wt% S, as well as the constant substitution of Co (Tables 4 and A5). The other secondary sulphides were also observed crystallizing along the fractures or cleavage of primary minerals (e.g., pyrrhotite and pentlandite). They formed as a result of the decomposition of high-temperature pentlandite-pyrrhotite solid solutions into their low-temperature products. They were represented mostly by thiospinels from the (Fe, Ni) (Co, Ni)$_2$S$_4$ terms [41]. EPMA showed that the proportions of metal contents indicated the presence of diadochid substitutions and/or micro-overgrowths of several transition phases in these minerals. Siegenite *((Ni,Co)$_3$S$_4$)*, bravoite *((Fe,Ni,Co)S$_2$)*, mackinawite *((Fe,Ni)S$_{0.9}$)*, and probably smithite *((Fe,Ni)$_9$S$_{11}$)* were identified. The average composition of siegenite was 26.3 wt% Ni, 22.0 wt% Co, 10.8 wt% Fe, and 41.1 wt% S (Tables 4 and A6). Bravoite had much higher Fe and lower Co contents, compared to siegenite (Table 4). Bravoite occurred inside pentlandite along its cleavage, in the form of veinlets or lattices up to approx. 30 μm thick. Moreover, lenticular or lanceolate exsolutions of mackinawite were also observed in pentlandite. Chalcopyrite accounted for about 10% of the sulphide contents in the investigated magnetite-ilmenite ores. It created small overgrowths with pyrrhotite or pentlandite, up to several hundred μm in diameter. The adhesive desorption of these sulphides by gangue minerals was visible. Chalcopyrite was more commonly observed as a thin (30–100 μm thick) lamellae exsolution in pyrrhotite. The chemical composition of chalcopyrite contained, on average, 34.5 wt% Cu, 30.5 wt% Fe, and 34.4 wt% S and the constant substitution of Co, Ni, and Zn (Table A7). Exsolutions of regular cubanite lamellae or sporadically millerite were also observed in the decomposition product of chalcopyrite in the pentlandite (Figure 7H). Cubanite was also found in the form of tiny xenomorphic crystals associated with pyrrhotite. The chemical composition of cubanite included 41.0 wt% Fe, 23.2 wt% Co, and 35.0 wt% S (Tables 3 and A8).

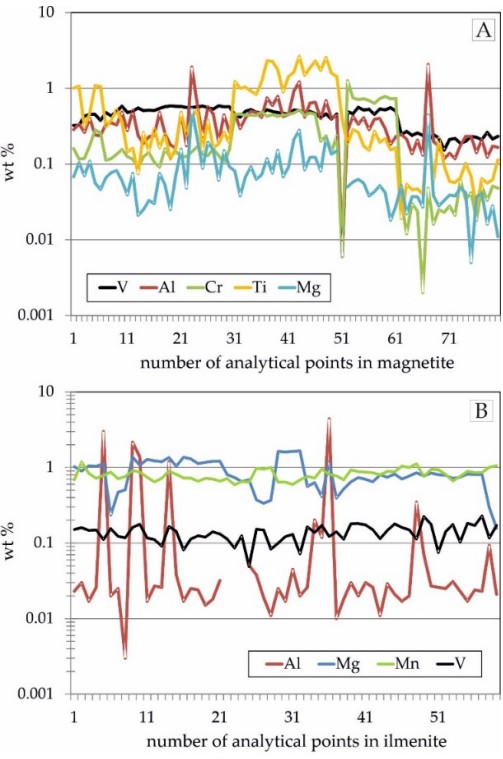

**Figure 9.** *Cont.*

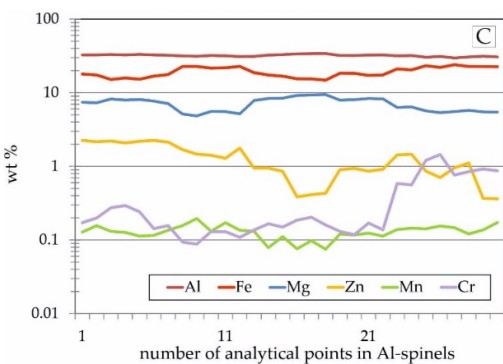

**Figure 9.** Variation in the content of elements in the chemical composition of magnetite (**A**), ilmenite (**B**), and Al-spinels (**C**) from the Krzemianka and Udryn Fe-Ti-V deposits in NE Poland.

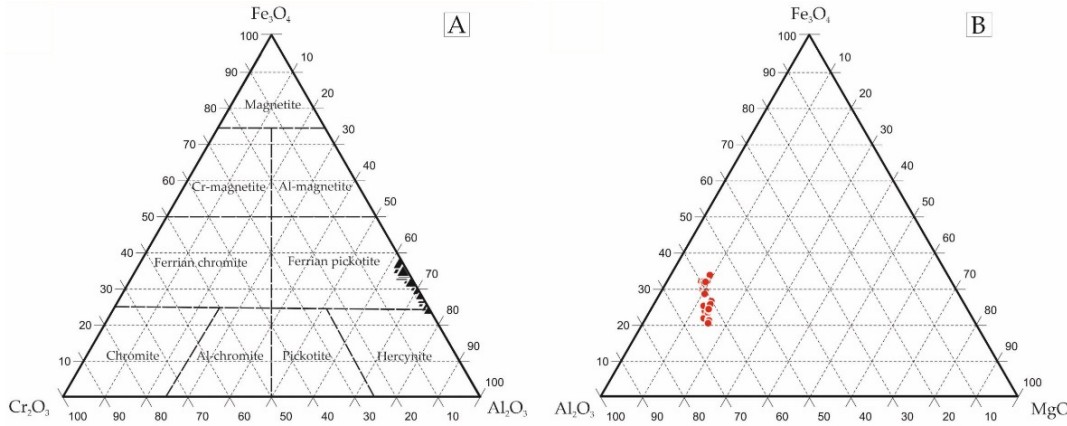

**Figure 10.** Ternary diagrams of the chemical composition of Al-spinels in the $Fe_3O_4$-$Al_2O_3$-$Cr_2O_3$ (**A**) and $Fe_3O_4$-MgO-$Al_2O_3$ (**B**) from the Krzemianka and Udryn Fe-Ti-V deposits in NE Poland.

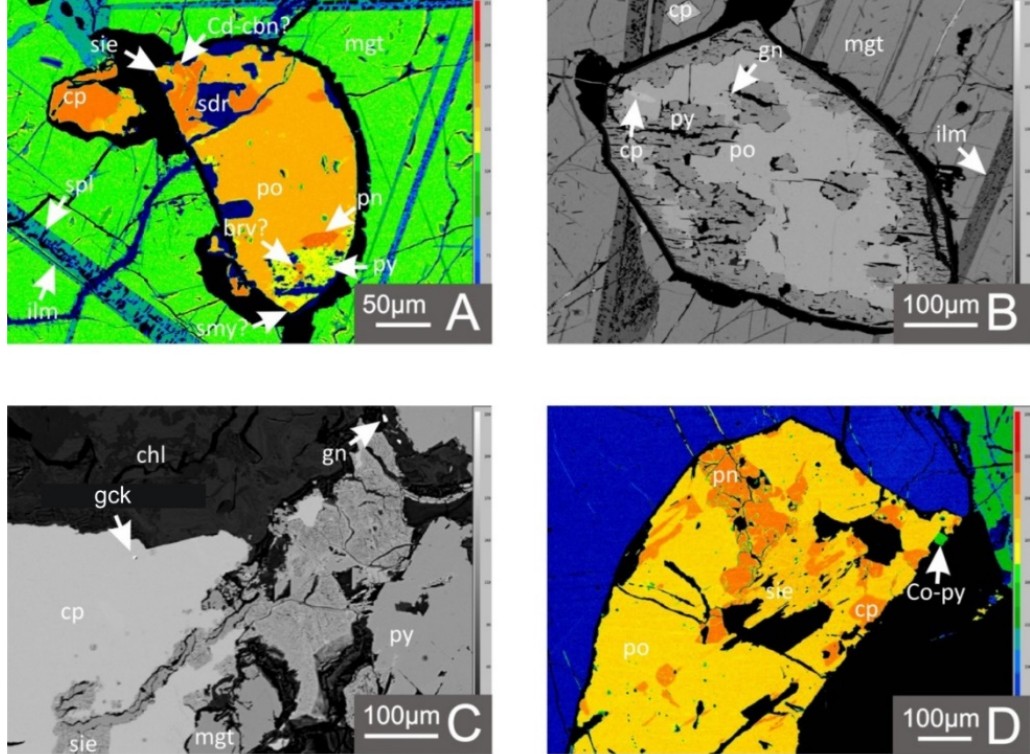

**Figure 11.** *Cont.*

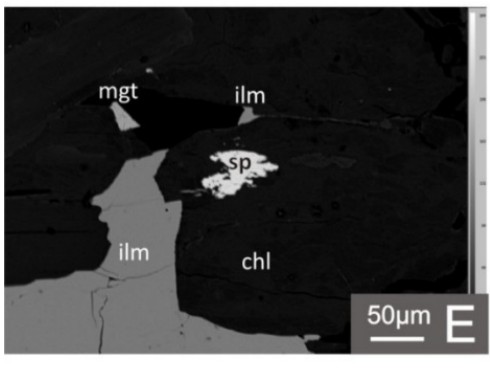
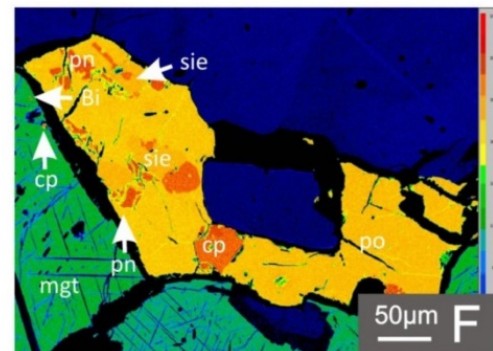

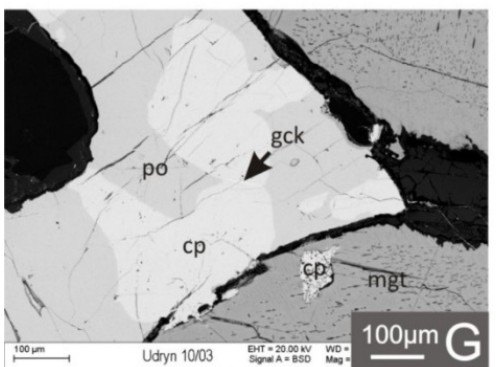
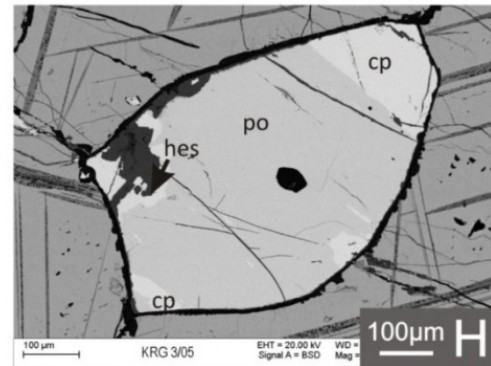

**Figure 11.** BSE images of characteristic sulphide mineralization hosted by magnetite-ilmenite ores from the Krzemianka and Udryn Fe-Ti-V deposits. (**A**) a typical sulphide aggregate in magnetite (mgt)-ilmenite (ilm) ore, in which the primary pyrrhotite (po) and pentlandite (pn) were subject to secondary mineralization processes. They were replaced by sulphide associations with chalcopyrite (cp), siegenite (sie), bravoite (brv), and pyrite with smythite (smy) and by thiospinels (spl). (**B**) pyrrhotite (po) grain subject to pyritization (py) within the magnetite (mgt)-ilmenite (ilm) aggregate; galena (gn) insert is present at the border of pyrite and pyrrhotite, cp—chalcopyrite; (**C**) siegenite (sie) replacement of pyrrhotite (cp) along the cracks. Greenockite and galena (gn) grains associated with low-temperature minerals of the chlorite group (chl); py—pyrite; mgt—magnetite; gck—greenockite; (**D**) sulphide aggregate consisting of pyrrhotite (po), chalcopyrite (cp), siegenite (sie), pentlandite (pn), and Co-bearing pyrite (Co-py), BSEI false colors; (**E**) sphalerite (sp), surrounded by chlorites (chl) in the ilmenite-magnetite ore; (**F**) native bismuth (Bi) inclusion in magnetite (mgt); the aggregate consists of pyrrhotite (po), chalcopyrite (cp), pentlandite (pn), and siegenite (sie), BSEI false colors; (**G**) greenockite (gck) vein in chalcopyrite (cp) and pyrrhotite (po) aggregate hosted by magnetite (mg). Greenockite can be a secondary product of the supergene alteration of Cd-rich sphalerite; (**H**) hessite (hes) inclusion in chalcopyrite (cp) and pyrrhotite (po) aggregate hosted by magnetite-ilmenite ore, BSE.

**Table 3.** The basic statistical parameters of the chemical composition of pyrrhotite, chalcopyrite, cubanite, talnakhite, and pyrite from the Krzemianka and Udryn Fe-Ti-V deposits.

| Mineral | Value | Fe | S | Ni | Co | Cu | Zn |
|---|---|---|---|---|---|---|---|
| | | wt% | wt% | wt% | wt% | wt% | wt% |
| pyrrhotite | arithmetic mean | 59.53 | 39.06 | 0.61 | 0.16 | 0.04 | 0.03 |
| *(FeS)* | standard deviation | 0.56 | 0.34 | 0.54 | 0.05 | 0.03 | 0.04 |
| | geometric mean | 59.53 | 39.06 | 0.48 | 0.15 | 0.51 (*n* = 46) | 0.38 (*n* = 42) |
| | median | 59.56 | 39.03 | 0.55 | 0.15 | 0.04 | 0.03 |
| (*n* = 61) | minimum content | 56.13 | 38.34 | 0.19 | 0.05 | b.d.l. | b.d.l. |
| | maximum content | 60.53 | 39.75 | 4.00 | 0.28 | 0.14 | 0.16 |

**Table 3.** *Cont.*

| Mineral | Value | Fe | S | Ni | Co | Cu | Zn |
|---------|-------|-----|-----|-----|-----|-----|-----|
| | | wt% | wt% | wt% | wt% | wt% | wt% |
| pyrite | arithmetic mean | 44.13 | 53.00 | 1.05 | 1.60 | 0.06 | 0.04 |
| $FeS_2$ | standard deviation | 2.49 | 0.50 | 1.90 | 2.36 | 0.06 | 0.03 |
| | geometric mean | 44.06 | 52.99 | 0.20 | 0.49 | 0.03 | 0.03 |
| | median | 45.44 | 53.07 | 0.11 | 0.25 | 0.04 | 0.03 |
| ($n = 37$) | minimum content | 38.02 | 51.96 | 0.01 | 0.06 | b.d.l. | b.d.l. |
| | maximum content | 47.12 | 53.98 | 8.72 | 8.61 | 0.21 | 0.13 |

| Mineral | Value | Cu | Fe | S | Ni | Co | Zn |
|---------|-------|-----|-----|-----|-----|-----|-----|
| | | wt% | wt% | wt% | wt% | wt% | wt% |
| chalcopyrite | arithmetic mean | 34.52 | 30.50 | 34.36 | 0.11 | 0.11 | 0.08 |
| $CuFeS_2$ | standard deviation | 0.37 | 0.31 | 0.26 | 0.13 | 0.10 | 0.14 |
| | geometric mean | 34.52 | 30.50 | 34.36 | 0.06 | 0.09 | 0.06 ($n = 42$) |
| | median | 34.52 | 30.46 | 34.36 | 0.06 | 0.08 | 0.06 |
| ($n = 50$) | minimum content | 33.65 | 29.88 | 33.83 | 0.01 | 0.02 | b.d.l. |
| | maximum content | 35.37 | 31.35 | 34.87 | 0.56 | 0.55 | 1.01 |
| cubanite | arithmetic mean | 23.19 | 40.97 | 35.01 | 0.05 | 0.08 | 0.04 |
| $CuFe_2S_3$ | standard deviation | 0.40 | 0.41 | 0.21 | 0.04 | 0.035 | 0.04 |
| | geometric mean | 23.18 | 40.97 | 35.01 | 0.03 | 0.07 | 0.02 |
| | median | 23.24 | 40.92 | 34.98 | 0.05 | 0.08 | 0.02 |
| ($n = 10$) | minimum content | 22.24 | 40.34 | 34.72 | 0.01 | 0.04 | 0.01 |
| | maximum content | 23.63 | 41.98 | 35.44 | 0.14 | 0.14 | 0.13 |
| talnakhite | arithmetic mean | 27.09 | 29.05 | 35.00 | 4.20 | 3.94 | 0.30 |
| $Cu_9(Fe,Ni)_8S_3$ | standard deviation | 0.47 | 0.80 | 0.71 | 0.93 | 1.20 | 0.16 |
| | geometric mean | 27.09 | 29.04 | 35.00 | 4.15 | 3.84 | 0.27 |
| | median | 27.09 | 29.05 | 35.00 | 4.20 | 3.94 | 0.30 |
| ($n = 3$) | minimum content | 26.76 | 28.48 | 34.50 | 3.54 | 3.09 | 0.18 |
| | maximum content | 27.42 | 29.61 | 35.50 | 4.86 | 4.78 | 0.41 |

b.d.l.—below detection limit.

**Table 4.** The basic statistical parameters of the chemical composition of pentlandite, siegenite, and bravoite from the Krzemianka and Udryn Fe-Ti-V deposits.

| Mineral | Value | Ni | Fe | Co | Cu | Zn | S |
|---------|-------|-----|-----|-----|-----|-----|-----|
| | | wt% | wt% | wt% | wt% | wt% | wt% |
| pentlandite | arithmetic mean | 34.89 | 27.21 | 4.41 | 0.35 | 0.05 | 33.30 |
| $(Fe,Ni)_9S_8$ | standard deviation | 3.58 | 2.15 | 3.09 | 0.85 | 0.04 | 1.59 |
| | geometric mean | 34.69 | 27.13 | 3.63 | 0.12 ($n = 42$) | 0.04 | 33.27 |
| | median | 36.33 | 27.10 | 3.19 | 0.11 | 0.05 | 32.96 |
| ($n = 45$) | minimum content | 23.70 | 22.26 | 1.12 | b.d.l. | b.d.l. | 32.14 |
| | maximum content | 39.23 | 37.35 | 15.30 | 4.38 | 0.15 | 41.19 |
| siegenite | arithmetic mean | 26.31 | 10.75 | 22.03 | 0.09 | 0.04 ($n = 16$) | 41.14 |
| $(Ni,Co)_3S_4$ | standard deviation | 2.61 | 2.80 | 3.29 | 0.06 | 0.03 | 1.04 |
| | geometric mean | 26.19 | 10.35 | 21.79 | 0.07 | 0.03 | 41.13 |
| | median | 25.50 | 10.84 | 22.14 | 0.08 | 0.04 | 41.54 |
| ($n = 18$) | minimum content | 22.89 | 4.33 | 13.84 | b.d.l. | b.d.l. | 37.71 |
| | maximum content | 33.22 | 17.21 | 29.67 | 0.20 | 0.12 | 42.10 |
| bravoite | arithmetic mean | 28.10 | 28.26 | 2.66 | 0.45 | - | 39.38 |
| $(Fe,Ni,Co)S_2$ | standard deviation | 2.19 | 2.24 | 1.14 | 0.74 | - | 0.48 |
| | geometric mean | 28.04 | 28.20 | 2.45 | - | - | 39.38 |
| | median | 27.94 | 28.29 | 3.22 | 0.05 | - | 39.12 |
| ($n = 7$) | minimum content | 25.99 | 26.01 | 1.35 | b.d.l. | b.d.l. | 39.09 |
| | maximum content | 30.36 | 30.48 | 3.40 | 1.31 | 0.04 | 39.94 |

b.d.l.—below detection limit.

Igneous pyrrhotite is a hexagonal polymorph that changes along the grain edges into a monoclinic polymorph, which is usually highlighted during the oxidation of polished sections. Monoclinic pyrrhotite was most often replaced by secondary pyrite (Figures 7G and 11A,B). Pyrite was formed in late processes as a monoclinic pyrrhotite replacement. The characteristic bird's-eye textures, as a result of oxidative conditions during increasing sulphur fugacity $f S_2$, were observed (Figure 7G). The chemical composition of pyrite was, on average, 44.1 wt% Fe and 53.0 wt% S and a constant substitution of Co and Ni (Tables 3 and A9). Another product of pyrrhotite transformations was smithite, which was probably formed as a product of the low-temperature decomposition of strongly magnetic monoclinic pyrrhotite (Figure 11A) [41]. Sphalerite, up to a few μm in diameter, is present in the form of inclusions within the younger generation of chalcopyrite, and next to chlorite, as a filling of cracks in pyroxenes and Al-spinels. Moreover, irregular, amoebic single grains of sphalerite with diameters up to 50 μm were found in the vicinity of chlorites forming massive clusters (Figure 11E). They had a chemical composition of 64.5–65.2 wt% Zn, 32.4–32.7 wt% S, and 1.9–2.3 wt% Fe. Galena was identified in the form of small grains, not exceeding 2–3 μm in diameter. This occurred in a dispersed form within rock-forming silicates, siegenite, and also in ilmenite admixing lamellae, and within pseudomorphoses of pyrite after pyrrhotite (Figure 11C). Single ingrowths or micro-grains of native bismuth, greenockite (*CdS*), and hessite (*$Ag_2Te$*) were identified for the first time in this area. Native bismuth ingrowths were observed in magnetite and pyrrhotite (Figure 11F). Greenockite was found in the form of ingrowths and veins in chalcopyrite (Figure 11G). The chemical composition of greenockite was 53–60 wt% Cd and 23–26 wt% S, with substitutions of Fe, Zn, and Cu (from 5.5 to 8.3 wt%). Additionally, accessory hessite micro-grains with a diameter of 1–5 μm were identified at the edges of chalcopyrite (Figure 11H).

### 4.2. Trace Element Distribution in Fe-Ti-Oxides and Sulphides Mineralization Determined by EPMA

EPMA analyses revealed that the most abundant trace elements in the sulphide mineralization were cobalt and nickel (Figure 12). These elements occurred mainly as substitutions in pentlandite, pyrrhotite, chalcopyrite, and pyrite, and as secondary minerals—thiospinels (*(Fe,Ni) (Co,Ni)$_2$S$_4$*), which formed as a result of pentlandite decomposition [41]. Cobalt and nickel substitutions in sulphides and thiospinels ranged from 0.1 to 40 wt% (Figure 12). The highest contents of Co were found in siegenite (29.7 wt%), and those of Ni were found in pentlandite (39.2 wt%) (Table 4). The contents of Co and Ni in siegenite averaged 22.0 wt% and 26.3 wt%, respectively. On the other hand, Co substitution in pentlandite was, in the vast majority of analyses, in the range of 2 to 7 wt% (Table A5). The remaining, definitely higher Co impurities in pentlandite (>14 wt%) should be interpreted as the presence of other thiospinels, most often identified as siegenite. The association of cobaltiferous-pentlandite with thiospinels (Fe, Co, Ni) showing lower Co substitution, such as bravoite or talnakhite, was confirmed via EPMA.

Bravoite showed nickel contents in the range of 26 to 30.4 wt% (mean = 28.1 wt%) and Co substitution from 1.4 to 3.4 wt% (mean = 2.7 wt%), (Table 4). In talnakhite, the mean values of Ni and Co substitutions were about 4 wt%. The presence of pentlandite and its decomposition products (thiospinels) was subordinate to that of pyrrhotite (approximately 1:10); therefore, much larger resources of Co and Ni should be associated with the presence of pyrrhotite, despite the fact that it contained much lower concentrations of these metals. The mean values of Co and Ni substitution in pyrrhotite were 0.16 and 0.61 wt%, respectively. The range of nickel substitution (0.2 to 4.0 wt%) in pyrrhotite was much higher than that of cobalt (0.05 to 0.3 wt%). In turn, the substitutions of Co and Ni in chalcopyrite were present at a similar level (mean = 0.11 wt%). The variability of Co and Ni substitution ranged from trace amounts up to 0.56 wt% (Table 3). Cubanite, which was formed as a result of the decomposition of a solid solution of magmatic chalcopyrite, contained traces of Co and Ni (with a maximum of 0.14 wt%). Figure 12 shows the diversity of the Co and Ni substitutions in pyrite. Pyrite was of secondary origin and replaced pyrrhotite. Co and Ni substitutions in pyrite ranged from trace amounts up to about 9 wt%, respectively. The

mean value of the Co and Ni contents was 1.6 and 1.1 wt% (*n* = 37), respectively. It was possible to recognize a Co-rich pyrite with Co substitution >1 wt%.

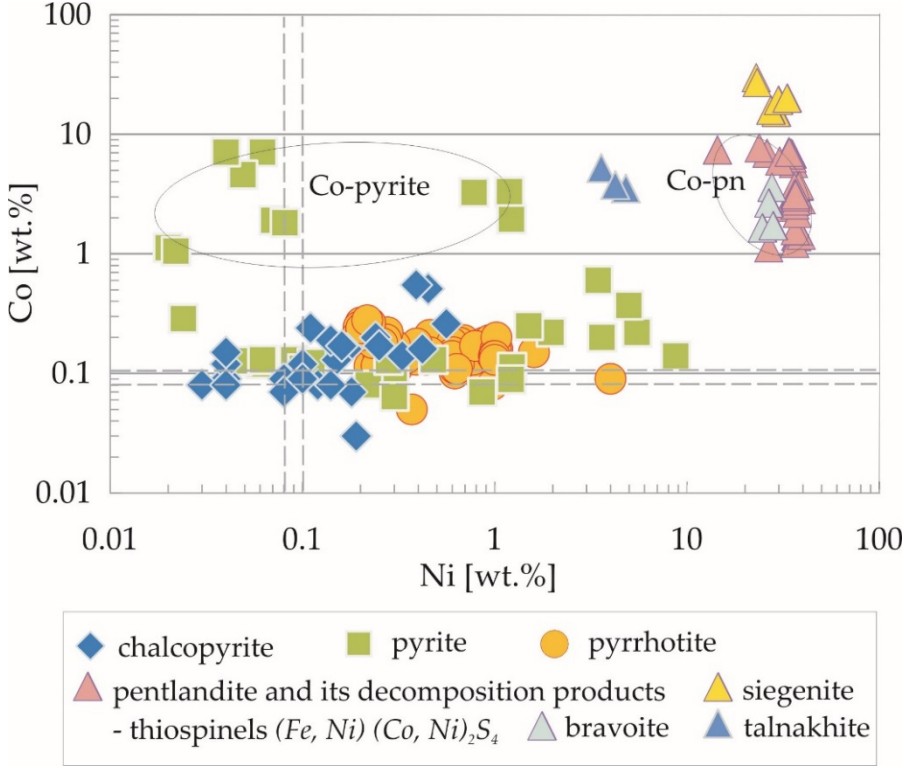

**Figure 12.** Bivariate diagram of Ni vs. Co in sulphides and thiospinels from the Udryn and Krzemianka Fe-Ti-V deposits. Zero values and values below low-precision limits are not presented on the logarithmic axis. Note: gray dashed lines indicate the range of precision of measurements in the entire population for a given element and sulphide.

Cobalt showed a variable correlation with the Ni distribution in sulphides and thiospinels. A strong positive correlation of Co with Ni was observed in chalcopyrite and cubanite (*r* = 0.73, and *r* = 0.62, respectively) but no correlations occurred in other sulphides (Table 5). As an example, there were no correlation between distributions of Co with Fe in pyrrhotite and pentlandite, which indicates the possibility of mutual substitution by both elements (Figure 13A,B).

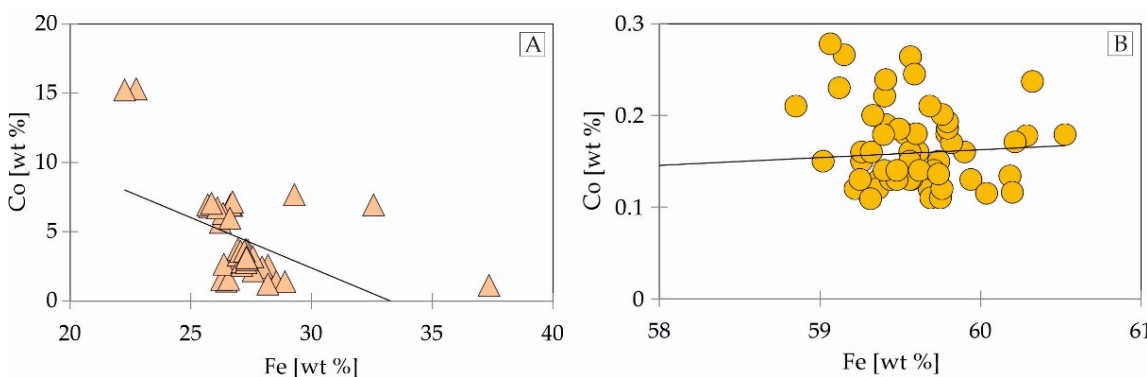

**Figure 13.** Bivariate diagrams of trace elements in sulphides from the magnetite-ilmenite ores in the Udryn and Krzemianka Fe-Ti-V deposits. (**A**) Fe vs. Co in pentlandite; (**B**) Fe vs. Co in pyrrhotite.

**Table 5.** Pearson's correlation coefficients of selected elements for the sulphides from magnetite-ilmenite ores in the Fe-Ti-V deposits in Poland.

| Mineral | Element | S | Cu | Ni | Co | Fe | Mineral |
|---|---|---|---|---|---|---|---|
| pentlandite | S | 1.00 | 0.08 | −0.34 | 0.08 | −0.01 | pyrrhotite |
| | Cu | −0.03 | 1.00 | −0.12 | −0.04 | 0.13 | |
| | Ni | −0.62 | −0.13 | 1.00 | −0.31 | −0.76 | |
| | Co | −0.02 | −0.09 | −0.54 | 1.00 | 0.10 | |
| | Fe | 0.30 | −0.02 | −0.29 | −0.50 | 1.00 | |
| chalcopyrite | S | 1.00 | 0.11 | −0.22 | −0.19 | 0.22 | pyrite |
| | Cu | −0.08 | 1.00 | 0.18 | −0.20 | 0.01 | |
| | Ni | −0.14 | −0.08 | 1.00 | −0.27 | −0.46 | |
| | Co | 0.14 | −0.18 | 0.73 | 1.00 | −0.71 | |
| | Fe | −0.31 | −0.28 | −0.31 | −0.41 | 1.00 | |
| cubanite | S | 1.00 | −0.39 | −0.78 | 0.16 | 0.73 | siegenite |
| | Cu | 0.26 | 1.00 | 0.38 | −0.36 | −0.31 | |
| | Ni | −0.15 | −0.33 | 1.00 | −0.42 | −0.62 | |
| | Co | −0.08 | −0.13 | 0.62 | 1.00 | −0.34 | |
| | Fe | −0.09 | −0.83 | 0.12 | 0.25 | 1.00 | |

Note: Pearson's correlation was based on EPMA in pentlandite ($n$ = 45 measurements), pyrrhotite ($n$ = 61), chalcopyrite ($n$ = 50), pyrite ($n$ = 37), cubanite ($n$ = 10), and siegenite ($n$ = 16). Coefficients with absolute values greater than 0.28 are statistically significant at the 95% level.

Magnetite is the major carrier of the critical element vanadium. The vanadium content in magnetite was in the range of 0.01 to 0.59 wt%, mean = 0.42 wt.% (Table 2, $n$ = 81, Figure 14A). Magnetite commonly contained an admixture of Ti (range from 0.02 to 2.7 wt%, mean = 0.60 wt% for $n$ = 81) which was substituted for $Fe^{2+}$ (Figure 14B). Figure 14C,D shows the vanadium to titanium and vanadium to chromium ratios observed in magnetite, revealing a weak correlation with the Ti and Cr distribution.

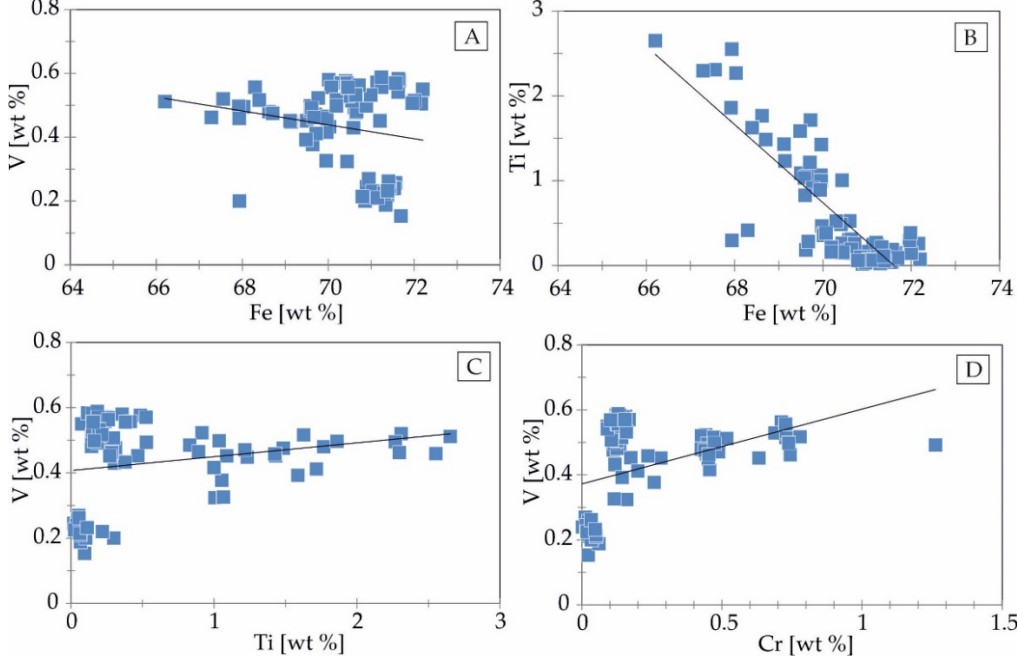

**Figure 14.** Bivariate diagrams of iron vs. vanadium (**A**), iron vs. titanium (**B**), vanadium vs. titanium (**C**), and chromium (**D**) in magnetite from the Udryn and Krzemianka Fe-Ti-V deposits. Trend lines represent correlations between elements. All plots represent the same set of samples; 0 values and samples with values below the level of precision of a single measurement are not presented on the axis.

Ilmenite contained vanadium from 0.05 to 0.24 wt% V (mean = 0.14 wt% V, *n* = 59), (Figure 15A,B). Massive ilmenites had the highest V contents, in the range of 0.18–0.24 wt%m with an increase in iron content ranging from 36 to 40 wt%. There was a slight enrichment of massive ilmenites with vanadium (>0.18 wt%) when compared to ilmenite lamellae exsolutions in magnetite or to ilmenite pseudomorphs after magnetite (Figure 15A).

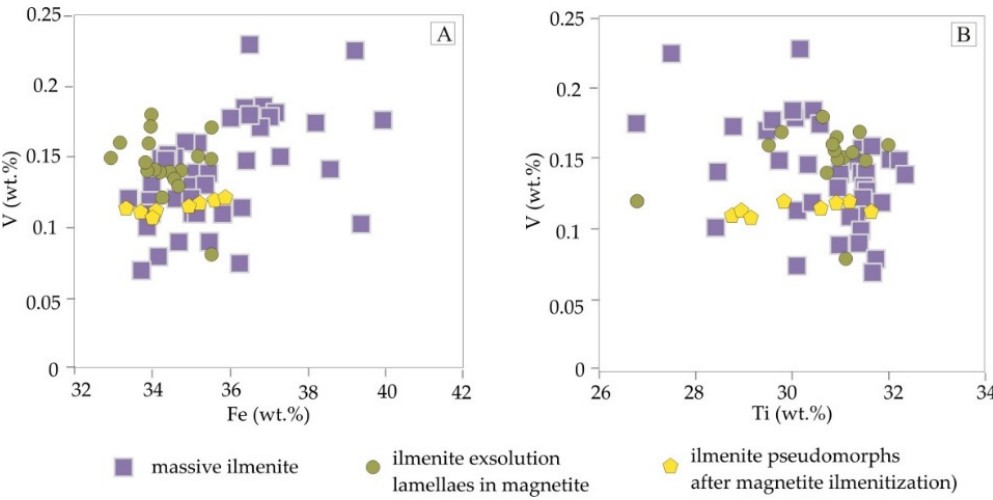

**Figure 15.** Bivariate diagrams of vanadium vs. iron (**A**) and vanadium vs. titanium (**B**) in ilmenite depending on crystal form in the Krzemianka and Udryn Fe-Ti-V deposits. All plots represent the same set of samples; 0 values and samples with values below the level of precision of a single measurement are not presented on the axis.

*4.3. Bulk-Rock Geochemical Investigation of Fe-Ti-V Oxide and Sulphide Ores*

The subject of the research was to determine the concentration of elements in the magnetite-ilmenite ores, along with the locally co-occurring polymetallic sulphides from the Krzemianka and Udryn deposits located in the SAM. Twenty-three ore-bearing samples were collected from the Krzemianka deposit and 16 samples from the Udryn deposit. The rocks hosting magnetite-ilmenite mineralization with sulphides were mainly rich in $Fe_3O_4$ and $TiO_2$ anorthosites, norites, and jotunites, and to a lesser extent pyroxenites and gabbros. In the studied bulk-rock samples, the iron contents ranged from 6.4 to 76.2% $Fe_2O_3t$ (*n* = 39; Tables 6 and 7). Most of the Fe-Ti rich ores showed massive and semi-massive textures and they were classified as "ferrolites" (Figure 6). The "ferrolites" showed concentrations in the range of 50–70% (*n* = 19) and 30–40% $Fe_2O_3t$ (*n* = 10), with samples containing 40–50% $Fe_2O_3t$ constituting approximately 10% of the population (*n* = 5). Only a few samples had a poorer Fe-Ti mineralization and showed impregnation-veinlet textures. The arithmetic mean of the total iron concentration $Fe_2O_3t$ was higher, at 48.6% (*n* = 39; Table 6). $Fe_2O_3t$ showed a very strong correlation with Zn (*r* = 0.94; Table 8) and a strong correlation with $TiO_2$ (*r* = 0.89; Figure 16C), Ga (*r* = 0.86), and Ni (*r* = 0.72) and a weak correlation with Co (*r* = 0.69) and Cd (*r* = 0.64) and no correlation, e.g., with V (*r* = 0.42; Figure 16B). The arithmetic mean content of $TiO_2$ was 7.6% (*n* = 39). $TiO_2$ showed a strong correlation with $Fe_2O_3t$ (*r* = 0.89), Zn (*r* = 0.88), and Ga (*r* = 0.75) and a weaker correlation with Co, Ni, Cd, and U (*r* = 0.6), and no correlation with V (Figure 16D). Vanadium concentrations ranged from 104 to 2208 ppm and in 40% of the population the vanadium concentration was above 1000 ppm (0.1%). The arithmetic mean for vanadium = 960.2 ppm (*n* = 39). Vanadium showed a good correlation with cobalt (*r* = 0.70, Figure 16D) and no correlation with $Fe_2O_3t$ (*r* = 0.42; Figure 16E), Ga, or $TiO_2$.

**Table 6.** Representative whole-rock chemistry of mineralized samples from the Fe-Ti-V deposits in Poland.

| Sample | K24/01 | K50/05 | K56/06 | U11/05 | K63/06 | U10/04 | K24/03 | U7/03 | K56/05 |
|---|---|---|---|---|---|---|---|---|---|
| | Anorthosite | Norite | | Jotunite | | Ferrolite | | | |
| SiO$_2$ (%) | 45.39 | 28.37 | 31.32 | 26.72 | 8.27 | 7.27 | 13.53 | 16.34 | 24.16 |
| TiO$_2$ | 1.573 | 6.037 | 5.153 | 6.269 | 7.221 | 10.78 | 9.562 | 9.372 | 6.838 |
| Al$_2$O$_3$ | 27.28 | 13.88 | 12.2 | 7.53 | 5.74 | 7.87 | 8.94 | 8.95 | 11.3 |
| Fe$_2$O$_3$ | 6.4 | 39.34 | 37.1 | 36.64 | 76.23 | 69.78 | 62.47 | 56.38 | 47.38 |
| MnO | 0.061 | 0.217 | 0.296 | 0.52 | 0.394 | 0.269 | 0.276 | 0.271 | 0.278 |
| MgO | 1.25 | 4.43 | 5.1 | 8.03 | 1.25 | 2.33 | 2.48 | 4.07 | 3.79 |
| CaO | 11.98 | 5.02 | 5.11 | 9.77 | 1.29 | 1.57 | 2.55 | 2.77 | 4.47 |
| Na$_2$O | 4.58 | 2.28 | 1.84 | 1.47 | 0.62 | 0.69 | 0.97 | 0.89 | 1.51 |
| K$_2$O | 0.72 | 0.4 | 0.56 | 0.27 | 0.97 | 0.19 | 0.19 | 0.22 | 0.55 |
| P$_2$O$_5$ | 0.332 | 0.054 | 0.177 | 2.388 | 0.027 | 0.028 | 0.009 | 0.022 | 0.364 |
| SO$_3$ | 0.06 | 0.44 | 0.03 | 1.49 | 0.2 | <0.01 | <0.01 | 0.03 | <0.01 |
| Cl | 0.082 | 0.144 | 0.085 | 0.089 | 0.127 | 0.129 | 0.091 | 0.115 | 0.087 |
| LOI | 0.3 | 0.5 | 1 | 0.9 | 2 | 1.3 | 1.1 | 0.2 | 0.9 |
| SUM | 99.71 | 99.17 | 99.21 | 99.29 | 98.91 | 98.14 | 99.02 | 98.08 | 99.17 |
| Ag ppm | 0.6 | 0.5 | 0.9 | 0.7 | 0.3 | 0.6 | 0.6 | <0.3 | 1.2 |
| Au ppb | <1 | 6 | 7 | 2 | 1 | 5 | 4 | 1 | 7 |
| Ba | 272 | 171 | 259 | 106 | <10 | <10 | <10 | <10 | 706 |
| Cd | 3 | 14 | 9 | 15 | 17 | 17 | 11 | 14 | 8 |
| Ce | 26.8 | 12.6 | 38.1 | 285,0 | 6.5 | 6.3 | 2.9 | 4.8 | 35.4 |
| Co | 15 | 138 | 205 | 65 | 128 | 144 | 135 | 143 | 161 |
| Cr | 2.5 | 362 | 34 | <5 | 133 | 2159 | 218 | 3777 | 217 |
| Cu | 38 | 325 | 604 | 117 | 247 | 94 | 231 | 61 | 658 |
| Dy | 1.89 | 0.64 | 2.79 | 23.56 | 0.3 | 0.38 | 0.15 | 0.29 | 2.19 |
| Er | 0.86 | 0.37 | 1.46 | 11.44 | 0.17 | 0.25 | 0.11 | 0.19 | 1.11 |
| Eu | 2.9 | 0.91 | 3.81 | 36.85 | 0.59 | 0.52 | 0.19 | 0.31 | 3.32 |
| Ga | 23 | 38 | 31 | 23 | 48 | 70 | 50 | 54 | 43 |
| Gd | 2.63 | 0.76 | 3.4 | 32.5 | 0.47 | 0.46 | 0.17 | 0.28 | 3,00 |
| Ge | 0.05 | 0.2 | 0.3 | 0.7 | 0.4 | 0.4 | 0.4 | 0.3 | 0.3 |
| Hf | <3 | <3 | 6 | 14 | 5 | 12 | 5 | 8 | <3 |
| Ho | 0.35 | 0.13 | 0.55 | 4.52 | 0.05 | 0.08 | <0.05 | 0.06 | 0.43 |
| In | 0.025 | 0.08 | 0.11 | 0.3 | 0.23 | 0.13 | 0.16 | 0.11 | 0.15 |
| La | 12.7 | 6.8 | 19.1 | 114.4 | 3.2 | 3,0 | 1.7 | 2.7 | 18.3 |
| Lu | 0.08 | <0.05 | 0.18 | 1.1 | <0.05 | <0.05 | <0.05 | <0.05 | 0.13 |
| Nb | 5 | 6 | 9 | 31 | 14 | 7 | 7 | 9 | 8 |
| Nd | 14.9 | 5.5 | 19.1 | 182.7 | 2.8 | 2.9 | 1.2 | 2,0 | 17.9 |
| Ni | 16 | 398 | 570 | 56 | 401 | 664 | 326 | 646 | 426 |
| Pb | 13 | 4 | 8 | 6 | <3 | <3 | <3 | <3 | 8 |
| Pd ppb | <5 | <5 | <5 | <5 | 12 | 5 | 5 | <5 | <5 |
| Pr | 3.6 | 1.5 | 4.8 | 41.3 | 0.8 | 0.8 | 0.25 | 0.6 | 4.5 |
| Pt ppb | <10 | <10 | <10 | <10 | <10 | <10 | 12 | 12 | <10 |
| Rb | 6 | 9 | 20 | 11 | 181 | 9 | 7 | 7 | 20 |
| Sc | 11.7 | 16.9 | 24.2 | 74.9 | 14.6 | 17.8 | 23.8 | 21.9 | 21.3 |
| Sm | 1.62 | 0.62 | 1.29 | 5.08 | 0.14 | 0.22 | 0.29 | 0.38 | 1.39 |
| Sn | 2 | 2 | 3 | 5 | 10 | 3 | 4 | 3 | 3 |
| Sr | 678 | 383 | 339 | 234 | 86 | 129 | 204 | 198 | 285 |
| Ta | 0.16 | 0.12 | 0.39 | 1.77 | 0.6 | 0.2 | 0.18 | 0.18 | 0.25 |
| Tb | 0.34 | 0.1 | 0.48 | 4.35 | 0.06 | 0.06 | <0.05 | <0.05 | 0.38 |
| Th | 0.56 | 0.77 | 1.36 | 2.59 | 1.31 | 0.27 | 0.07 | 0.18 | 1.05 |
| Tm | 0.1 | 0.05 | 0.2 | 1.41 | <0.05 | <0.05 | <0.05 | <0.05 | 0.15 |
| V | 104 | 1242 | 860 | 456 | 767 | 1099 | 915 | 980 | 2009 |
| W | 0.3 | 0.3 | 0.5 | 0.3 | 2.2 | 0.1 | 0.5 | 0.3 | 1.3 |
| Y | 8.3 | 3.2 | 13,00 | 105.6 | 1.4 | 1.8 | 0.8 | 1.5 | 10.3 |
| Yb | 0.57 | 0.33 | 1.21 | 7.79 | 0.24 | 0.23 | 0.14 | 0.27 | 0.89 |
| Zn | 62 | 324 | 271 | 481 | 641 | 701 | 630 | 519 | 448 |
| Zr | 86 | 51 | 99 | 595 | 28 | 46 | 54 | 53 | 110 |

**Table 7.** Basic statistical parameters for trace elements (in ppm, and in ppb for Au, Pt, and Pd) and major oxides (as a percentage) of the bulk-rock samples (*n* = 39) from the Fe-Ti-V oxide deposits (the Krzemianka deposit, *n* = 23 samples; the Udryn deposit, *n* = 16 samples) in NE Poland.

| Element/ Compound | Arithmetic Mean | Geometric Mean | Median | Minimum Content | Maximum Content | Standard Deviation |
|---|---|---|---|---|---|---|
| Fe$_2$O$_3$ total (%) | 48.61 | 44.99 | 51.43 | 6.4 | 76.23 | 15.69 |
| TiO$_2$ (%) | 7.56 | 7.11 | 8.05 | 1.57 | 11.44 | 2.26 |
| V (ppm) | 960.23 | 861.84 | 977 | 104 | 2208 | 403.57 |
| Co | 122.33 | 111.75 | 137 | 15 | 205 | 40.44 |
| Ga | 44.33 | 41.98 | 45 | 21 | 84 | 14.46 |
| Ge | 0.29 | 0.27 | 0.3 | 0.05 | 0.7 | 0.12 |
| Hf | 0.95 | 0.87 | 0.93 | 0.38 | 2.09 | 0.4 |
| In | 0.13 | 0.12 | 0.12 | 0.03 | 0.3 | 0.06 |
| Nb | 9.38 | 8.27 | 8 | 5 | 32 | 6.42 |
| Sc | 24.29 | 21.63 | 19.8 | 10.8 | 74.9 | 15 |
| W | 3.59 | 0.46 | 0.3 | 0.1 | 101 | 16.17 |
| Y | 11.38 | 3.98 | 3.8 | 0.25 | 105.6 | 25.4 |
| La | 15.13 | 7.07 | 6.8 | 0.8 | 118.3 | 28.69 |
| Ce | 33.93 | 13.8 | 12.6 | 1.3 | 285 | 71.17 |
| Pr | 4.59 | 1.63 | 1.7 | 0.25 | 41.3 | 10.27 |
| Nd | 19.36 | 6.51 | 6.8 | 0.5 | 182.7 | 45.06 |
| Eu | 3.76 | 1.16 | 1.23 | 0.07 | 36.85 | 8.94 |
| Sm | 0.97 | 0.61 | 0.62 | 0.09 | 5.27 | 1.27 |
| Gd | 3.3 | 1.03 | 0.92 | 0.07 | 32.5 | 7.8 |
| Tb | 0.44 | 0.14 | 0.13 | 0.03 | 4.35 | 1.03 |
| Dy | 2.48 | 0.83 | 0.83 | 0.06 | 23.56 | 5.63 |
| Ho | 0.48 | 0.16 | 0.16 | 0.03 | 4.52 | 1.07 |
| Er | 1.25 | 0.47 | 0.44 | 0.03 | 11.44 | 2.71 |
| Yb | 0.94 | 0.42 | 0.38 | 0.03 | 7.79 | 1.83 |
| Cr | 678.22 | 219.58 | 171 | 2.5 | 5705 | 1164.87 |
| Cu | 328.54 | 226.55 | 231 | 38 | 1756 | 323.94 |
| Ni | 436.36 | 302.06 | 421 | 16 | 1370 | 301.16 |
| Zn (ppm) | 453 | 415.65 | 481 | 62 | 701 | 152.79 |
| Au (ppb) | 6.62 | 3.46 | 4 | <1 | 37 | 8.32 |
| Ag (ppm) | 0.66 | 0.5 | 0.6 | <0.30 | 2.5 | 0.54 |
| Cd | 13.77 | 12.9 | 14 | 3 | 24 | 4.37 |
| Br | 4.64 | 4.55 | 5 | 3 | 7 | 0.93 |
| Rb | 16.97 | 11.55 | 9 | 6 | 181 | 29.56 |
| Sr | 271.49 | 247.15 | 254 | 86 | 678 | 123.62 |
| Sn | 3.15 | 2.96 | 3 | 2 | 10 | 1.41 |
| Ta | 0.34 | 0.24 | 0.23 | 0.08 | 1.77 | 0.4 |
| Th | 0.75 | 0.52 | 0.5 | 0.07 | 2.83 | 0.68 |
| U | 2.9 | 2.55 | 3 | 1 | 5 | 1.29 |
| Zr | 105 | 72.31 | 66 | 25 | 737 | 142.92 |
| SiO$_2$ (%) | 21.47 | 19.1 | 18.44 | 6.27 | 45.39 | 10.15 |
| Al$_2$O$_3$ | 11.27 | 10.72 | 10.35 | 5.74 | 27.28 | 4.14 |
| MnO | 0.26 | 0.25 | 0.26 | 0.06 | 0.52 | 0.08 |
| MgO | 3.88 | 3.47 | 3.43 | 1.25 | 8.97 | 1.9 |
| CaO | 4.46 | 3.77 | 3.69 | 1.29 | 11.98 | 2.75 |
| Na2O | 1.62 | 1.45 | 1.51 | 0.62 | 4.58 | 0.84 |
| K2O | 0.38 | 0.34 | 0.34 | 0.11 | 1.11 | 0.21 |
| P$_2$O$_5$ | 0.23 | 0.07 | 0.05 | 0.01 | 2.39 | 0.54 |
| (SO$_3$) | 0.33 | 0.09 | 0.21 | 0.01 | 1.7 | 0.47 |
| (Cl) | 0.11 | 0.11 | 0.12 | 0.08 | 0.17 | 0.02 |

**Table 8.** Pearson's correlation coefficients of selected elements for the bulk-rock samples from Fe-Ti-V deposits in Poland.

| | $SiO_2$ | $TiO_2$ | $Al_2O_3$ | $Fe_2O_3$ | MnO | MgO | CaO | $Na_2O$ | $K_2O$ | $P_2O_5$ | $SO_3$ | Ag | Au | Br | Cd | Ce | Co | Cr | Cu | Ga | Ge | Hf | In | La | Nb | Ni | Rb | Sc | Sr | Ta | Th | U | V | Y | Zn | Zr |
|---|---|---|---|---|---|---|---|---|---|---|---|---|---|---|---|---|---|---|---|---|---|---|---|---|---|---|---|---|---|---|---|---|---|---|---|---|
| $SiO_2$ | 1.00 | | | | | | | | | | | | | | | | | | | | | | | | | | | | | | | | | | | |
| $TiO_2$ | 0.90 | 1.00 | | | | | | | | | | | | | | | | | | | | | | | | | | | | | | | | | | |
| $Al_2O_3$ | 0.71 | −0.74 | 1.00 | | | | | | | | | | | | | | | | | | | | | | | | | | | | | | | | | |
| $Fe_2O_3$ | −0.98 | 0.89 | −0.75 | 1.00 | | | | | | | | | | | | | | | | | | | | | | | | | | | | | | | | |
| MnO | −0.14 | 0.16 | −0.67 | 0.17 | 1.00 | | | | | | | | | | | | | | | | | | | | | | | | | | | | | | | |
| MgO | 0.45 | −0.34 | −0.14 | −0.42 | 0.47 | 1.00 | | | | | | | | | | | | | | | | | | | | | | | | | | | | | | |
| CaO | 0.83 | −0.81 | 0.61 | −0.89 | 0.07 | 0.31 | 1.00 | | | | | | | | | | | | | | | | | | | | | | | | | | | | | |
| $Na_2O$ | 0.86 | −0.84 | 0.94 | −0.90 | −0.46 | 0.06 | 0.78 | 1.00 | | | | | | | | | | | | | | | | | | | | | | | | | | | | |
| $K_2O$ | 0.32 | −0.39 | 0.31 | −0.27 | −0.10 | −0.21 | 0.22 | 0.35 | 1.00 | | | | | | | | | | | | | | | | | | | | | | | | | | | |
| $P_2O_5$ | 0.33 | −0.29 | −0.10 | −0.34 | 0.73 | 0.46 | 0.58 | 0.09 | −0.03 | 1.00 | | | | | | | | | | | | | | | | | | | | | | | | | | |
| $SO_3$ | 0.02 | 0.00 | −0.11 | −0.05 | 0.30 | 0.20 | 0.11 | −0.01 | −0.14 | 0.26 | 1.00 | | | | | | | | | | | | | | | | | | | | | | | | | |
| Ag | −0.11 | 0.17 | −0.06 | 0.07 | 0.02 | −0.11 | −0.08 | −0.05 | 0.06 | −0.04 | 0.50 | 1.00 | | | | | | | | | | | | | | | | | | | | | | | | |
| Au | −0.35 | 0.36 | −0.23 | 0.35 | −0.07 | −0.27 | −0.33 | −0.27 | −0.13 | −0.18 | 0.32 | 0.54 | 1.00 | | | | | | | | | | | | | | | | | | | | | | | |
| Br | −0.59 | 0.56 | −0.44 | 0.56 | 0.09 | −0.21 | −0.46 | −0.51 | −0.26 | −0.12 | −0.04 | 0.18 | 0.53 | 1.00 | | | | | | | | | | | | | | | | | | | | | | |
| Cd | −0.65 | 0.61 | −0.59 | 0.64 | 0.28 | −0.05 | −0.54 | −0.64 | −0.36 | −0.08 | 0.20 | −0.11 | 0.19 | 0.50 | 1.00 | | | | | | | | | | | | | | | | | | | | | |
| Ce | 0.35 | −0.30 | −0.10 | −0.36 | 0.72 | 0.46 | 0.58 | 0.10 | −0.01 | 0.99 | 0.22 | −0.05 | −0.19 | −0.13 | −0.08 | 1.00 | | | | | | | | | | | | | | | | | | | | |
| Co | −0.64 | 0.61 | −0.53 | 0.69 | 0.00 | −0.21 | −0.75 | −0.67 | −0.21 | −0.50 | −0.05 | 0.14 | 0.39 | 0.41 | 0.42 | −0.50 | 1.00 | | | | | | | | | | | | | | | | | | | |
| Cr | −0.25 | 0.29 | −0.08 | 0.23 | −0.13 | −0.14 | −0.25 | −0.21 | −0.25 | −0.18 | −0.28 | −0.23 | −0.12 | 0.18 | 0.20 | −0.18 | 0.21 | 1.00 | | | | | | | | | | | | | | | | | | |
| Cu | −0.28 | 0.29 | −0.25 | 0.30 | −0.02 | −0.20 | −0.33 | −0.28 | 0.01 | −0.18 | 0.41 | 0.65 | 0.85 | 0.41 | 0.14 | −0.19 | 0.48 | −0.28 | 1.00 | | | | | | | | | | | | | | | | | |
| Ga | −0.87 | 0.75 | −0.41 | 0.86 | −0.17 | −0.58 | −0.78 | −0.67 | −0.34 | −0.42 | −0.16 | −0.01 | 0.28 | 0.50 | 0.57 | −0.43 | 0.57 | 0.39 | 0.16 | 1.00 | | | | | | | | | | | | | | | | |
| Ge | −0.21 | 0.23 | −0.58 | 0.24 | 0.86 | 0.25 | 0.02 | −0.47 | −0.17 | 0.78 | 0.17 | −0.04 | −0.01 | 0.21 | 0.26 | 0.78 | −0.06 | 0.04 | −0.03 | 0.05 | 1.00 | | | | | | | | | | | | | | | |
| Hf | −0.04 | 0.23 | −0.38 | 0.03 | 0.49 | 0.23 | 0.06 | −0.23 | −0.05 | 0.45 | 0.09 | 0.26 | 0.13 | 0.17 | 0.01 | 0.47 | −0.11 | 0.00 | 0.05 | −0.13 | 0.55 | 1.00 | | | | | | | | | | | | | | |
| In | −0.15 | 0.17 | −0.61 | 0.19 | 0.92 | 0.29 | 0.06 | −0.42 | −0.02 | 0.77 | 0.29 | 0.10 | −0.02 | 0.08 | 0.21 | 0.77 | −0.14 | −0.14 | 0.02 | −0.10 | 0.92 | 0.56 | 1.00 | | | | | | | | | | | | | |
| La | 0.35 | −0.29 | −0.03 | −0.35 | 0.58 | 0.41 | 0.50 | 0.15 | 0.01 | 0.83 | 0.18 | −0.11 | −0.18 | −0.15 | −0.06 | 0.84 | −0.38 | −0.12 | −0.19 | −0.40 | 0.63 | 0.36 | 0.59 | 1.00 | | | | | | | | | | | | |
| Nb | 0.23 | −0.17 | −0.25 | −0.22 | 0.82 | 0.41 | 0.45 | −0.03 | 0.05 | 0.94 | 0.27 | 0.01 | −0.14 | 0.01 | 0.01 | 0.96 | −0.10 | −0.15 | −0.14 | −0.37 | 0.84 | 0.54 | 0.87 | 0.79 | 1.00 | | | | | | | | | | | |
| Ni | −0.73 | 0.61 | −0.39 | 0.72 | −0.15 | −0.39 | −0.71 | −0.62 | −0.32 | −0.40 | 0.04 | 0.24 | 0.62 | 0.64 | 0.51 | −0.41 | 0.68 | 0.32 | 0.55 | 0.82 | 0.01 | −0.09 | −0.13 | −0.35 | −0.35 | 1.00 | | | | | | | | | | |
| Rb | −0.19 | 0.01 | −0.24 | 0.28 | 0.24 | −0.24 | −0.22 | −0.21 | 0.70 | −0.07 | −0.10 | −0.04 | −0.08 | 0.06 | 0.07 | −0.06 | 0.05 | −0.13 | 0.02 | 0.02 | 0.13 | −0.05 | 0.30 | −0.08 | 0.12 | −0.03 | 1.00 | | | | | | | | | |
| Sc | 0.37 | −0.23 | −0.25 | −0.36 | 0.72 | 0.74 | 0.47 | 0.00 | −0.12 | 0.86 | 0.21 | −0.06 | −0.25 | −0.13 | −0.07 | 0.86 | −0.43 | −0.19 | −0.10 | −0.40 | 0.83 | 0.57 | 0.73 | 0.69 | 0.84 | −0.48 | −0.10 | 1.00 | | | | | | | | |
| Sr | 0.85 | −0.81 | 0.94 | −0.86 | −0.44 | 0.01 | 0.75 | 0.97 | 0.28 | 0.12 | 0.02 | −0.03 | −0.20 | −0.50 | −0.63 | 0.13 | −0.58 | −0.17 | −0.20 | −0.61 | −0.40 | −0.24 | −0.41 | 0.19 | −0.01 | −0.53 | 0.28 | −0.03 | 1.00 | | | | | | | |
| Ta | 0.24 | −0.18 | −0.25 | −0.23 | 0.81 | 0.41 | 0.45 | −0.03 | 0.12 | 0.94 | 0.28 | 0.04 | −0.11 | −0.04 | 0.00 | 0.95 | −0.43 | −0.19 | −0.10 | −0.40 | 0.83 | 0.57 | 0.87 | 0.79 | 0.99 | −0.35 | 0.15 | 0.85 | −0.01 | 1.00 | | | | | | |
| Th | 0.27 | −0.23 | −0.19 | −0.24 | 0.68 | 0.32 | 0.40 | 0.00 | 0.41 | 0.80 | 0.12 | 0.04 | −0.18 | −0.19 | −0.11 | 0.83 | −0.33 | −0.28 | −0.08 | −0.40 | 0.68 | 0.51 | 0.75 | 0.72 | 0.83 | −0.38 | 0.29 | 0.69 | 0.00 | 0.87 | 1.00 | | | | | |
| U | −0.67 | 0.63 | −0.64 | 0.65 | 0.28 | −0.15 | −0.53 | −0.66 | −0.09 | −0.09 | 0.13 | 0.06 | 0.32 | 0.46 | −0.09 | 0.29 | −0.03 | 0.09 | 0.47 | 0.21 | 0.18 | 0.30 | −0.08 | 0.03 | 0.32 | 0.34 | −0.02 | −0.73 | 0.04 | 0.04 | 0.04 | 1.00 | | | | |
| V | −0.43 | 0.39 | −0.28 | 0.42 | −0.08 | −0.13 | −0.44 | −0.38 | −0.21 | −0.39 | −0.09 | 0.01 | 0.10 | 0.16 | 0.39 | −0.40 | 0.70 | 0.32 | 0.15 | 0.42 | −0.18 | −0.23 | −0.19 | −0.26 | −0.40 | 0.37 | −0.08 | −0.39 | −0.35 | −0.42 | −0.35 | 0.17 | 1.00 | | | |
| Y | 0.36 | −0.30 | −0.12 | −0.36 | 0.73 | 0.50 | 0.59 | 0.09 | −0.02 | 0.99 | 0.23 | −0.06 | −0.21 | −0.13 | −0.09 | 1.00 | −0.51 | −0.18 | −0.21 | −0.45 | 0.78 | 0.48 | 0.77 | 0.83 | 0.95 | −0.42 | −0.06 | 0.89 | 0.11 | 0.95 | 0.82 | −0.09 | −0.41 | 1.00 | | |
| Zn | −0.92 | 0.88 | −0.79 | 0.94 | 0.35 | −0.34 | −0.77 | −0.90 | −0.36 | −0.08 | 0.03 | 0.09 | 0.34 | 0.53 | 0.64 | −0.10 | 0.56 | 0.17 | 0.26 | 0.80 | 0.46 | 0.18 | 0.40 | −0.15 | 0.03 | 0.64 | 0.18 | −0.15 | −0.84 | 0.01 | −0.06 | 0.64 | 0.31 | −0.11 | 1.00 | |
| Zr | 0.32 | −0.25 | −0.12 | −0.32 | 0.67 | 0.42 | 0.52 | 0.08 | −0.03 | 0.92 | 0.29 | 0.01 | −0.12 | −0.14 | −0.09 | 0.94 | −0.46 | −0.15 | −0.11 | −0.39 | 0.74 | 0.54 | 0.76 | 0.80 | 0.92 | −0.35 | −0.09 | 0.81 | 0.11 | 0.92 | 0.78 | −0.04 | −0.39 | 0.93 | −0.08 | 1.00 |

Note: Pearson's correlation was based on 39 samples with Fe-Ti-V oxide and polymetallic sulphide mineralization. Coefficients with absolute values greater than 0.26 are statistically significant at the 95% level.

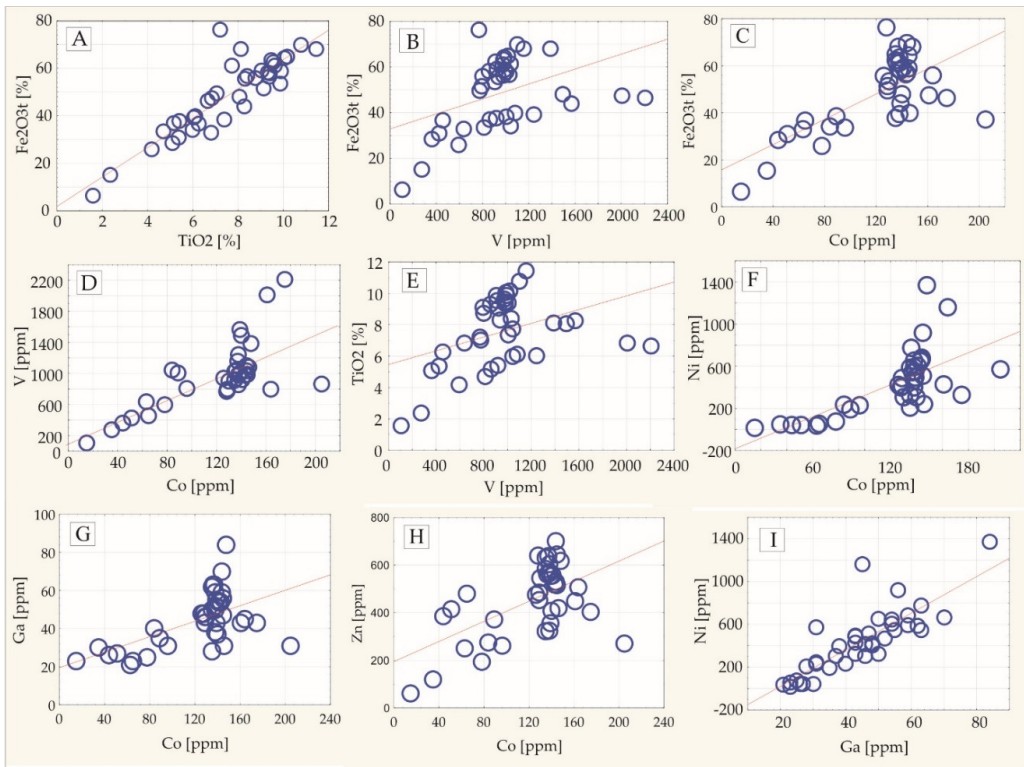

**Figure 16.** Bivariate diagrams of trace elements with trend lines in magnetite-ilmenite ores from the Krzemianka and Udryn Fe-Ti-V deposits on the Suwałki Anorthosite Massif in NE Poland. (**A**) $TiO_2$ vs. $Fe_2O_3t$; (**B**) V vs. $Fe_2O_3t$; (**C**) Co vs. $Fe_2O_3t$; (**D**) Co vs. V; (**E**) V vs. $TiO_2$; (**F**) Co vs. Ni; (**G**) Co vs. Ga; (**H**) Co vs. Zn; (**I**) Ga vs. Ni. Note: the line of best fit is shown as the red line.

The base metals that appeared with magnetite ilmenite ores were mainly related to the presence of magmatic sulphides, e.g., pyrrhotite, pentlandite, and chalcopyrite. The cobalt concentration ranged from 15 to 205 ppm. Samples with cobalt contents from 120 ppm to 180 ($n = 23$) were the most abundant. The average arithmetic content of Co was 122.3 ppm ($n = 39$). Cobalt showed a good correlation with V ($r = 0.70$), (Figure 16D), $Fe_2O_3t$, and Ni ($r = 0.69$; Figure 16F) and a weaker correlation with $TiO_2$, Ga ($r = 0.6$, Figure 16G), and Zn ($r = 0.56$; Figure 16H). The contents of Cu and Ni in bulk-rock samples ranged from 0.18% to 0.14%, respectively. Copper was present in elevated concentrations, ranging from 38 to 1756 ppm. The Cu concentration was <400 ppm in most of the samples ($n = 30$), and only nine samples had contents >400 ppm. The arithmetic mean was 329 ppm ($n = 39$). Copper showed a strong positive correlation with Au ($r = 0.85$) and a very weak correlation with Ag, Ni, and Co ($r = 0.65–0.50$). Nickel concentrations above 0.06%, which were found in 25% of the population, can also be taken into account. Nickel contents ranged from 16 to 1370 ppm, and the arithmetic mean = 436.4 ppm (Table 7). Nickel showed a strong correlation with Ga ($r = 0.82$) and $Fe_2O_3$ and a weaker correlation with Co, $TiO_2$, Zn, and Cu. Zinc contents ranged from 62 to 701 ppm, with an arithmetic mean of 453 ppm. Zinc showed a positive correlation with a few elements (Table 6). The occurrence of ore mineralization was also associated with an increase in the concentration of chromium, up to 0.57%. In 25% of the sample population, the Cr content ranged from >0.1% to 0.571%. The arithmetic mean for Cr was 678.2 ppm. Chromium showed no correlation with other metals (Table 8).

Elements defined as critical (e.g., Co, Ga, Ge, and In) were associated mainly with sulphide mineralization. Gallium contents ranged from 21 to 84 ppm. Most samples had Ga contents in the range of 30 to 60 ppm ($n = 26$). The arithmetic mean for Ga was 44.3 ppm ($n = 39$) and it showed a strong correlation with $Fe_2O_3t$ ($r = 0.87$), Ni ($r = 0.82$, Figure 16I), Zn ($r = 0.80$), and $TiO_2$ ($r = 0.75$), and a weaker one with Co and Cd ($r = 0.57$). Indium concentrations ranged from 0.03 to 0.30 ppm. Most of the samples showed contents from

0.1 to 0.15 ppm of In. The arithmetic mean was 0.13 ppm (*n* = 39). Indium showed a very strong correlation with Ge and MnO (*r* = 0.92). Germanium was present in the range of <0.1–0.5 ppm. About 50% of the sample population was between 0.2 and 0.3 ppm (*n* = 20). The arithmetic mean for Ge was 0.29 ppm (n = 39). Germanium showed a very strong correlation with In (*r* = 0.92), and a strong correlation with Nb (*r* = 0.84) and Ta (*r* = 0.83). Antimony and tellurium concentrations were mostly below detection levels (e.g., 0.5 ppm). Gold concentrations ranged from <1 ppb to 37 ppb. Platinum was present in very low concentrations; only 30% of the sample population showed Pt contents in the range of 10–22 ppb. Likewise, trace concentrations were also present for Pd (>5 ppb), Ag (<0.3 to 2.5 ppm), Nb (5–32 ppm), and Hf (0.38 to 2.09 ppm). Scandium contents ranged from 11 to 75 ppm. The arithmetic mean for scandium was 24.3 ppm.

Among the REEs, the highest arithmetic mean was found for cerium, at only 34 ppm, with a range from 1.3 to 285 ppm. The sum of the light REEs (La to Gd) ranged from 12.0 to 697.8 ppm, and the arithmetic mean was 97 ppm. On the other hand, the sum of the heavy HREEs (from Tb-Lu + Y) was very low, ranging from approximately 0.05 to 4.35 ppm, and the arithmetic mean was 0.53 ppm. The ratio of LREE to HREE ranged from 0.3 to 23.6 (mean = 3). Yttrium contents were also low, in the range of <0.5 to 105.6 ppm, and the arithmetic mean was 11.4 ppm (*n* = 39). All of the samples from magnetite-ilmenite ore showed a similar pattern, with a LREE/HREE fractionation and a negative Eu anomaly and a positive Sm anomaly (Figure 17). These are characteristic REE patterns for a magmatic origin of magnetite-apatite ores [42]. The fractionation of Fe-Ti-oxides was important in the ore-bearing rock samples and this indicates that these ores had a common origin. In addition, there was significant (>400×) enrichment in the REE (especially in Ce, La and Nd) in three mineralized samples from the Udryn deposit. The REE concentrations in these three samples were elevated (ΣREE = 0.1%) and associated with enrichments in $P_2O_5$, CaO, and MgO, as well as to the presence of carbonatites related to Variscan alkaline magmatism [32,34]. REEs spanned a relatively large interval of concentrations, which can be explained partly by fractional crystallization and partly by the varying amounts of apatite crystallized from trapped liquid (these samples had the highest REE and $P_2O_5$ contents). Anorthosites and norites exhibited compositions typical of cumulate rocks. The contents of elements incompatible with plagioclase and of transition elements were low and were mostly controlled by the mafic mineral content [29].

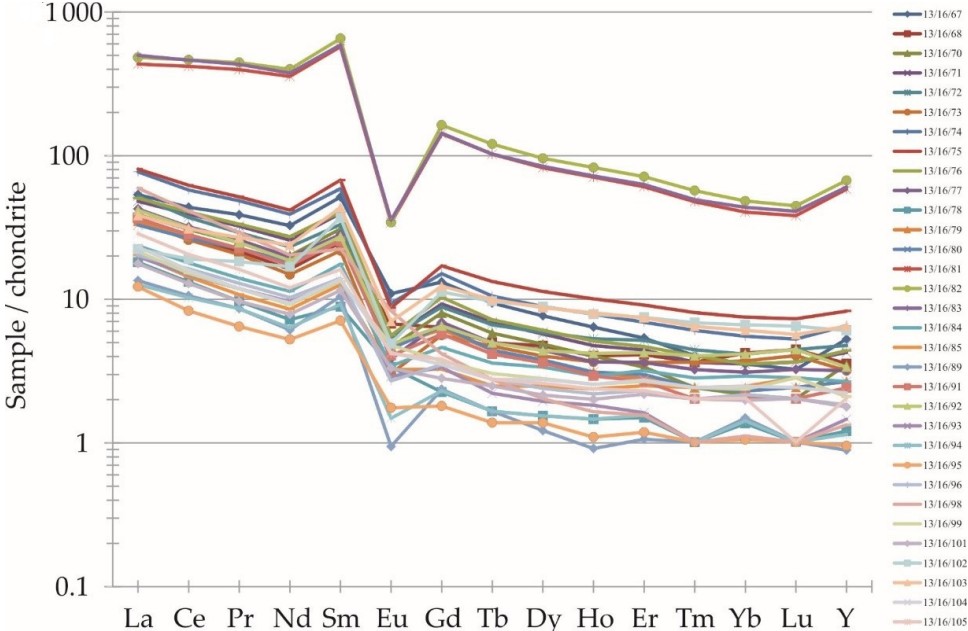

**Figure 17.** REE diagram normalized to chondrite data after [43] for the Fe-Ti-V ore samples from the Krzemianka and Udryn deposits in NE Poland.

## 5. Discussion

### 5.1. Genetic Implications for Vanadium and Cobalt Occurrence in the SAM

The origin of the Fe-Ti-V oxide mineralization hosted by Proterozoic massif-type anorthosite has been explained through different processes, such as fractional crystallization [44–46], magma mixing [47], immiscibility between silicate and oxide magmas [48], stress-driven melt segregation [49], solid-state remobilization [50,51], and hydrothermal remobilization [52]. For the Fe-Ti-V mineralization in the SAM, it was suggested that oxide-silicate melt moved from the deeper parts into the faulted zones of the anorthosite intrusion [16,23]. Moreover, magnetite-ilmenite melt with sulphides followed the same route due to activity of convection flow [29,46]. The Fe-Ti-oxide and sulphide mineralization in SAM was formed from a common parental magma through separate processes [16,53].

In bulk-rock samples, the mean arithmetic content of vanadium and cobalt was 960 ppm (0.17 $V_2O_5$) and 122 ppm, respectively. The dominant ore minerals were magnetite, ilmenite and Al-spinels with subordinate quantities (1–3% of rock volume) of pyrrhotite (c.a. 80% of total sulphides) and pentlandite and chalcopyrite. These ore minerals were formed in the magmatic stage. The early-magmatic stage included the main concentrations of the oxide minerals of the titanomagnetite–ulvöspinel–ilmenite series (in the temperature range of 575 °C–700 °C), and the late-magmatic stage included sulphides crystallizing at high temperatures (~600 °C), such as pyrrhotite, pentlandite, and chalcopyrite [26]. Various structures of ilmenite and spinel separation in magnetite were common, as well as the structures of these minerals, resulting from the decomposition of solid solutions during differentiation in the primary igneous associations during the change of PT conditions [23,26,27,29,41]. In the post-magmatic stage, in connection with the changes in the primary mineral composition, several processes were recognized, such as oxidation separation (temperature range 700 °C–400 °C); separation (500 °C–200 °C); and deuteric (400 °C), hydrothermal, and hypergenic transformations [23,27,29,41].

Vanadium and cobalt have geochemically different characteristics. Vanadium has oxyphilic properties and has been found to be concentrated during the initial stage of crystallization of metallic minerals in magma. Cobalt is sulphophilic and it crystallizes with other Fe, Cu, and Ni sulphides in the later stages of magma differentiation. However, in our bulk-rock geochemical study, cobalt showed a good correlation with vanadium, $Fe_2O_3t$, and nickel ($r$ = ~0.70) and a weaker correlation with $TiO_2$ ($r$ = 0.60). This indicates not only that there was a close spatial coexistence of magnetite-ilmenite mineralization, but also a common source of metals related to the differentiation of anorthosite-norite magma in the AMCG.

Based on the EPMA results, the contents of vanadium in magnetite, ilmenite, and Al-spinels were highly variable. Magnetite, on average, contained 0.42 wt% V (0.75 $V_2O_5$). With progressive oxidation and recrystallization, vanadium moved gradually into magnetite. Vanadium was an isomorphic admixture in magnetite, substituting for $Fe^{+3}$ in its crystal lattice. The vanadium content varied slightly in magnetite, in contrast to the variable Ti content which was a substitution for $Fe^{+2}$. Vanadium had a weak positive correlation with Ti and Cr in the magnetite (Figure 14C,D). The redistribution of V and $Fe^{3+}$ substitution ions occurred under variable PT conditions during the evolution of the anorthosite massif [24,44]. Ilmenite contained 0.14 wt% V on average (0.25 $V_2O_5$). We recognized a slight enrichment of V in massive ilmenites (0.17 to 0.24 wt%) with an increase in the Fe content (36–40 wt%) when compared to ilmenite exsolutions' lamellae in magnetite or to ilmenite pseudomorphs after magnetite. Generally, oxide minerals that have the highest Fe/Ti ratio are the richest in (V + Cr + Al), whereas those with the lowest Fe/Ti ratio are the richest in (Mg + Mn) [29,54–56].

Cobalt was concentrated mainly as isomorphic substitutions in magmatic pentlandite, pyrrhotite, and chalcopyrite and in secondary minerals such as siegenite, pyrite, bravoite, cubanite, and talnakhite. A constant admixture of cobalt in magmatic pentlandite was observed, ranging from 1 to 15.3 wt% (mean = 4.4 wt%). However, higher cobalt contents at the level of 14 to 30 wt% were found in siegenite, and some lower substitution of Co was

observed at the level of 3–4 wt% in talnakhite and 1–3 wt% in bravoite. The concentrations of cobalt in pyrrhotite and chalcopyrite were one order lower, in the range of 0.1–0.6 wt% (means = 0.16 and 0.11 wt%, respectively). Pyrrhotite and chalcopyrite also contained nickel substitutions, with mean values of 0.6 and 0.11 wt%, respectively. The low amounts of Co substitutions were also found in cubanite replacing chalcopyrite (maximum 0.14 wt%). The decomposition processes resulted in the disintegration of sulphide solutions such as pyrrhotite ± pentlandite ± chalcopyrite and the formation of subsequent generations of Fe, Cu, Ni, and Co sulphides and thiospinels *((Fe Ni) (Co, Ni)$_2$S$_4$)*. The proportions of metals indicated the presence of diadochid substitutions and/or micro overgrowths of several transition phases in pentlandite or pyrrhotite. According to [41], the transition phases represent a mixture of different thiospinels *((FeCo)$_2$S$_4$, (NiCo)$_2$S$_4$, and (FeNi)$_2$S$_4$))* exsoluted from pentlandite. They described the decomposition of pentlandite into unidentified thiospinels *(Fe,Ni)(Co,Ni)$_2$S$_4$* and Ni-mackinawite, that may later cause the formation of Co-rich pyrite [41]. The presence of Co-rich pyrite was also recognized during our EPMA, but we additionally identified siegenite, bravoite, cubanite, and talnakhite (Tables 3 and 4, Figure 11). Cobaltiferous pyrite contained from 1 to 8.6 wt% Co (mean = ~2 wt%). In the case of the pyrite replacing pyrrhotite, there was no visible increase in cobalt concentration (<0.1 wt%) compared to primary pyrrhotite.

According to [24], during the first stage of accessory or segregation at low contents of interstitial liquid, one can observe high $f$O$_2$, and gradual cooling metallic minerals crystallized with silicates. During the second process, Fe-Ti-rich anorthosite-norite magma formed and metallic minerals settled at the bottom of the magma chamber at high temperature and an intermediate oxygen partial pressure. The injection of Fe-Ti-rich magma or tectonic introduction of the partially solidified ore chamber into the upper part of the solidified pluton, followed by an immiscible segregation process, explains the concordant character of major Fe-Ti-V ore bodies with disseminated sulphides (Fe, Cu, Co, Ni) [46,56]. Parental ferrodioritic (jotunite) magma of the SAM anorthosite, enriched in plagioclase and orthopyroxene under high-pressure conditions, is saturated in plagioclase and magnetite-ilmenite assemblage at the pressure of crystallization of Fe-Ti oxide emplacement [29]. The REE concentrations in the studied ore samples, normalized to chondrite, had very characteristic clear negative Eu and positive Sm anomalies, indicating the common magmatic origin of Fe-Ti-oxides and sulphide mineralizations hosted by anorthosites and norites.

We suggest that the crystal sorting of Fe-Ti oxide minerals was responsible for the formation of the SAM Fe-Ti-V deposits relative to plagioclase and ferromagnesian silicates [46,57]. Moreover, diapiric uprising of the anorthosite crystal mush favored the sorting during Fe-Ti-enriched cumulate crystallization. Polybaric conditions of mineral crystallization can be shown by the variable Al$_2$O$_3$ content in orthopyroxene [46]. These petrological diagnostic features point to a crustal origin, a conclusion supported by Sm–Nd and Re–Os isotopic data [16,46,53]. The Re-Os model age obtained for pyrrhotite and magnetite give an age of 1536 ± 67 million years [29,37,38]. The time range of their crystallization is consistent with the results of U-Pb determination via the zircon SHRIMPIIe method [15] which indicates that the SAM anorthosites are the result of several magmatic pulses, culminating ca. 1515 Ma and 1507 Ma. The development of post-ore tectonics caused the division of Fe-Ti-V ore fields into separate deposits and the network of discontinuities and fault zones were filled by post-ore *S*-type granites. The crystallization ages of individual granite veins range between 1489 ± 6 Ma and 1475 ± 5 Ma [15]. Hydrothermal processes were probably associated with these granite veins, which were also responsible for magmatic sulphide replacement in the Fe-Ti-V orebodies. The contribution of hydrothermal and pneumatolitic processes in general only locally influenced the major magmatic metallic mineralization. New hydrothermal base metal sulphides formed, such as chalcopyrite, sphalerite and galena, associated with accessory ore minerals (greenockite, hessite, and native bismuth). The enrichment in cadmium of the studied samples—by over 70 times compared to the average Cd content in the earth's crust according to [58]—is also noteworthy. Cadmium is associated with the presence of fine-grained sphalerite that

contains substitutions of Cd, as well as the presence of greenockite (*CdS*) associated with chalcopyrite. They are minerals of hydrothermal origin.

*5.2. Economic Potential of Vanadium and Cobalt in the SAM*

The Fe-Ti-V mineralization in the Krzemianka and Udryn deposits has been considered to be sub-economic since 1996, despite their large geological resources. Identified resources comprise a total of 1.34 billion tons of Fe-Ti-V ore, containing 388.2 million tons of iron and 98 million tons of titanium [30]. The laboratory tests carried out in the 1970s showed the possibility of obtaining separate concentrates of magnetite, ilmenite, and sulphides from these ores. The main component of the sulphide concentrates was pyrrhotite, which was definitely dominant among sulphides and contained, on average, approx. 0.16 wt% Co and 0.61 wt% Ni. These metals can be recovered from the sulphide concentrate through further technological processing [23]. It should be noted that the vanadium contents obtainable in the magnetite and ilmenite concentrates amounted to approximately 0.75 wt% and 0.2 wt% $V_2O_5$, respectively [29]. The vanadium resources were roughly documented at ca. 4.1 million tons, with an average grade of 0.26–0.31% $V_2O_5$. Our geochemical bulk-rock studies of samples from the Krzemianka and Udryn Fe-Ti-V deposits hosted by the Suwałki Anorthosite Massif showed that they were enriched in some elements relative to the respective average elemental contents in the earth's crust [36]. Among them there were also critical elements such as Co and V, and other accompanying metals, represented by Ni, Cu, and Cr, which show only about six times the enrichment in relation to their average content in the earth's crust. The resources of Co were roughly estimated to be greater than 150,000 tons and those of Zn, Ni, Cu, and Cr were up to several hundred thousand tons each [33,59].

It is important to note that, according to the latest estimates—which, however, were made almost two decades ago—the marginal $V_2O_5$ value in economic ore should be 0.73% $V_2O_5$. This means that, according to this criterion, the resources of the two deposits in the SAM would amount to only 1% of the previously documented resources [31].

There are several known examples of this type of magmatic Fe-Ti-V oxide deposits in the world, located in Proterozoic anorthosite massifs and at much more favorable depth conditions and without strong environmental restrictions. They have been discovered in outcrops or shallow sub-surface formations in Africa, Canada, the USA, and Scandinavia [60–65]. The vanadium mean contents in these deposits are typically 0.1–1% $V_2O_5$ and these vary, for example, in Canada's Charles and Buttercup deposits (0.1 and 0.67% $V_2O_5$, respectively) [66]. Cobalt is also recognized as a strategic metal which is crucial for the world's development and transition to a low-carbon economy. Orthomagmatic Ni-Cu-Co sulphide deposits are one of the three main sources of cobalt supplied to the world economy [6]. However, the Fe-Ti-V magmatic deposits hosted by anorthosite complexes are impoverished in terms of their concentrations of Ni, Co, and Cu when compared with other orthomagmatic resources, despite the fact that in some cases they may contain local enrichments of these elements in sulphides and are strongly associated with magnetite-ilmenite oxide mineralization. In the SAM they are considered to be sub-economic because of the depth of the occurrence of ores; their insufficient metal contents, and environmental protection aspects.

## 6. Conclusions

On the basis of detailed geochemical (ICP-MS, WDS-XRF, and GF AAS) and mineralogical (EPMA and SEM) investigations, concentrations of critical elements, such as V and Co, in magnetite-ilmenite and associated sulphide mineralizations from the Krzemianka and Udryn Fe-Ti-V deposits in the Suwałki Anorthosite Massif (NE Poland) were recognized. The mean vanadium and cobalt contents in bulk-rock samples were 0.175% $V_2O_5$ and 0.012% Co, respectively. The main minerals carrying vanadium were magnetite (mean = 0.75 wt% $V_2O_5$) and ilmenite (mean = 0.25 wt% $V_2O_5$). Cobalt occurred mainly as isomorphic substitutions in magmatic pentlandite (mean = 4.4 wt%), pyrrhotite (0.16 wt%),

and chalcopyrite (0.11 wt%). Cobalt was also locally present in the form of different thiospinels *((Fe, Ni) (Co, Ni)$_2$S$_4$)* replacing pyrrhotite-pentlandite solid solutions. The most common of these was siegenite. Moreover, the identified secondary pyrite and bravoite also revealed constant enrichments of Co (*ca.* 2 wt%). Magmatic sulphides were the main source of cobalt, which was locally redistributed during the post-magmatic processes (deuteric, hydrothermal, and hypergenic) and incorporated by the secondary minerals.

**Author Contributions:** Field work, S.Z.M., K.S. and R.M.; data analysis, S.Z.M. and K.S.; ore microscopy, K.S. and S.Z.M.; figures, S.Z.M., K.S. and R.M.; writing—original draft preparation, S.Z.M. and J.W.; writing—review and editing, S.Z.M. and J.W. All authors have read and agreed to the published version of the manuscript.

**Funding:** This report is part of a project that has received funding from the European Union's Horizon 2020 research and innovation programme under grant agreement number 731166. Scientific work is co-funded by the Geological Surveys and national funds allocated for science within the period 2018–2021 under grant agreement 4091/H2020/2018/2 and by the PGI-NRI through internal grants nos. 61.2905.1802.00.0 and 62.9012.2061.00.

**Institutional Review Board Statement:** Not applicable.

**Informed Consent Statement:** Not applicable.

**Data Availability Statement:** Not applicable.

**Acknowledgments:** We would like to thank the anonymous reviewers for a thorough analysis of our article, which improved its quality.

**Conflicts of Interest:** The authors declare no conflict of interest.

## Appendix A

**Table A1.** Chemical composition of magnetite from the Krzemianka and Udryn Fe-Ti-V deposits in NE Poland identified via EMPA.

| Analytical Point | Fe | Al | Mn | V | Cr | Mg | Ti | O | Total | Fe/Ti | Al + V + Cr | Mg + Mn |
|---|---|---|---|---|---|---|---|---|---|---|---|---|
| KR-24_03_obsz-01_fot-1.1 | 70.436 | 0.285 | 0.027 | 0.324 | 0.16 | 0.068 | 1.006 | 21.394 | 93.751 | 70.0 | 0.769 | 0.095 |
| KR-24_03_obsz-01_fot-1.2 | 69.954 | 0.338 | 0.015 | 0.326 | 0.116 | 0.104 | 1.065 | 21.359 | 93.349 | 65.7 | 0.78 | 0.119 |
| KR-24_03_obsz-01_fot-1.3 | 70.587 | 0.261 | 0.025 | 0.43 | 0.119 | 0.07 | 0.303 | 20.985 | 92.817 | 233.0 | 0.81 | 0.095 |
| KR-24_03_obsz-01_fot-1.4 | 69.974 | 0.402 | 0.052 | 0.453 | 0.175 | 0.109 | 0.468 | 21.108 | 92.783 | 149.5 | 1.03 | 0.161 |
| KR-24_03_obsz-03_fot-6.6 | 69.5 | 0.241 | 0.026 | 0.452 | 0.282 | 0.056 | 1.089 | 21.271 | 93.042 | 63.8 | 0.975 | 0.082 |
| KR-24_03_obsz-03_fot-6.7 | 69.64 | 0.238 | 0.019 | 0.377 | 0.258 | 0.046 | 1.055 | 21.224 | 92.959 | 66.0 | 0.873 | 0.065 |
| KR-24_03_obsz-03_fot-6.8 | 70.639 | 0.334 | 0.033 | 0.477 | 0.112 | 0.066 | 0.31 | 21.1 | 93.165 | 227.9 | 0.923 | 0.099 |
| KR-24_03_obsz-03_fot-6.9 | 70.027 | 0.35 | 0.014 | 0.433 | 0.118 | 0.077 | 0.377 | 20.954 | 92.398 | 185.7 | 0.901 | 0.091 |
| KR-24_03_obsz-03_fot-6.10 | 70.611 | 0.328 | 0.014 | 0.494 | 0.135 | 0.082 | 0.527 | 21.26 | 93.567 | 134.0 | 0.957 | 0.096 |
| KRZ-56_02_obsz-01_fot-1.1 | 70.011 | 0.487 | 0.033 | 0.58 | 0.156 | 0.06 | 0.357 | 21.152 | 92.935 | 196.1 | 1.223 | 0.093 |
| KRZ-56_02_obsz-01_fot-1.2 | 70.66 | 0.276 | b.d.l. | 0.481 | 0.125 | 0.039 | 0.144 | 20.912 | 92.676 | 490.7 | 0.882 | 0.039 |
| KRZ-56_02_obsz-01_fot-1.3 | 70.883 | 0.51 | 0.011 | 0.498 | 0.116 | 0.075 | 0.162 | 21.26 | 93.683 | 437.5 | 1.124 | 0.086 |
| KRZ-56_02_obsz-01_fot-2.6 | 72.198 | 0.238 | 0.005 | 0.55 | 0.09 | 0.021 | 0.074 | 21.272 | 94.483 | 975.6 | 0.878 | 0.026 |
| KRZ-56_02_obsz-01_fot-2.7 | 72.164 | 0.186 | b.d.l. | 0.505 | 0.125 | 0.025 | 0.261 | 21.373 | 94.863 | 276.5 | 0.816 | 0.025 |
| KRZ-56_02_obsz-01_fot-2.8 | 72.002 | 0.229 | 0.014 | 0.515 | 0.142 | 0.033 | 0.145 | 21.289 | 94.442 | 496.6 | 0.886 | 0.047 |
| KRZ-56_02_obsz-01_fot-2.9 | 71.958 | 0.24 | 0.035 | 0.507 | 0.106 | 0.03 | 0.296 | 21.392 | 94.734 | 243.1 | 0.853 | 0.065 |
| KRZ-56_02_obsz-01_fot-2.10 | 71.627 | 0.48 | 0.037 | 0.542 | 0.091 | 0.075 | 0.163 | 21.443 | 94.527 | 439.4 | 1.113 | 0.112 |
| KR-63_05_obsz-02_fot-1.3 | 71.136 | 0.252 | 0.018 | 0.572 | 0.159 | 0.053 | 0.249 | 21.169 | 93.675 | 285.7 | 0.983 | 0.071 |
| KR-63_05_obsz-02_fot-1.4 | 71.634 | 0.18 | 0.012 | 0.584 | 0.122 | 0.025 | 0.116 | 21.139 | 93.927 | 617.5 | 0.886 | 0.037 |
| KR-63_05_obsz-03_fot-1.1 | 71.659 | 0.161 | 0.008 | 0.579 | 0.147 | 0.07 | 0.161 | 21.179 | 94.007 | 445.1 | 0.887 | 0.078 |
| KR-63_05_obsz-03_fot-1.2 | 70.406 | 0.484 | 0.016 | 0.576 | 0.126 | 0.157 | 0.485 | 21.372 | 93.649 | 145.2 | 1.186 | 0.173 |
| KR-63_05_obsz-03_fot-1.3 | 71.237 | 0.175 | 0.001 | 0.557 | 0.125 | 0.048 | 0.238 | 21.091 | 93.54 | 299.3 | 0.857 | 0.049 |
| KR-63_05_obsz-04_fot-1.5 | 68.297 | 1.903 | 0.051 | 0.557 | 0.145 | 0.534 | 0.416 | 22.244 | 94.19 | 164.2 | 2.605 | 0.585 |
| KR-63_05_obsz-04_fot-1.6 | 70.304 | 0.619 | 0.019 | 0.57 | 0.169 | 0.219 | 0.525 | 21.568 | 94.126 | 133.9 | 1.358 | 0.238 |
| KR-63_05_obsz-05_fot-1.10 | 71.631 | 0.231 | 0.034 | 0.583 | 0.135 | 0.088 | 0.15 | 21.268 | 94.269 | 477.5 | 0.949 | 0.122 |
| KR-63_05_obsz-05_fot-1.11 | 70.069 | 0.573 | 0.007 | 0.556 | 0.129 | 0.194 | 0.381 | 21.309 | 93.287 | 183.9 | 1.258 | 0.201 |
| KR-63_05_obsz-05_fot-1.12 | 70.988 | 0.458 | b.d.l. | 0.532 | 0.159 | 0.154 | 0.164 | 21.297 | 93.789 | 432.9 | 1.149 | 0.154 |
| KR-63_05_obsz-08_fot-1.5 | 71.235 | 0.197 | 0.017 | 0.588 | 0.129 | 0.061 | 0.185 | 21.117 | 93.649 | 385.1 | 0.914 | 0.078 |
| KR-63_05_obsz-08_fot-1.6 | 70.437 | 0.446 | 0.015 | 0.569 | 0.103 | 0.15 | 0.263 | 21.19 | 93.223 | 267.8 | 1.118 | 0.165 |
| KR-63_05_obsz-08_fot-1.7 | 71.57 | 0.266 | 0.013 | 0.57 | 0.154 | 0.092 | 0.189 | 21.318 | 94.304 | 378.7 | 0.99 | 0.105 |
| UDR-7_01_obsz-01_fot-1.1 | 69.137 | 0.465 | 0.062 | 0.448 | 0.451 | 0.081 | 1.233 | 21.564 | 93.523 | 56.1 | 1.364 | 0.143 |
| UDR-7_01_obsz-01_fot-1.2 | 69.962 | 0.294 | 0.027 | 0.416 | 0.456 | 0.038 | 1 | 21.434 | 93.719 | 70.0 | 1.166 | 0.065 |
| UDR-7_01_obsz-01_fot-1.3 | 69.586 | 0.405 | 0.028 | 0.498 | 0.441 | 0.068 | 1.036 | 21.499 | 93.642 | 67.2 | 1.344 | 0.096 |
| UDR-7_01_obsz-01_fot-1.4 | 69.769 | 0.498 | 0.019 | 0.523 | 0.442 | 0.092 | 0.916 | 21.582 | 93.934 | 76.2 | 1.463 | 0.111 |

**Table A1.** *Cont.*

| Analytical Point | Fe | Al | Mn | V | Cr | Mg | Ti | O | Total | Fe/Ti | Al + V + Cr | Mg + Mn |
|---|---|---|---|---|---|---|---|---|---|---|---|---|
| UDR-7_01_obsz-01_fot-1.5 | 69.595 | 0.481 | 0.019 | 0.485 | 0.439 | 0.063 | 0.829 | 21.411 | 93.392 | 84.0 | 1.405 | 0.082 |
| UDR-7_01_obsz-01_fot-1.6 | 69.702 | 0.412 | 0.075 | 0.471 | 0.488 | 0.072 | 1.216 | 21.688 | 94.174 | 57.3 | 1.371 | 0.147 |
| UDR-7_01_obsz-01_fot-1.7 | 67.557 | 0.746 | 0.102 | 0.52 | 0.427 | 0.118 | 2.31 | 22.134 | 93.98 | 29.2 | 1.693 | 0.22 |
| UDR-7_01_obsz-01_fot-1.8 | 68.033 | 0.593 | 0.08 | 0.495 | 0.449 | 0.076 | 2.268 | 22.082 | 94.222 | 30.0 | 1.537 | 0.156 |
| UDR-7_01_obsz-01_fot-1.9 | 68.621 | 0.783 | 0.075 | 0.48 | 0.426 | 0.152 | 1.767 | 22.096 | 94.457 | 38.8 | 1.689 | 0.227 |
| UDR-7_01_obsz-01_fot-1.10 | 69.937 | 0.425 | 0.048 | 0.465 | 0.444 | 0.068 | 0.892 | 21.517 | 93.883 | 78.4 | 1.334 | 0.116 |
| UDR-7_01_obsz-02_fot-1.1 | 69.115 | 0.448 | 0.066 | 0.452 | 0.451 | 0.091 | 1.432 | 21.673 | 93.805 | 48.3 | 1.351 | 0.157 |
| UDR-7_01_obsz-02_fot-1.2 | 68.396 | 0.914 | 0.051 | 0.516 | 0.472 | 0.21 | 1.627 | 22.121 | 94.356 | 42.0 | 1.902 | 0.261 |
| UDR-7_01_obsz-02_fot-1.3 | 66.203 | 1.216 | 0.128 | 0.512 | 0.517 | 0.28 | 2.652 | 22.543 | 94.139 | 25.0 | 2.245 | 0.408 |
| UDR-7_01_obsz-03_fot-1.1 | 68.706 | 0.415 | 0.053 | 0.475 | 0.439 | 0.084 | 1.484 | 21.56 | 93.246 | 46.3 | 1.329 | 0.137 |
| UDR-7_01_obsz-03_fot-1.2 | 67.921 | 0.639 | 0.094 | 0.497 | 0.473 | 0.122 | 1.86 | 21.862 | 93.551 | 36.5 | 1.609 | 0.216 |
| UDR-7_01_obsz-03_fot-1.3 | 67.283 | 0.656 | 0.1 | 0.462 | 0.453 | 0.123 | 2.297 | 21.946 | 93.363 | 29.3 | 1.571 | 0.223 |
| UDR-10_3_obsz-03_fot-2.5 | 69.72 | 0.4 | 0.069 | 0.412 | 0.2 | 0.081 | 1.717 | 21.865 | 94.515 | 40.6 | 1.012 | 0.15 |
| UDR-10_3_obsz-03_fot-2.6 | 67.932 | 0.686 | 0.1 | 0.459 | 0.235 | 0.219 | 2.552 | 22.293 | 94.551 | 26.6 | 1.38 | 0.319 |
| UDR-10_3_obsz-03_fot-2.7 | 69.483 | 0.4 | 0.076 | 0.392 | 0.144 | 0.135 | 1.586 | 21.691 | 93.932 | 43.8 | 0.936 | 0.211 |
| UDR-10_3_obsz-03_fot-2.8 | 69.964 | 0.44 | 0.1 | 0.458 | 0.236 | 0.149 | 1.426 | 21.851 | 94.682 | 49.1 | 1.134 | 0.249 |
| UDR-10_3_obsz-03_fot-2.9 | 71.984 | 0.01 | b.d.l. | 0.006 | 0.006 | b.d.l. | 0.387 | 20.916 | 93.349 | 186.0 | 0.022 | b.d.l. |
| DR-10_4_obsz-03_fot-1.5 | 69.612 | 0.272 | b.d.l. | 0.492 | 1.262 | 0.05 | 0.185 | 21.183 | 93.166 | 376.3 | 2.026 | 0.05 |
| UDR-10_4_obsz-03_fot-1.6 | 69.668 | 0.428 | 0.031 | 0.462 | 0.744 | 0.058 | 0.285 | 21.156 | 92.868 | 244.4 | 1.634 | 0.089 |
| UDR-10_4_obsz-04_fot-1.5 | 70.716 | 0.336 | 0.004 | 0.563 | 0.712 | 0.063 | 0.261 | 21.378 | 94.063 | 270.9 | 1.611 | 0.067 |
| UDR-11_05_obsz-04_fot-1.5 | 71.428 | 0.198 | 0.009 | 0.245 | 0.016 | 0.038 | 0.059 | 20.856 | 92.907 | 1210.6 | 0.459 | 0.047 |
| UDR-11_05_obsz-04_fot-1.6 | 71.334 | 0.171 | 0.014 | 0.22 | 0.025 | 0.028 | 0.22 | 20.924 | 93.078 | 324.2 | 0.416 | 0.042 |
| UDR-11_06_obsz-02_fot-1.7 | 71.689 | 0.118 | b.d.l. | 0.153 | 0.023 | 0.033 | 0.094 | 20.851 | 93.049 | 762.6 | 0.294 | 0.033 |
| UDR-11_06_obsz-02_fot-1.8 | 71.301 | 0.139 | 0.002 | 0.211 | 0.028 | 0.039 | 0.059 | 20.77 | 92.684 | 1208.5 | 0.378 | 0.041 |
| UDR-11_06_obsz-02_fot-1.9 | 71.372 | 0.118 | 0.026 | 0.218 | 0.021 | 0.038 | 0.057 | 20.788 | 92.786 | 1252.1 | 0.357 | 0.064 |
| UDR-11_06_obsz-02_fot-1.10 | 71.334 | 0.175 | b.d.l. | 0.188 | 0.061 | 0.048 | 0.066 | 20.84 | 92.88 | 1080.8 | 0.424 | 0.048 |
| UDR-11_06_obsz-02_fot-1.11 | 70.858 | 0.232 | 0.041 | 0.2 | 0.034 | 0.042 | 0.102 | 20.766 | 92.39 | 694.7 | 0.466 | 0.083 |
| UDR-11_06_obsz-04_fot-1.1 | 71.036 | 0.231 | 0.013 | 0.231 | 0.042 | 0.005 | 0.069 | 20.781 | 92.515 | 1029.5 | 0.504 | 0.018 |
| UDR-11_06_obsz-04_fot-1.2 | 71.3 | 0.141 | b.d.l. | 0.225 | 0.018 | 0.037 | 0.026 | 20.76 | 92.65 | 2742.3 | 0.384 | 0.037 |
| UDR-11_06_obsz-04_fot-1.3 | 71.144 | 0.21 | 0.006 | 0.21 | 0.051 | 0.041 | 0.064 | 20.825 | 92.697 | 1111.6 | 0.471 | 0.047 |
| UDR-11_06_obsz-04_fot-1.4 | 71.397 | 0.123 | 0.002 | 0.262 | 0.033 | 0.016 | 0.056 | 20.791 | 92.855 | 1274.9 | 0.418 | 0.018 |
| UDR-11_06_obsz-06_fot-1.1 | 70.791 | 0.172 | 0.013 | 0.214 | 0.051 | 0.028 | 0.062 | 20.659 | 92.072 | 1141.8 | 0.437 | 0.041 |
| UDR-11_06_obsz-06_fot-1.2 | 71.377 | 0.166 | 0.023 | 0.232 | 0.048 | 0.011 | 0.113 | 20.858 | 92.937 | 631.7 | 0.446 | 0.034 |

b.d.l.—below detection limit.

## Appendix B

**Table A2.** Chemical composition of ilmenite from the Krzemianka and Udryn Fe-Ti-V deposits in NE Poland identified via EMPA.

| Analytical Point | Fe | Al | Mn | V | Cr | Mg | Ti | O | Total | Fe/Ti | Al + V + Cr | Mn + Mg |
|---|---|---|---|---|---|---|---|---|---|---|---|---|
| KR-24_03_obsz-01_fot-1.5 | 34.816 | 0.023 | 0.687 | 0.151 | 0.027 | 1.029 | 31.097 | 31.771 | 99.727 | 1.12 | 0.201 | 1.716 |
| KR-24_03_obsz-03_fot-6.11 | 33.898 | 0.03 | 1.179 | 0.16 | 0.006 | 0.897 | 31.864 | 32.056 | 100.191 | 1.06 | 0.196 | 2.076 |
| KR-24_03_obsz-03_fot-6.12 | 34.182 | 0.017 | 0.832 | 0.147 | 0.037 | 1.053 | 31.891 | 32.162 | 100.388 | 1.07 | 0.201 | 1.885 |
| KR-24_03_obsz-03_fot-6.13 | 34.43 | 0.026 | 0.72 | 0.149 | b.d.l. | 1.038 | 31.324 | 31.782 | 99.477 | 1.10 | 0.175 | 1.758 |
| KRZ-56_02_obsz-01_fot-1.4 | 33.751 | 2.994 | 0.785 | 0.111 | 0.125 | 1.115 | 28.613 | 32.547 | 100.156 | 1.18 | 3.23 | 1.9 |
| KRZ-56_02_obsz-01_fot-1.5 | 35.652 | 0.02 | 0.87 | 0.155 | 0.041 | 0.234 | 31.475 | 31.793 | 100.391 | 1.13 | 0.216 | 1.104 |
| KRZ-56_02_obsz-01_fot-2.1 | 35.859 | 0.025 | 0.709 | 0.123 | 0.043 | 0.47 | 29.858 | 30.845 | 97.982 | 1.20 | 0.191 | 1.179 |
| KRZ-56_02_obsz-01_fot-2.2 | 35.583 | 0.003 | 0.76 | 0.118 | 0.018 | 0.507 | 31.219 | 31.716 | 100.081 | 1.14 | 0.139 | 1.267 |
| KR-63_05_obsz-02_fot-1.7 | 33.15 | 2.163 | 0.91 | 0.161 | 0.067 | 1.375 | 29.353 | 32.331 | 99.599 | 1.13 | 2.391 | 2.285 |
| KR-63_05_obsz-02_fot-1.8 | 34.189 | 1.378 | 0.866 | 0.177 | 0.074 | 1.075 | 30.45 | 32.457 | 100.688 | 1.12 | 1.629 | 1.941 |
| KR-63_05_obsz-03_fot-1.4 | 34.242 | 0.017 | 0.741 | 0.117 | b.d.l. | 1.284 | 31.416 | 31.95 | 99.839 | 1.09 | 0.134 | 2.025 |
| KR-63_05_obsz-03_fot-1.5 | 35.033 | 0.027 | 0.645 | 0.111 | 0.022 | 1.226 | 31.196 | 31.984 | 100.332 | 1.12 | 0.16 | 1.871 |
| KR-63_05_obsz-04_fot-1.4 | 34.668 | 0.026 | 0.719 | 0.091 | 0.036 | 1.194 | 31.272 | 31.932 | 100.062 | 1.11 | 0.153 | 1.913 |
| KR-63_05_obsz-05_fot-1.6 | 33.937 | 1.186 | 0.896 | 0.166 | 0.082 | 1.352 | 29.661 | 31.888 | 99.261 | 1.14 | 1.434 | 2.248 |
| KR-63_05_obsz-05_fot-1.7 | 34.769 | 0.038 | 0.813 | 0.143 | 0.042 | 1.037 | 30.583 | 31.45 | 98.931 | 1.14 | 0.223 | 1.85 |
| KR-63_05_obsz-05_fot-1.8 | 34.142 | 0.017 | 0.727 | 0.081 | 0.033 | 1.366 | 31.603 | 32.076 | 100.058 | 1.08 | 0.131 | 2.093 |
| KR-63_05_obsz-05_fot-1.9 | 33.966 | 0.025 | 0.722 | 0.113 | 0.019 | 1.3 | 31.214 | 31.753 | 99.173 | 1.09 | 0.157 | 2.022 |
| KR-63_05_obsz-08_fot-1.1 | 35.008 | 0.024 | 0.66 | 0.125 | 0.003 | 1.127 | 30.302 | 31.318 | 98.699 | 1.16 | 0.152 | 1.787 |
| KR-63_05_obsz-08_fot-1.2 | 34.613 | 0.015 | 0.718 | 0.12 | 0.012 | 1.168 | 31.365 | 31.953 | 100.038 | 1.10 | 0.147 | 1.886 |
| KR-63_05_obsz-08_fot-1.3 | 35.232 | 0.018 | 0.703 | 0.141 | 0.016 | 1.208 | 30.875 | 31.818 | 100.052 | 1.14 | 0.175 | 1.911 |
| KR-63_05_obsz-08_fot-1.4 | 35.008 | 0.032 | 0.661 | 0.132 | 0.019 | 1.211 | 30.839 | 31.755 | 99.77 | 1.14 | 0.183 | 1.872 |
| UDR-7_01_obsz-02_fot-1.4 | 35.119 | b.d.l. | 0.712 | 0.113 | b.d.l | 0.803 | 31.191 | 31.721 | 99.782 | 1.13 | 0.113 | 1.515 |
| UDR-7_01_obsz-02_fot-1.5 | 35.459 | 0.007 | 0.592 | 0.086 | 0.025 | 0.737 | 30.943 | 31.577 | 99.564 | 1.15 | 0.118 | 1.329 |
| UDR-7_01_obsz-02_fot-1.6 | 35.309 | b.d.l. | 0.654 | 0.125 | b.d.l. | 0.651 | 31.45 | 31.848 | 100.177 | 1.12 | 0.125 | 1.305 |
| UDR-7_01_obsz-02_fot-1.7 | 35.122 | 0.051 | 0.639 | 0.049 | 0.005 | 0.689 | 31.568 | 31.883 | 100.1 | 1.11 | 0.105 | 1.328 |
| UDR-7_01_obsz-03_fot-1.4 | 35.19 | 0.038 | 0.974 | 0.152 | 0.051 | 0.373 | 30.946 | 31.439 | 99.259 | 1.14 | 0.241 | 1.347 |
| UDR-7_01_obsz-03_fot-1.5 | 35.495 | 0.019 | 0.951 | 0.149 | 0.027 | 0.336 | 31.009 | 31.505 | 99.582 | 1.14 | 0.195 | 1.287 |
| UDR-7_01_obsz-03_fot-1.6 | 35.526 | 0.011 | 1 | 0.083 | 0.028 | 0.369 | 31.096 | 31.561 | 99.733 | 1.14 | 0.122 | 1.369 |
| UDR-10_3_obsz-03_fot-2.1 | 34.103 | 0.025 | 0.641 | 0.1 | 0.04 | 1.63 | 31.384 | 32.11 | 100.084 | 1.09 | 0.165 | 2.271 |
| UDR-10_3_obsz-03_fot-2.2 | 33.651 | 0.017 | 0.641 | 0.122 | 0.007 | 1.603 | 31.745 | 32.176 | 100.029 | 1.06 | 0.146 | 2.244 |
| UDR-10_3_obsz-03_fot-2.3 | 34.205 | 0.045 | 0.596 | 0.13 | b.d.l. | 1.631 | 31.516 | 32.21 | 100.379 | 1.09 | 0.175 | 2.227 |
| UDR-10_3_obsz-03_fot-2.4 | 33.959 | 0.02 | 0.685 | 0.072 | 0.015 | 1.659 | 31.636 | 32.245 | 100.436 | 1.07 | 0.107 | 2.344 |
| UDR-10_4_obsz-03_fot-1.3 | 35.145 | 0.026 | 0.763 | 0.164 | 0.033 | 0.56 | 31.681 | 31.941 | 100.318 | 1.11 | 0.223 | 1.323 |
| UDR-10_4_obsz-03_fot-1.4 | 34.935 | 0.202 | 0.724 | 0.138 | 0.082 | 0.631 | 31.478 | 31.96 | 100.21 | 1.11 | 0.422 | 1.355 |

**Table A2.** *Cont.*

| Analytical Point | Fe | Al | Mn | V | Cr | Mg | Ti | O | Total | Fe/Ti | Al + V + Cr | Mn + Mg |
|---|---|---|---|---|---|---|---|---|---|---|---|---|
| UDR-10_4_obsz-04_fot-1.9 | 35.536 | 0.119 | 0.954 | 0.172 | 0.073 | 0.411 | 31.27 | 31.854 | 100.448 | 1.14 | 0.364 | 1.365 |
| UDR-10_4_obsz-04_fot-1.10 | 34.149 | 4.388 | 0.81 | 0.122 | 0.586 | 1.189 | 26.678 | 32.885 | 100.889 | 1.28 | 5.096 | 1.999 |
| UDR-10_4_obsz-01_fot-1.5 | 35.32 | 0.01 | 0.776 | 0.141 | 0.027 | 0.392 | 31.523 | 31.764 | 99.999 | 1.12 | 0.178 | 1.168 |
| UDR-10_4_obsz-01_fot-1.6 | 35.792 | 0.018 | 0.688 | 0.112 | b.d.l. | 0.52 | 31.344 | 31.822 | 100.358 | 1.14 | 0.13 | 1.208 |
| UDR-11_04_obsz-02_fot-2.11 | 36.507 | 0.03 | 0.928 | 0.18 | 0.027 | 0.646 | 30.642 | 31.789 | 100.875 | 1.19 | 0.237 | 1.574 |
| UDR-11_04_obsz-01_fot-4.11 | 37.156 | 0.02 | 0.881 | 0.181 | b.d.l. | 0.736 | 30.086 | 31.624 | 100.793 | 1.23 | 0.201 | 1.617 |
| UDR-11_04_obsz-01_fot-4.12 | 38.197 | 0.03 | 0.867 | 0.174 | 0.016 | 0.697 | 28.812 | 31.05 | 99.937 | 1.33 | 0.22 | 1.564 |
| UDR-11_05_obsz-01_fot-1.7 | 37.282 | 0.026 | 0.851 | 0.15 | 0.026 | 0.647 | 29.766 | 31.396 | 100.29 | 1.25 | 0.202 | 1.498 |
| UDR-11_05_obsz-01_fot-1.8 | 36.302 | 0.011 | 0.798 | 0.114 | b.d.l. | 0.797 | 30.146 | 31.38 | 99.615 | 1.20 | 0.125 | 1.595 |
| UDR-11_06_obsz-02_fot-1.2 | 36.005 | 0.027 | 0.941 | 0.177 | 0 | 0.845 | 30.63 | 31.787 | 100.611 | 1.18 | 0.204 | 1.786 |
| UDR-11_06_obsz-02_fot-1.3 | 36.247 | 0.026 | 0.935 | 0.075 | 0.028 | 0.804 | 30.118 | 31.434 | 99.863 | 1.20 | 0.129 | 1.739 |
| UDR-11_06_obsz-02_fot-1.4 | 38.566 | 0.025 | 0.811 | 0.141 | 0.003 | 0.786 | 28.496 | 30.989 | 99.985 | 1.35 | 0.169 | 1.597 |
| UDR-11_06_obsz-02_fot-1.5 | 39.905 | 0.031 | 0.663 | 0.176 | 0.037 | 0.717 | 26.826 | 30.172 | 98.608 | 1.49 | 0.244 | 1.38 |
| UDR-11_06_obsz-02_fot-1.6 | 39.352 | 0.023 | 0.762 | 0.103 | 0.061 | 0.743 | 28.445 | 31.145 | 100.82 | 1.38 | 0.187 | 1.505 |
| UDR-11_06_obsz-04_fot-1.5 | 36.368 | 0.017 | 0.89 | 0.183 | 0.035 | 0.821 | 30.057 | 31.46 | 99.967 | 1.21 | 0.235 | 1.711 |
| UDR-11_06_obsz-04_fot-1.6 | 36.796 | 0.024 | 0.858 | 0.171 | b.d.l. | 0.808 | 29.522 | 31.206 | 99.514 | 1.25 | 0.195 | 1.666 |
| UDR-11_06_obsz-04_fot-1.7 | 36.502 | 0.023 | 0.858 | 0.229 | b.d.l. | 0.806 | 30.188 | 31.606 | 100.415 | 1.21 | 0.252 | 1.664 |
| UDR-11_06_obsz-06_fot-1.3 | 35.281 | 0.093 | 1.009 | 0.117 | 0.024 | 0.295 | 31.791 | 32.033 | 100.78 | 1.11 | 0.234 | 1.304 |
| UDR-11_05_obsz-03_fot-1.7 | 36.408 | 0.029 | 0.889 | 0.147 | 0.031 | 0.745 | 30.365 | 31.612 | 100.376 | 1.20 | 0.207 | 1.634 |
| UDR-11_05_obsz-03_fot-1.8 | 37.015 | 0.021 | 0.878 | 0.179 | b.d.l. | 0.818 | 29.643 | 31.372 | 100.097 | 1.25 | 0.2 | 1.696 |
| UDR-11_05_obsz-04_fot-1.7 | 34.846 | 0.017 | 1.045 | 0.161 | 0.039 | 0.703 | 31.466 | 31.913 | 100.3 | 1.11 | 0.217 | 1.748 |
| UDR-11_05_obsz-04_fot-1.8 | 34.359 | 0.02 | 0.999 | 0.149 | 0.011 | 0.771 | 31.327 | 31.705 | 99.431 | 1.10 | 0.18 | 1.77 |
| UDR-11_05_obsz-04_fot-1.9 | 33.303 | 0.348 | 1.117 | 0.113 | 0.001 | 0.853 | 31.636 | 31.976 | 99.537 | 1.05 | 0.462 | 1.97 |
| UDR-11_06_obsz-02_fot-1.1 | 39.206 | 0.071 | 0.759 | 0.225 | 0.019 | 0.772 | 27.528 | 30.563 | 99.24 | 1.42 | 0.315 | 1.531 |

## Appendix C

**Table A3.** Chemical composition of Al-spinels from the Krzemianka and Udryn Fe-Ti-V deposits in NE Poland identified via EMPA.

| Analytical Point | Fe | Al | Mn | V | Cr | Mg | Ti | Zn | O | Total | Fe/Ti | Al + V + Cr | Mn + Mg |
|---|---|---|---|---|---|---|---|---|---|---|---|---|---|
| KR-24_03_obsz-01_fot-1.6 | 17.956 | 32.652 | 0.128 | 0.029 | 0.172 | 7.44 | 0.179 | 2.254 | 39.913 | 100.782 | 100.3 | 32.853 | 7.568 |
| KR-24_03_obsz-01_fot-1.7 | 17.446 | 32.675 | 0.157 | 0.031 | 0.199 | 7.301 | 0.244 | 2.163 | 39.737 | 100.031 | 71.5 | 32.905 | 7.458 |
| KR-24_03_obsz-03_fot-6.1 | 15.136 | 33.091 | 0.131 | 0.023 | 0.274 | 8.223 | 0.317 | 2.217 | 40.127 | 99.559 | 47.7 | 33.388 | 8.354 |
| KR-24_03_obsz-03_fot-6.2 | 15.922 | 32.714 | 0.127 | 0.046 | 0.294 | 7.955 | 0.427 | 2.084 | 39.896 | 99.479 | 37.3 | 33.054 | 8.082 |
| KR-24_03_obsz-03_fot-6.3 | 15.281 | 33.157 | 0.113 | 0.001 | 0.242 | 8.111 | 0.313 | 2.197 | 40.121 | 99.605 | 48.8 | 33.4 | 8.224 |
| KR-24_03_obsz-03_fot-6.4 | 16.837 | 32.689 | 0.116 | 0.033 | 0.143 | 7.717 | 0.069 | 2.246 | 39.711 | 99.617 | 244.0 | 32.865 | 7.833 |
| KR-24_03_obsz-03_fot-6.5 | 17.716 | 32.332 | 0.136 | 0.041 | 0.157 | 7.119 | 0.026 | 2.141 | 39.202 | 98.9 | 681.4 | 32.53 | 7.255 |
| KRZ-56_02_obsz-01_fot-1.6 | 22.707 | 31.653 | 0.157 | 0.041 | 0.094 | 5.144 | 0.02 | 1.69 | 38.612 | 100.184 | 1135.4 | 31.788 | 5.301 |
| KRZ-56_02_obsz-01_fot-1.7 | 22.74 | 31.296 | 0.196 | 0.065 | 0.088 | 4.852 | 0.009 | 1.467 | 38.045 | 98.775 | 2526.7 | 31.449 | 5.048 |
| KRZ-56_02_obsz-01_fot-2.3 | 21.653 | 32.146 | 0.132 | 0.028 | 0.13 | 5.584 | 0.519 | 1.42 | 39.288 | 100.916 | 41.7 | 32.304 | 5.716 |
| KRZ-56_02_obsz-01_fot-2.4 | 21.852 | 31.73 | 0.172 | 0.038 | 0.13 | 5.556 | 0.61 | 1.286 | 38.998 | 100.39 | 35.8 | 31.898 | 5.728 |
| KRZ-56_02_obsz-01_fot-2.5 | 22.833 | 31.003 | 0.136 | 0.06 | 0.109 | 5.18 | 0.023 | 1.773 | 38.097 | 99.225 | 992.7 | 31.172 | 5.316 |
| KR-63_05_obsz-02_fot-1.5 | 18.734 | 31.2 | 0.131 | 0.063 | 0.137 | 7.849 | 0.029 | 0.946 | 38.678 | 97.781 | 646.0 | 31.4 | 7.98 |
| KR-63_05_obsz-04_fot-1.2 | 17.419 | 32.533 | 0.079 | 0.017 | 0.166 | 8.368 | 0.179 | 0.95 | 39.917 | 99.719 | 97.3 | 32.716 | 8.447 |
| KR-63_05_obsz-04_fot-1.3 | 16.769 | 32.971 | 0.111 | 0.054 | 0.15 | 8.439 | 0.28 | 0.855 | 40.231 | 99.901 | 59.9 | 33.175 | 8.55 |
| KR-63_05_obsz-05_fot-1.1 | 15.522 | 33.757 | 0.076 | 0.057 | 0.186 | 9.205 | 0.078 | 0.387 | 40.826 | 100.185 | 199.0 | 34 | 9.281 |
| KR-63_05_obsz-05_fot-1.2 | 15.406 | 33.842 | 0.098 | 0.052 | 0.204 | 9.352 | 0.092 | 0.413 | 40.993 | 100.513 | 167.5 | 34.098 | 9.45 |
| KR-63_05_obsz-05_fot-1.3 | 14.822 | 34.099 | 0.075 | 0.055 | 0.16 | 9.441 | 0.066 | 0.431 | 41.082 | 100.288 | 224.6 | 34.314 | 9.516 |
| KR-63_05_obsz-05_fot-1.4 | 18.37 | 32.113 | 0.12 | 0.042 | 0.131 | 7.931 | 0.019 | 0.902 | 39.406 | 99.051 | 966.8 | 32.286 | 8.051 |
| KR-63_05_obsz-05_fot-1.5 | 18.331 | 31.976 | 0.117 | 0.072 | 0.118 | 8.062 | 0.032 | 0.941 | 39.399 | 99.131 | 572.8 | 32.166 | 8.179 |
| KR-63_05_obsz-08_fot-1.8 | 17.23 | 32.582 | 0.124 | 0.032 | 0.17 | 8.36 | 0.244 | 0.859 | 39.952 | 99.667 | 70.6 | 32.784 | 8.484 |
| KR-63_05_obsz-08_fot-1.9 | 17.483 | 32.697 | 0.113 | 0.054 | 0.138 | 8.264 | 0.213 | 0.916 | 40.028 | 99.93 | 82.1 | 32.889 | 8.377 |
| UDR-7_01_obsz-01_fot-1.11 | 21.062 | 31.72 | 0.139 | 0.05 | 0.585 | 6.315 | 0.035 | 1.432 | 39.13 | 100.516 | 601.8 | 32.355 | 6.454 |
| UDR-7_01_obsz-01_fot-1.12 | 20.479 | 32.037 | 0.145 | 0.043 | 0.561 | 6.419 | 0.024 | 1.461 | 39.302 | 100.523 | 853.3 | 32.641 | 6.564 |
| UDR-10_4_obsz-03_fot-1.1 | 23.269 | 30.323 | 0.142 | 0.055 | 1.206 | 5.678 | 0.342 | 0.865 | 38.461 | 100.443 | 68.0 | 31.584 | 5.82 |
| UDR-10_4_obsz-03_fot-1.2 | 22.083 | 31.074 | 0.155 | 0.047 | 1.449 | 5.378 | 0.459 | 0.705 | 38.732 | 100.121 | 48.1 | 32.57 | 5.533 |
| UDR-10_4_obsz-04_fot-1.1 | 24.015 | 29.755 | 0.147 | 0.064 | 0.761 | 5.562 | 0.035 | 0.957 | 37.696 | 99.007 | 686.1 | 30.58 | 5.709 |
| UDR-10_4_obsz-04_fot-1.2 | 22.769 | 30.63 | 0.121 | 0.04 | 0.851 | 5.761 | 0.016 | 1.109 | 38.298 | 99.621 | 1423.1 | 31.521 | 5.882 |
| UDR-10_4_obsz-04_fot-1.3 | 22.656 | 31.383 | 0.137 | 0.06 | 0.924 | 5.501 | 0.049 | 0.369 | 38.661 | 99.8 | 462.4 | 32.367 | 5.638 |
| UDR-10_4_obsz-04_fot-1.4 | 22.583 | 30.658 | 0.173 | 0.052 | 0.871 | 5.478 | 0.121 | 0.362 | 38.037 | 98.486 | 186.6 | 31.581 | 5.651 |

## Appendix D

**Table A4.** Chemical composition of pyrrhotite from the Krzemianka and Udryn Fe-Ti-V deposits in NE Poland identified via EMPA.

| Analytical Point | S | Ni | Co | Fe | Total | Analytical Point | S | Ni | Co | Fe | Total |
|---|---|---|---|---|---|---|---|---|---|---|---|
| 10/1 | 39.45 | 0.55 | 0.23 | 59.12 | 99.35 | 4/1 | 38.87 | 0.79 | 0.17 | 59.82 | 99.65 |
| 11/1 | 39.5 | 0.46 | 0.21 | 58.85 | 99.02 | 6/1 | 39.3 | 1.01 | 0.16 | 59.32 | 99.79 |
| 11/1 | 39 | 0.66 | 0.18 | 59.53 | 99.37 | 8/1 | 38.55 | 1.02 | 0.2 | 59.33 | 99.1 |
| 12/1 | 39.27 | 0.76 | 0.12 | 59.22 | 99.37 | 9/1 | 38.61 | 0.99 | 0.14 | 59.4 | 99.14 |
| 6/1 | 38.72 | 0.67 | 0.12 | 59.68 | 99.19 | 10/1 | 38.99 | 0.97 | 0.08 | 59.12 | 99.16 |
| 7/1 | 38.72 | 0.75 | 0.13 | 59.57 | 99.17 | 8/1 | 38.82 | 1 | 0.14 | 59.48 | 99.44 |
| 8/1 | 38.65 | 0.72 | 0.13 | 59.37 | 98.87 | 12/1 | 39.1 | 1 | 0.13 | 59.25 | 99.48 |
| 3/1 | 39.41 | 0.62 | 0.1 | 59.74 | 99.87 | 1/1 | 39.052 | 0.223 | 0.115 | 60.038 | 99.428 |
| 4/1 | 39.74 | 0.62 | 0.11 | 59.69 | 100.16 | 2/1 | 38.991 | 0.249 | 0.136 | 59.738 | 99.114 |
| 9/1 | 38.58 | 0.76 | 0.16 | 59.61 | 99.11 | 7/1 | 39.117 | 0.258 | 0.099 | 59.08 | 98.554 |
| 10/1 | 39.36 | 0.77 | 0.13 | 59.56 | 99.82 | 5/1 | 38.627 | 0.392 | 0.178 | 60.288 | 99.485 |
| 11/1 | 39.21 | 0.37 | 0.05 | 59.58 | 99.21 | 6/1 | 38.742 | 0.347 | 0.134 | 60.183 | 99.406 |
| 6/1 | 39.23 | 0.6 | 0.16 | 59.56 | 99.55 | 7/1 | 38.883 | 0.303 | 0.171 | 60.215 | 99.572 |
| 7/1 | 39.34 | 0.6 | 0.18 | 59.79 | 99.91 | 3/1 | 39.73 | 0.275 | 0.098 | 59.715 | 99.818 |
| 7/1 | 38.55 | 0.87 | 0.18 | 59.6 | 99.2 | 4/1 | 39.42 | 0.246 | 0.186 | 59.794 | 99.646 |
| 3/1 | 39.02 | 0.65 | 0.12 | 59.36 | 99.15 | 3/1 | 39.685 | 0.635 | 0.109 | 59.315 | 99.744 |
| 4/1 | 39.29 | 0.61 | 0.15 | 59.74 | 99.79 | 1/1 | 39.028 | 0.227 | 0.185 | 59.493 | 98.933 |
| 10/1 | 38.49 | 1 | 0.13 | 59.94 | 99.56 | 2/1 | 39.093 | 0.277 | 0.221 | 59.403 | 98.994 |
| 11/1 | 38.75 | 1.03 | 0.16 | 59.9 | 99.84 | 3/1 | 38.891 | 0.272 | 0.193 | 59.794 | 99.15 |
| 2/1 | 38.34 | 1.6 | 0.15 | 59.02 | 99.11 | 1/1 | 39.478 | 0.26 | 0.179 | 59.396 | 99.313 |
| 3/1 | 38.76 | 4 | 0.09 | 56.13 | 98.98 | 2/1 | 39.542 | 0.225 | 0.264 | 59.564 | 99.595 |
| 10/1 | 38.77 | 0.6 | 0.14 | 59.71 | 99.22 | 1/1 | 39.459 | 0.201 | 0.266 | 59.151 | 99.077 |
| 6/1 | 39.25 | 0.44 | 0.13 | 59.44 | 99.26 | 2/1 | 39.054 | 0.196 | 0.245 | 59.589 | 99.084 |
| 7/1 | 39.22 | 0.31 | 0.11 | 59.75 | 99.39 | 1/1 | 38.969 | 0.243 | 0.116 | 60.198 | 99.526 |
| 9/1 | 39.01 | 0.23 | 0.15 | 59.56 | 98.95 | 2/1 | 38.689 | 0.232 | 0.179 | 60.526 | 99.626 |
| 10/1 | 39.14 | 0.24 | 0.12 | 59.76 | 99.26 | 1/1 | 39.699 | 0.19 | 0.201 | 59.761 | 99.851 |
| 1/1 | 39.04 | 0.46 | 0.15 | 59.26 | 98.91 | 2/1 | 39.751 | 0.213 | 0.21 | 59.684 | 99.858 |
| 2/1 | 39.04 | 0.39 | 0.13 | 59.48 | 99.04 | 9/1 | 38.914 | 0.214 | 0.237 | 60.323 | 99.688 |
| 5/1 | 38.97 | 0.94 | 0.19 | 59.41 | 99.51 | 5/1 | 38.952 | 0.202 | 0.239 | 59.41 | 98.803 |
| 6/1 | 38.76 | 0.94 | 0.16 | 59.26 | 99.12 | 6/1 | 38.996 | 0.217 | 0.278 | 59.065 | 98.556 |
| 3/1 | 39.11 | 0.97 | 0.14 | 59.62 | 99.84 | | | | | | |

# Appendix E

Table A5. Chemical composition of pentlandite from the Krzemianka and Udryn Fe-Ti-V deposits in NE Poland identified via EMPA.

| Analytical Point | S | Cu | Ni | Co | Fe | Total | Analytical Point | S | Cu | Ni | Co | Fe | Total |
|---|---|---|---|---|---|---|---|---|---|---|---|---|---|
| 5/1 | 33.51 | 0.07 | 26.06 | 6.93 | 32.57 | 99.14 | 3/1 | 33.22 | 0.09 | 34.45 | 6.71 | 26.1 | 100.57 |
| 6/1 | 32.15 | 0.09 | 29.24 | 15.3 | 22.74 | 99.52 | 4/1 | 33.44 | 4.38 | 30.33 | 5.97 | 26.62 | 100.74 |
| 7/1 | 32.95 | 0.09 | 29.08 | 15.23 | 22.26 | 99.61 | 5/1 | 33.61 | 0.21 | 33.94 | 7.04 | 25.86 | 100.66 |
| 5/1 | 32.96 | 0.21 | 35.78 | 2.57 | 28.2 | 99.72 | 2/1 | 32.74 | 0.12 | 37.4 | 2.18 | 27.58 | 100.02 |
| 6/1 | 33.21 | 3.2 | 33.01 | 2.44 | 27.95 | 99.81 | 3/1 | 32.73 | 0.07 | 36.95 | 3.38 | 26.99 | 100.12 |
| 1/1 | 32.8 | 0.11 | 36.76 | 1.4 | 28.54 | 99.61 | 4/1 | 32.91 | 0.05 | 36.79 | 3.02 | 26.99 | 99.76 |
| 2/1 | 33.09 | 0.06 | 37.19 | 1.39 | 28.9 | 100.63 | 1/1 | 33.08 | 0.11 | 35.92 | 2.65 | 26.37 | 98.13 |
| 3/1 | 33.2 | 0.11 | 36.33 | 1.22 | 28.19 | 99.05 | 2/1 | 33.25 | 0.01 | 36.11 | 2.74 | 27.1 | 99.21 |
| 4/1 | 35.26 | 0.23 | 26.59 | 1.12 | 37.35 | 100.55 | 3/1 | 33.41 | 0.41 | 36.41 | 2.56 | 27.11 | 99.9 |
| 8/1 | 33.82 | 0.06 | 33.58 | 6.89 | 26.7 | 101.05 | 4/1 | 33.22 | 0.05 | 36.58 | 3.32 | 26.96 | 100.13 |
| 9/1 | 33.5 | 0.03 | 33.49 | 6.94 | 26.64 | 100.6 | 3/1 | 33.03 | 0.19 | 39.22 | 2.82 | 27.12 | 102.38 |
| 1/1 | 32.24 | 0.13 | 33.92 | 7.15 | 26.73 | 100.17 | 4/1 | 33.02 | 0.05 | 38.35 | 3.67 | 27.37 | 102.46 |
| 4/1 | 32.89 | 0.11 | 36.33 | 3.16 | 27.23 | 99.72 | 12/1 | 32.93 | 0.05 | 38.08 | 3.72 | 27.28 | 102.06 |
| 5/1 | 32.7 | 0.07 | 36.42 | 3.09 | 27.14 | 99.42 | 13/1 | 32.72 | 0.07 | 38.42 | 3.73 | 26.95 | 101.89 |
| 6/1 | 32.43 | 0.05 | 36.65 | 2.96 | 27.27 | 99.36 | 5/1 | 33.01 | 0.11 | 36.14 | 3.6 | 27.06 | 99.92 |
| 1/1 | 33.01 | 0.36 | 39.23 | 1.43 | 26.47 | 100.5 | 6/1 | 33.09 | 0.11 | 36.36 | 3.19 | 27.57 | 100.32 |
| 2/1 | 32.74 | 1.05 | 38.06 | 1.55 | 26.28 | 99.68 | 7/1 | 41.19 | 0.08 | 27.63 | 3.6 | 27.22 | 99.72 |
| 3/1 | 32.96 | 2.5 | 36.54 | 1.56 | 26.57 | 100.13 | 3/1 | 32.88 | 0.11 | 36.89 | 2.76 | 27.29 | 99.93 |
| 1/1 | 32.54 | 0.07 | 35.25 | 6.86 | 25.88 | 100.6 | 4/1 | 32.83 | 0.03 | 36.73 | 3.16 | 27.3 | 100.05 |
| 2/1 | 32.5 | 0.16 | 35.14 | 6.8 | 25.77 | 100.37 | 5/1 | 32.81 | 0.04 | 36.71 | 2.99 | 27.33 | 99.88 |
| 3/1 | 32.59 | 0.13 | 35.78 | 5.67 | 26.22 | 100.39 | 6/1 | 33 | 0.09 | 36.67 | 3.11 | 27.31 | 100.18 |
| 6/1 | 32.32 | 0.11 | 35.25 | 6.31 | 26.34 | 100.33 | 7/1 | 39.03 | 0 | 23.7 | 7.67 | 29.29 | 99.69 |
| 7/1 | 32.14 | 0.12 | 34.71 | 6.94 | 25.7 | 99.61 | | | | | | | |

## Appendix F

**Table A6.** Chemical composition of siegenite from the Krzemianka and Udryn Fe-Ti-V deposits in NE Poland identified via EMPA.

| Analytical Point | S | Cu | Ni | Co | Fe | Total |
|---|---|---|---|---|---|---|
| 2/1 | 42.1 | 0.19 | 25.05 | 20.82 | 12.52 | 100.68 |
| 3/1 | 41.59 | 0.03 | 25.1 | 20.9 | 12.77 | 100.39 |
| 4/1 | 41.57 | 0.17 | 25.51 | 19.57 | 14.34 | 101.16 |
| 3/1 | 37.71 | 0.2 | 33.22 | 19.63 | 4.33 | 95.09 |
| 4/1 | 40.37 | 0.16 | 32.13 | 18.84 | 7.28 | 98.78 |
| 1/1 | 40.72 | 0.09 | 25.15 | 23.2 | 10.39 | 99.55 |
| 2/1 | 40.88 | 0.09 | 25.4 | 22.4 | 10.84 | 99.61 |
| 3/1 | 40.86 | 0.05 | 25.47 | 23.04 | 10.23 | 99.65 |
| 1/1 | 41.87 | 0.08 | 25.89 | 21.6 | 10.65 | 100.09 |
| 2/1 | 42.09 | 0.15 | 25.67 | 21.54 | 10.75 | 100.2 |
| 5/1 | 41.44 | b.d.l. | 27.05 | 23.61 | 10.84 | 102.94 |
| 6/1 | 41.54 | b.d.l. | 26.64 | 23.29 | 11.37 | 102.84 |
| 7/1 | 41.84 | 0.05 | 26.63 | 23.31 | 11.39 | 103.22 |
| 1/1 | 42.04 | 0.04 | 25.49 | 21.94 | 11.05 | 100.56 |
| 2/1 | 41.56 | 0.13 | 25.2 | 22.34 | 11.05 | 100.28 |
| 8/1 | 41.53 | 0.07 | 27.85 | 13.84 | 17.21 | 100.5 |
| 8/1 | 40.61 | 0.07 | 22.89 | 29.67 | 6.75 | 99.99 |
| 9/1 | 40.19 | 0.02 | 23.15 | 27.03 | 9.78 | 100.17 |

## Appendix G

**Table A7.** Chemical composition of chalcopyrite from the Krzemianka and Udryn Fe-Ti-V deposits in NE Poland identified via EMPA.

| Analytical Point | S | Zn | Cu | Ni | Co | Fe | Total | Analytical Point | S | Zn | Cu | Ni | Co | Fe | Total |
|---|---|---|---|---|---|---|---|---|---|---|---|---|---|---|---|
| 4/1 | 34.65 | 0.08 | 34.24 | 0.45 | 0.51 | 30.05 | 99.98 | 2/1 | 34.54 | 0.03 | 34.49 | 0.15 | 0.13 | 30.42 | 99.76 |
| 7/1 | 34.52 | 0.1 | 34.26 | 0.24 | 0.2 | 30.26 | 99.58 | 1/1 | 34.6 | 0.05 | 34.96 | 0.05 | 0.05 | 30.24 | 99.95 |
| 8/1 | 34.57 | b.d.l. | 34.65 | 0.08 | 0.09 | 30.25 | 99.64 | 2/1 | 34.57 | 0.16 | 34.7 | 0.04 | 0.04 | 30.43 | 99.94 |
| 9/1 | 34.8 | b.d.l. | 34.38 | 0.04 | 0.09 | 30.85 | 100.16 | 4/1 | 34.34 | 0.05 | 34.66 | 0.01 | 0.1 | 30.38 | 99.54 |
| 9/1 | 33.83 | 0.06 | 33.73 | 0.12 | 0.08 | 31.17 | 98.99 | 5/1 | 34.32 | 1.01 | 33.92 | 0.11 | 0.24 | 30.46 | 100.06 |
| 10/1 | 33.89 | 0.11 | 34.5 | 0.01 | 0.06 | 30.75 | 99.32 | 5/1 | 34.71 | 0.08 | 34.53 | 0.01 | 0.08 | 30.32 | 99.73 |
| 1/1 | 34.43 | 0.09 | 34.87 | 0.01 | 0.06 | 30.95 | 100.41 | 6/1 | 34.61 | 0.13 | 34.63 | 0.01 | 0.07 | 30.38 | 99.83 |
| 2/1 | 34.32 | 0.15 | 34.22 | 0.03 | 0.08 | 30.83 | 99.63 | 7/1 | 34.55 | b.d.l. | 34.71 | 0.01 | 0.08 | 30.52 | 99.87 |

**Table A7.** *Cont.*

| Analytical Point | S | Zn | Cu | Ni | Co | Fe | Total | Analytical Point | S | Zn | Cu | Ni | Co | Fe | Total |
|---|---|---|---|---|---|---|---|---|---|---|---|---|---|---|---|
| 10/1 | 34.38 | 0.07 | 34.75 | 0.08 | 0.07 | 30.6 | 99.95 | 5/1 | 34.31 | 0.02 | 34.6 | 0.14 | 0.19 | 30.46 | 99.72 |
| 11/1 | 34.58 | 0.02 | 35.11 | 0.1 | 0.12 | 30.31 | 100.24 | 1/1 | 34.19 | 0.13 | 35.37 | 0.05 | 0.07 | 30.46 | 100.27 |
| 2/1 | 34.28 | 0.02 | 34.37 | 0.25 | 0.17 | 30.68 | 99.77 | 2/1 | 34.22 | 0.01 | 35.12 | 0.04 | 0.15 | 30.27 | 99.81 |
| 6/1 | 34.74 | 0.03 | 34.69 | 0.04 | 0.12 | 30.05 | 99.67 | 11/1 | 34.15 | 0.01 | 35.17 | 0.16 | 0.17 | 30.09 | 99.75 |
| 7/1 | 34.54 | 0.06 | 34.05 | 0.39 | 0.55 | 30.05 | 99.64 | 3/1 | 34.27 | 0.12 | 34.64 | 0.33 | 0.14 | 30.71 | 100.21 |
| 7/1 | 34.07 | 0.05 | 34.23 | 0.02 | 0.03 | 30.65 | 99.05 | 4/1 | 34.22 | 0.02 | 34.47 | 0.14 | 0.08 | 30.96 | 99.89 |
| 8/1 | 34.15 | 0.1 | 34.37 | 0.02 | 0.05 | 30.5 | 99.19 | 1/1 | 34.2 | 0.05 | 34.61 | 0.42 | 0.16 | 30.44 | 99.88 |
| 9/1 | 34.56 | 0.01 | 34.97 | 0.04 | 0.08 | 30.36 | 100.02 | 2/1 | 34.05 | 0.1 | 34.59 | 0.56 | 0.26 | 30.07 | 99.63 |
| 1/1 | 33.87 | 0.13 | 34.38 | 0.07 | 0.06 | 31.35 | 99.86 | 4/1 | 33.995 | 0.055 | 34.321 | 0.006 | 0.047 | 30.455 | 98.879 |
| 4/1 | 34.03 | 0.03 | 34.94 | 0.19 | 0.03 | 30.47 | 99.69 | 6/1 | 34.265 | 0.031 | 34.158 | 0.047 | 0.074 | 30.76 | 99.335 |
| 5/1 | 34.13 | 0.11 | 35.22 | 0.07 | 0.07 | 30.4 | 100 | 8/1 | 34.443 | 0.114 | 34.55 | 0.01 | 0.071 | 30.491 | 99.679 |
| 4/1 | 33.97 | 0.06 | 34.83 | 0.18 | 0.07 | 30.4 | 99.51 | 9/1 | 34.692 | 0.011 | 34.194 | 0.01 | 0.117 | 30.724 | 99.748 |
| 5/1 | 34.01 | b.d.l. | 34.7 | 0.1 | 0.09 | 30.64 | 99.54 | 4/1 | 34.482 | b.d.l. | 34.051 | 0.075 | 0.063 | 29.875 | 98.546 |
| 6/1 | 34.17 | 0.13 | 34.63 | 0.08 | 0.05 | 30.22 | 99.28 | 3/1 | 34.523 | 0.125 | 34.495 | 0.049 | 0.065 | 30.407 | 99.664 |
| 1/1 | 34.34 | b.d.l. | 34.34 | 0.03 | 0.05 | 31.18 | 99.94 | 3/1 | 34.698 | 0.104 | 34.452 | 0.018 | 0.058 | 30.417 | 99.747 |
| 2/1 | 34.39 | 0.1 | 34.14 | 0.04 | 0.05 | 30.51 | 99.23 | 3/1 | 34.871 | b.d.l. | 34.3 | 0.042 | 0.02 | 30.705 | 99.938 |
| 1/1 | 34.59 | 0.15 | 34.12 | 0.17 | 0.16 | 30.35 | 99.54 | 7/1 | 34.451 | b.d.l. | 33.651 | 0.074 | 0.053 | 30.864 | 99.093 |

# Appendix H

**Table A8.** Chemical composition of cubanite from the Krzemianka and Udryn Fe-Ti-V deposits in NE Poland identified via EMPA.

| Analytical Point | S | Zn | Cu | Ni | Co | Fe | Total |
|---|---|---|---|---|---|---|---|
| 4/1 | 35.11 | 0.13 | 23.22 | 0.14 | 0.14 | 40.79 | 99.53 |
| 5/1 | 35.19 | 0.01 | 23.63 | 0.06 | 0.06 | 40.34 | 99.29 |
| 8/1 | 34.82 | 0.01 | 23.23 | 0.08 | 0.1 | 40.97 | 99.21 |
| 9/1 | 34.95 | 0.01 | 22.24 | 0.09 | 0.1 | 41.98 | 99.37 |
| 3/1 | 35.1 | 0.01 | 23.56 | 0.01 | 0.11 | 41.03 | 99.82 |
| 6/1 | 34.72 | 0.01 | 23.31 | 0.05 | 0.05 | 40.85 | 98.99 |
| 7/1 | 34.9 | 0.06 | 23.21 | 0.01 | 0.09 | 40.96 | 99.23 |
| 8/1 | 34.89 | 0.03 | 22.81 | 0.04 | 0.05 | 40.88 | 98.7 |
| 3/1 | 35 | 0.04 | 23.4 | 0.01 | 0.04 | 40.76 | 99.25 |
| 4/1 | 35.44 | 0.06 | 23.24 | 0.01 | 0.04 | 41.12 | 99.91 |

## Appendix I

**Table A9.** Chemical composition of pyrite from the Krzemianka and Udryn Fe-Ti-V deposits in NE Poland identified via EMPA.

| Analytical Point | S | Zn | Cu | Ni | Co | Fe | Total | Analytical Point | S | Zn | Cu | Ni | Co | Fe | Total |
|---|---|---|---|---|---|---|---|---|---|---|---|---|---|---|---|
| 13/1 | 52.87 | 0.07 | 0.03 | 1.22 | 1.95 | 43.99 | 100.29 | 1/1 | 52.201 | b.d.l. | 0.01 | 0.005 | 1.268 | 46.165 | 99.74 |
| 14/1 | 51.96 | 0.03 | 0.18 | 3.45 | 0.6 | 42.81 | 100.35 | 2/1 | 52.639 | b.d.l. | 0.031 | 0.071 | 1.902 | 45.437 | 100.105 |
| 11/1 | 53.13 | 0.01 | 0.1 | 0.06 | 0.12 | 47.12 | 100.64 | 3/1 | 52.292 | 0.074 | 0.01 | 0.01 | 1.552 | 45.972 | 99.964 |
| 12/1 | 52.8 | b.d.l. | 0.2 | 5.49 | 0.22 | 41.88 | 100.77 | 4/1 | 52.006 | 0.027 | 0.01 | 0.081 | 1.815 | 46.01 | 100.052 |
| 5/1 | 53.11 | 0.03 | 0.09 | 3.61 | 0.2 | 43.21 | 100.37 | 5/1 | 53.069 | 0.039 | 0.047 | 1.192 | 3.314 | 42.133 | 99.957 |
| 6/1 | 53.18 | b.d.l. | 0.02 | 0.48 | 0.13 | 46.83 | 100.67 | 6/1 | 53.099 | 0.028 | 0.01 | 1.23 | 0.113 | 45.376 | 99.897 |
| 1/1 | 52.33 | b.d.l. | 0.01 | 4.98 | 0.37 | 41.68 | 99.45 | 1/1 | 53.001 | 0.01 | 0.074 | 0.022 | 1.054 | 45.853 | 100.05 |
| 2/1 | 52.79 | 0.06 | 0.01 | 1.96 | 0.22 | 44.88 | 100 | 2/1 | 53.492 | 0.124 | 0.033 | 0.28 | 0.128 | 45.97 | 100.085 |
| 8/1 | 52.27 | 0.03 | 0.01 | 1.5 | 0.25 | 44.74 | 99.28 | 4/1 | 53.323 | b.d.l. | 0.056 | 0.062 | 7.063 | 39.434 | 100.05 |
| 12/1 | 53.09 | b.d.l. | 0.08 | 0.01 | 4.59 | 42.04 | 100 | 5/1 | 52.962 | 0.046 | 0.016 | 0.049 | 4.594 | 42.067 | 99.93 |
| 6/1 | 53.56 | 0.05 | 0.18 | 0.09 | 0.13 | 46.07 | 100.15 | 6/1 | 52.846 | 0.003 | 0.03 | 0.009 | 8.608 | 38.347 | 99.911 |
| 7/1 | 53.46 | b.d.l. | 0.21 | 0.1 | 0.11 | 46.04 | 99.96 | 7/1 | 52.794 | 0.045 | 0.042 | 0.04 | 7.059 | 39.602 | 99.724 |
| 7/1 | 53.73 | 0.03 | 0.13 | 0.3 | 0.09 | 45.84 | 100.24 | 3/1 | 53.136 | 0.036 | 0.052 | 1.215 | 0.089 | 45.618 | 100.283 |
| 8/1 | 53.57 | 0.02 | 0.08 | 0.22 | 0.08 | 45.81 | 99.87 | 10/1 | 53.418 | 0.127 | 0.003 | 0.024 | 0.288 | 46.37 | 100.321 |
| 9/1 | 53.17 | 0.07 | 0.2 | 0.86 | 0.07 | 45.62 | 100.17 | 1/1 | 53.977 | 0.01 | 0.017 | 0.112 | 0.123 | 45.972 | 100.394 |
| 8/1 | 53 | 0.06 | 0.06 | 8.72 | 0.14 | 38.02 | 100.1 | 2/1 | 53.927 | 0.084 | 0.001 | 0.301 | 0.064 | 45.56 | 100.073 |
| 9/1 | 53.21 | 0.03 | 0.01 | 0.78 | 3.26 | 43.53 | 100.91 | 3/1 | 53.453 | 0.073 | 0.01 | 0.046 | 0.127 | 45.435 | 99.575 |
| 5/1 | 52.678 | b.d.l. | 0.066 | 0.02 | 1.122 | 45.388 | 99.345 | 4/1 | 52.871 | 0.042 | 0.087 | 0.062 | 0.13 | 45.448 | 99.253 |
| 3/1 | 52.451 | 0.038 | 0.062 | 0.082 | 6.373 | 40.552 | 99.67 | | | | | | | | |

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
