# Peer review of "Vanadium and Cobalt Occurrence in the Fe-Ti-V Oxide Deposits Related to Mesoproterozoic AMCG Complex in NE Poland"

_applsci, doi:10.3390/app12126277_

Round 1
Reviewer 1 Report
This research investigated the formation of V and Co in oxide deposits at the Krzemianka and Udryn deposits in the Mesoproterozoic SuwaÅ‚ki Anorthosite Massif (SAM) in NE Poland, and delivered interesting results for readers, as well as that at the end of the paper the economic potential of the minerals were evaluated from the exploration and utilization of the deposits. Moreover, this article also presented many professional microphotographs and many great analysis results. I believe this study will bring some meaningful reference for future research in this field. Just list little point for this paper’s improvement.
1. Some expressions may be clearer or more accurate, such as line 82: the aim of this article was to …. Please check other places.
2. Line 347 – Table 2. Ferrous content of the magnetite is 77.24%? According to this chemical composition, the structure of this mineral should be wustite instead of spinel – magnetite, and this seems not right from natural logic. Please check the data with similar detection and if there are some issues on your calibration.
Author Response
Response to the Reviewer #1:
Thank you very much for such a nice and insightful review of the article.
Ad. 1. These relevant wording have been revised.
Ad. 2. We are very grateful to the reviewer for noticing the incorrectly stated iron content in the magnetite caused by errors in the calibration of the electron microprobe. Of course, this was an obvious mistake that was completely removed from the original version of the manuscript. All the results were recalculated and the documentation of the analyzes in this regard was changed. We corrected the relevant statistical data for the discussed Fe-Ti-Al spinels both in the text of the manuscript as well as in the table 2 and 3 appendices (nos. A1-A3). Moreover, we verified the content of the respective figures (nos. 8, 9A-B and 10A-B) and corrected them in terms of Fe content in magnetite, ilmenite and Al-spinels. Once again, we are very grateful to the anonymous reviewer for noticing this mistake.
.

Reviewer 2 Report
Excellent job very well done and practically nothing to point out. The authors are to be congratulated for carrying out meticulous work.
Just a small suggestion, not mandatory:
Line 236 - Please provide the model of the XRF device.
Author Response
Response to the Reviewer #2:
Thank you very much for such a very good review. We have completed the information about model on the portable XRF spectrometer. We also corrected the spelling errors we noticed.

Reviewer 3 Report
The explanation of bivariate diagrams should be reconsidered. Because the R2 values are quite low in some. Statistically, the more significant ones should be considered. An explanation should be made taking into account these values.
Author Response
Response to the Reviewer #3:
Thank you very much for such a nice and insightful review of the article. We verified the explanations of bivariate diagrams as suggested by the Reviewer. Low values ​​of the R2 were treated as the lack of correlation between the pairs of chemical elements. Throughout the text, a uniform rule has been adopted. The degree of correlation of parameters was interpreted as follows: r ≤ 0.5 no correlation, r > 0.5 to 0.7 weak correlation, r > 0.7 to 0.9 strong correlation, and r > 0.9 very strong correlation. Corrections of the relevant descriptions have been made.
